# Durable lymph-node expansion is associated with the efficacy of therapeutic vaccination

Alexander J. Najibi [1,2], Ryan S. Lane [3], Miguel C. Sobral[1,2], Giovanni Bovone[2], Shawn Kang[1,2], Benjamin R. Freedman[1,2], Joel Gutierrez Estupinan[1,2], Alberto Elosegui-Artola[1,2,4], Christina M. Tringides[1,2,5], Maxence O. Dellacherie[1,2], Katherine Williams [3], Hamza Ijaz[2], Sören Müller [3], Shannon J. Turley [3] & David J. Mooney [1,2] ✉

Following immunization, lymph nodes dynamically expand and contract. The mechanical and cellular changes enabling the early-stage expansion of lymph nodes have been characterized, yet the durability of such responses and their implications for adaptive immunity and vaccine efficacy are unknown. Here, by leveraging high-frequency ultrasound imaging of the lymph nodes of mice, we report more potent and persistent lymph-node expansion for animals immunized with a mesoporous silica vaccine incorporating a model antigen than for animals given bolus immunization or standard vaccine formulations such as alum, and that durable and robust lymph-node expansion was associated with vaccine efficacy and adaptive immunity for 100 days post-vaccination in a mouse model of melanoma. Immunization altered the mechanical and extracellular-matrix properties of the lymph nodes, drove antigen-dependent proliferation of immune and stromal cells, and altered the transcriptional features of dendritic cells and inflammatory monocytes. Strategies that robustly maintain lymph-node expansion may result in enhanced vaccination outcomes.

Lymph nodes (LNs) undergo dramatic changes in volume and composition as they orchestrate adaptive immune responses to disease and immunization[1-5]. Recently, LN expansion was found to be driven by changes in cellular contractility, proliferation and lymphocyte retention[2,6-10]. Although the expanding LN has been widely explored, most studies do not consider the durability of LN expansion, or its implications on vaccine outcomes. A successful immune response engages a variety of immune and stromal cell types, culminating in lymphocyte proliferation and effector functions. Currently, the time-dependent cellular and transcriptional profiles within LNs beyond the initial phase of LN expansion remain unclear and may inform differential responses to vaccine formulations. In addition, long-term remodelling of the LN matrix may result in changes in LN tissue-scale mechanical properties,

which may further impact cellular behaviour[11,12]. With a diversity of vaccine strategies under exploration[13], discerning the mechanisms underlying durable LN expansion could support vaccine development.

In this work, we assess the cellular and tissue-scale LN responses to 'strong' and 'weak' vaccines with distinct therapeutic outcomes. Vaccine-draining LNs (dLNs) were imaged longitudinally using high-frequency ultrasound (HFUS), a non-invasive technique for visualizing the bulk properties of superficial internal structures with high (~20 μm) axial resolution, to establish the magnitude and kinetics of LN responses[14]. HFUS has previously been used to image and assess human LNs with potential diagnostic capacity[15-17]. Here, HFUS identified a biomaterial-based vaccine formulation eliciting prolonged LN enlargement (months) compared with a standard bolus vaccine,

[1]John A. Paulson School of Engineering and Applied Sciences, Harvard University, Cambridge, MA, USA. [2]Wyss Institute for Biologically Inspired Engineering at Harvard University, Boston, MA, USA. [3]Department of Cancer Immunology, Genentech, Inc., South San Francisco, CA, USA. [4]Institute for Bioengineering of Catalonia, Barcelona, Spain. [5]Harvard Program in Biophysics, Harvard University, Cambridge, MA, USA. ✉e-mail: mooneyd@seas.harvard.edu

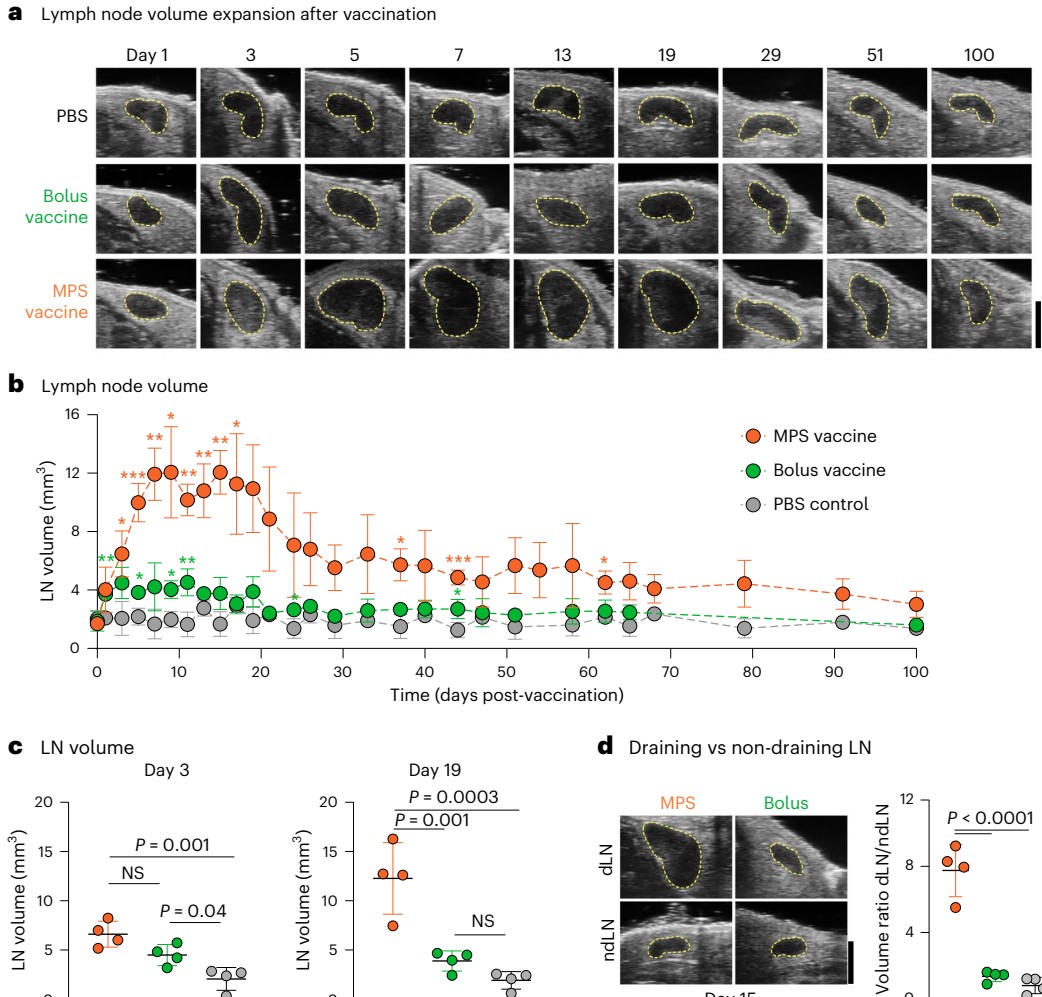

**a** Lymph node volume expansion after vaccination

**Fig. 1 | MPS vaccination induces robust, prolonged LN expansion.** Mice were immunized with MPS or bolus vaccines delivering GM-CSF, CpG and OVA protein, and compared to PBS-injected controls. Vaccine-draining and non-draining LNs were longitudinally imaged using HFUS. **a**, Representative HFUS images of vaccine-draining LNs (defined by yellow dashed area) out to 100 days after vaccination. Scale bar, 2 mm. **b**, Quantification of vaccine-draining LN volume over time. Statistical analysis was performed using a two-way analysis of variance (ANOVA) with repeated measures. Significance relative to the PBS group is depicted at each timepoint (*$P < 0.05$, **$P < 0.01$, ***$P < 0.001$, ****$P < 0.0001$). Exact $P$ values between MPS and PBS are $P = 0.04$, day 3; $P = 0.0008$, day 5;

$P = 0.006$, day 7; $P = 0.01$, day 9; $P = 0.001$, day 11; $P = 0.004$, day 13; $P = 0.004$, day 15; $P = 0.03$, day 17; $P = 0.01$, day 37; $P = 0.001$, day 44; $P = 0.01$, day 62. Exact $P$ values between bolus and PBS are $P = 0.009$, day 1; $P = 0.02$, day 5; $P = 0.03$, day 9; $P = 0.008$, day 11; $P = 0.04$, day 44. **c**, Plots of LN volume among groups on days 3 (left) and 19 (right). Statistical analysis was performed using ANOVA with Tukey's post hoc test. **d**, Representative HFUS images of MPS or bolus vaccine-draining or non-draining LNs 15 days after vaccination (left) and quantification of dLN/ndLN volume ratio (right). Statistical analysis was performed using ANOVA with Tukey's post hoc test. For **a–d**, $n = 7$ (MPS and bolus) or 8 (PBS) biologically independent animals per group, imaged longitudinally in two cohorts; means ± s.d.

enabling investigation of the expanded LN state. LN expansion was then analysed in comparison to the outcomes of immunization (that is, antigen-specific T cell and antibody responses, and therapeutic antitumour efficacy), tissue-scale mechanical and matrix properties, cellular composition and transcriptional signatures. Finally, we engineered LN expansion to directly enhance the efficacy of vaccination.

## Results

### Vaccine formulation alters the durability of LN expansion

First, we identified a vaccine formulation eliciting robust and durable LN expansion. Mesoporous silica (MPS) rod-based vaccines, previously found to elicit strong cellular and humoral responses against diverse antigen targets compared with a traditional bolus (liquid) vaccine, were explored[18–21]. These high-aspect ratio, silica-based nanoparticles can adsorb vaccine antigens and adjuvants for sustained release, and form a three-dimensional scaffold promoting antigen-presenting cell (APC) recruitment in mouse models. MPS vaccines previously induced

potent and long-lived germinal centre responses dependent on sustained antigen release from the vaccine site[22,23]. Here, MPS rods used in vaccine formulation had an average length of 85.9 μm and released vaccine components cytosine-guanosine oligodeoxynucleotide (CpG) and granulocyte-macrophage colony-stimulating factor (GM-CSF) in a sustained manner (Extended Data Fig. 1a–e). Draining (dLN; ipsilateral to vaccine site) and non-draining (ndLN; contralateral) inguinal LNs of mice immunized with MPS or bolus vaccines were imaged for 100 days post-vaccination using HFUS.

Although PBS injection did not affect LN volume, both vaccine formulations induced LN expansion, but with markedly different durability (Fig. 1a,b and Supplementary Fig. 1a–c). At the early stage of expansion (within days), both MPS and bolus-vaccinated mouse LNs expanded to a similar extent (Fig. 1c). However, while the bolus vaccine LNs peaked at this time, resulting in a two-fold transient increase in LN volume, the MPS vaccine induced a significantly more substantial (~7×) LN expansion over 1 week which was maintained for ~3 weeks

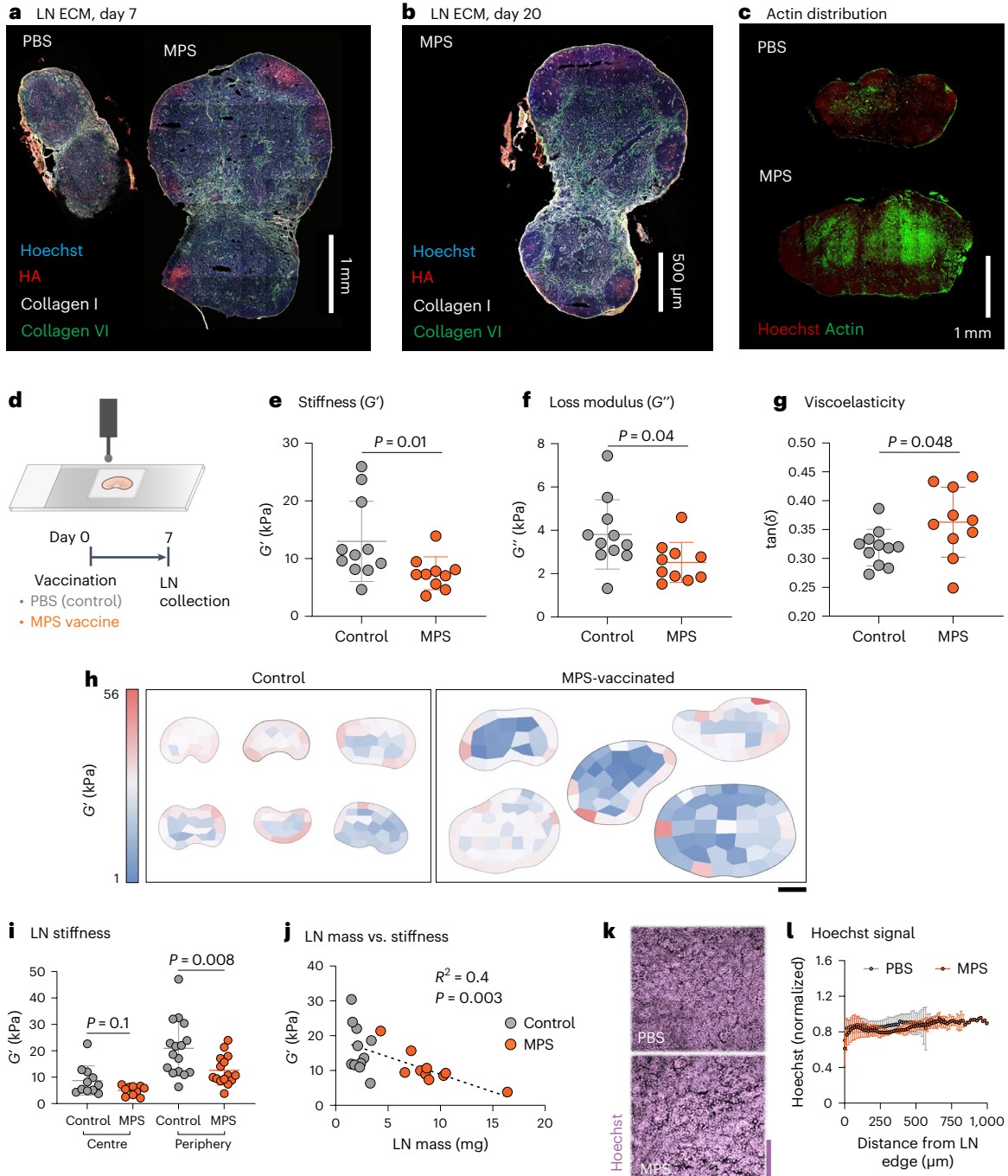

**Fig. 2 | Location-specific alterations in lymph node matrix and mechanics accompany expansion.** Mice were treated with MPS vaccines (delivering GM-CSF, CpG, OVA) or PBS, and LNs were collected after 7 and 20 days. **a**, Representative immunohistochemistry (IHC) images depicting LN ECM on day 7. **b**, Representative IHC image depicting LN ECM on day 20. **c**, Representative IHC images of LNs stained for F-actin on day 7. For **a**–**c**, $n = 3$ biologically independent animals per group. **d**, Schematic depicting nanoindentation of a thick LN slice (above) and experimental timeline (below). **e**–**g**, Mean $G'$ (**e**), $G''$ (**f**) and tan($\delta$) (**g**) across LNs. Statistical analysis was performed using Mann–Whitney test (**e**) or two-tailed $t$-test (**f,g**). **h**, Heat maps depicting $G'$ across individual LNs. Scale bar, 1 mm. **i**, Mean $G'$ of sample points across each LN, separated into those collected at the centre or periphery. $n = 11$ (control, centre), 10 (MPS, centre), 16 (control, periphery) and 15 (MPS, periphery) biologically independent animals per group; results are mean ± s.d., combined from three independent experiments. Statistical analysis was performed using Mann–Whitney test. **j**, Plot of LN mass versus mean $G'$. For **e**–**g**, **i** and **j**, each data point represents a unique LN per mouse; $n = 10$ (MPS) or 11 (PBS) biologically independent animals per group; mean ± s.d., combined from two independent experiments. **k**, Representative IHC images depicting Hoechst stain within LNs on day 7. Scale bar, 100 μm. **l**, Quantification of Hoechst signal across LNs; $n = 3$ (MPS) or 4 (PBS) biologically independent animals per group; mean ± s.d.

(Fig. 1b,c). Although LN volume in the MPS-vaccinated mice began to decrease ~20 days after immunization, it remained elevated out to 100 days (Supplementary Fig. 1d). NdLNs did not change in volume with either vaccine, and normalizing the dLN to ndLN volume within each mouse indicated a similar pattern of dynamic LN expansion and contraction (Fig. 1d and Supplementary Fig. 1e). The removal of either CpG or GM-CSF from the vaccine formulation diminished the magnitude of dLN expansion (Extended Data Fig. 2a). An MPS

vaccine with log-fold lower doses of ovalbumin (OVA) and CpG also induced long-term LN expansion (Extended Data Fig. 2b). While other published depot-based vaccine formulations including alum, MF59 emulsion and cryogel-based scaffolds also induced LN expansion, expansion was notably lower than with the MPS vaccine (Extended Data Fig. 2c). The MPS vaccine formulation was thus selected as a model of 'strong' vaccination resulting in persistent LN expansion for subsequent investigation.

## LN matrix and mechanics are altered with expansion

To assess tissue-scale alterations involved in durable LN expansion, LN mechanical properties and extracellular matrix (ECM) distribution were next characterized. Here, MPS-vaccinated mouse dLNs were collected 7 days after immunization, beyond the initial expansion phase (when MPS outpaced bolus vaccine LN expansion). At this time, LN collagen architecture was largely maintained, as expected (Fig. 2a)[6]. Hyaluronic acid (HA) localization was increased in the periphery/follicle, most visibly 7 days after immunization, although still notable up to 3 weeks later, demonstrating persistent alterations (Fig. 2a,b and Supplementary Fig. 2a). In contrast, the cellular F-actin signal was greater towards the centre of both control and MPS dLNs, with greater polarization between the centre and periphery in the MPS condition (Fig. 2c and Supplementary Fig. 2b–d). These changes suggest that LN expansion may be accompanied by changes in tissue mechanical properties, as both HA and F-actin are involved in cellular mechanotransduction and signalling pathways. Through nanoindentation of thick (~500 μm) LN slices (Fig. 2d), we found that LNs with enduring expansion had reduced stiffness ($G'$) and loss modulus ($G''$) compared with control LNs (Fig. 2e,f). Viscoelasticity, measured by $G''/G'$ ($\tan(\delta)$), was significantly increased in MPS dLNs compared with LNs from control mice, suggesting decreased matrix crosslinking (Fig. 2g).

Spatial variations in mechanics across LNs were next investigated using nanoindentation (Fig. 2h and Supplementary Fig. 3). Both control and MPS-vaccinated mouse LNs were softer and more viscoelastic in the centre than in the periphery, and this finding was confirmed through intentional sampling at the centre or periphery of naïve LNs (Supplementary Fig. 4a–e). The LN periphery (~12 kPa) was approximately twice as stiff as the centre (~6 kPa). Interestingly, after vaccination, LN $G'$ and $G''$ were only significantly altered at the periphery, while $\tan(\delta)$ increased only in the LN centre (Fig. 2i and Supplementary Fig. 5a–e). LN peripheral stiffness correlated negatively with LN mass, suggesting that the degree of tissue softening relates to the extent of LN enlargement induced by vaccination (Fig. 2j). LN cellular distribution and tissue density remained unaltered, despite expansion (Fig. 2k,l and Supplementary Fig. 6a–d). Taken together, these results suggest that LN tissue encompasses a range of mechanical properties, dependent on location within the node, and these parameters change as LNs expand.

## Vaccine formulation shapes cellular activity in LNs

Considering that tissue-level changes may impact or reflect cellular responses, changes in LN cellularity during expansion were next characterized, ranging from early-stage (day 4) to long-term (day 51) changes (Supplementary Figs. 7 and 8a). Cellular expansion was greater and more sustained in MPS-vaccinated mice than in the bolus-vaccinated mice or PBS-injected control; notably, the total cell counts within a LN correlated with its volume (Fig. 3a,b). As early as day 4, monocytes, neutrophils and macrophages were expanded in MPS dLNs, while conventional dendritic cells (DCs), plasmacytoid DCs and T cells peaked at day 7 before declining over time (Fig. 3c,d and Supplementary Fig. 8b–i). Monocytes in particular expanded ~80-fold in MPS dLNs compared with PBS controls 4 days after vaccination, relative to ~25× expansion in the bolus group, but this increase was maintained for several weeks in the MPS condition only (Supplementary Fig. 8n). B cells also significantly expanded by day 7 and remained elevated until day 17 (Fig. 3e). A variety of stromal cells expanded following MPS vaccination, typically peaking later (days 11–17) than the immune cells, except for natural killer (NK) cells, which also tended to expand later (days 7–11) (Fig. 3f and Supplementary Fig. 8j–m). By comparison, changes in the bolus vaccine group were more modest beyond 4 days, and PBS-treated control dLNs and ndLNs from all groups demonstrated minimal changes in cell populations. These results indicate that vaccine-induced LN expansion engages the temporal dynamics of a pathogen-induced immune response, with innate immune cells rapidly responding followed by lymphocytes at later times.

Because LN expansion is known to be mediated by myeloid interactions with LN stromal cells, we next performed single-cell RNA sequencing (scRNA-seq) on the LN myeloid compartment after vaccination (Fig. 3g and Supplementary Fig. 9a)[2]. LNs were examined at a late timepoint (days 20–21) to consider mediators of durable expansion. After removal of lymphocytes and stromal cells, we identified nine clusters from the remaining 20,858 cells analysed (Fig. 3h–j). Clusters were annotated as type-2 conventional DCs (cDC2s; c0, *Sirpa*, *H2-Ab1*), plasmacytoid DCs (c1, *Siglech*, *Bst2*), migratory DCs (c2, *Ccr7*, *Clu*), type-1 conventional DCs (c3, *Xcr1*, *Clec9a*), Langerhans cells (c4, *Cd207*), plasma cells (c5, *Ighg2b*, *Ighg1*), inflammatory monocytes (c6, *Csf1r*, *Ly6c2*), neutrophils (c7, *S100a8*, *S100a9*) and proliferating cDC2s (c8, *Top2a*, *Mki67*) (Fig. 3j and Supplementary Fig. 9b). Consistent with the flow cytometry analysis, scRNA-seq identified broad changes in LN cell populations after immunization, with notable differences based on vaccine strength (Fig. 3i,k). Compared with the PBS condition, both bolus and MPS vaccines increased DC2 proportions and decreased frequencies of migratory DCs and DC1s. Maintenance of LN expansion was associated with increased frequencies of inflammatory monocytes and plasma cells and decreased Langerhans cells (Fig. 3k).

## MPS vaccination alters LN myeloid cell gene expression

Given the importance of sustained antigen presentation in maintenance of LN immune responses[24,25], we hypothesized that vaccine antigen availability and APC populations may affect LN expansion. Compared with LNs of mice given the full MPS vaccine, LNs of mice given an MPS vaccine without antigen became prominently less enlarged and contracted sooner (Fig. 4a,b and Extended Data Fig. 3a). This indicates

**Fig. 3 | MPS vaccination alters cellularity of lymph nodes.** Mice were immunized with MPS or bolus vaccines containing GM-CSF, CpG and OVA protein, euthanized on days 4, 7, 11, 17 and 51 for LN collection and analysis through flow cytometry and compared to PBS-injected controls. **a**, Total LN cell counts over time. **b**, Linear regression of LN cell count on a given day versus volume (measured through HFUS). **c–f**, Numbers of dendritic cells (**c**), T cells (**d**), B cells (**e**) and follicular dendritic cells (FDCs; CD45⁻ CD31⁻ CD21/35⁺) (**f**) over time. For **a–f**, $n = 4$ (MPS dLN days 4, 11 and MPS ndLN day 7) or 5 (all other timepoints and groups) biologically independent animals per group per timepoint; mean ± s.d. For **a** and **c–e**, statistical analysis was performed using ANOVA with Tukey's post hoc test; differences present between one group and all other groups are shown. For **f**, statistical analysis was performed using Kruskal–Wallis test with Dunn's post hoc test; the statistical difference between the MPS and PBS dLN groups is shown. For **g–j**, mice were injected with MPS or bolus vaccines (GM-CSF, CpG, OVA) and dLNs were collected at a late timepoint (days 20–21). Naïve mice were included as controls. $n = 5$ biologically independent animals per group, barcoded and pooled for sequencing. **g**, Schematic of processing pipeline for single-cell sequencing. LNs were digested and FACS-sorted to enrich live, CD45⁺ CD3⁻ CD19⁻ cells for sequencing. **h**, UMAP of 20,858 cells across conditions coloured by cluster membership. **i**, UMAP as in **h**, here coloured by cell density. Red indicates high cell density, blue low density. **j**, Heat map of relative average expression of marker genes in each cluster from **h**. Colour bar indicates relative gene expression as $z$-score. a.u., arbitrary units. **k**, Frequency of individual cell clusters within each sample. Statistical analysis was performed using ANOVA with Tukey's post hoc test. pDCs, plasmacytoid DCs; Mig., migratory; Infl., inflammatory.

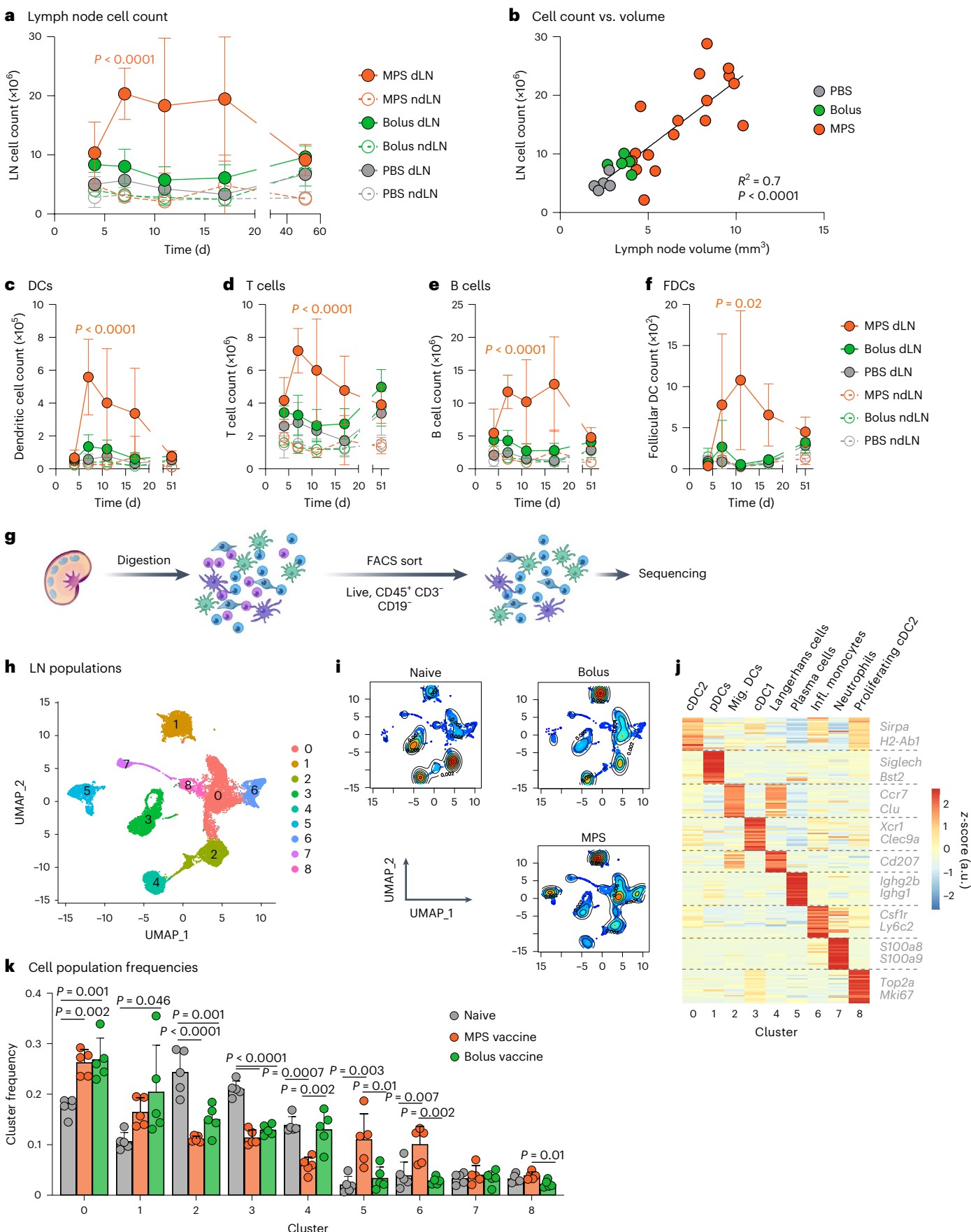

that long-term antigen presentation at the vaccine site is important for sustained LN expansion. Indeed, injecting the antigen separately as a bolus (that is, not delivered from the MPS scaffold) similarly reduced the degree and duration of expansion, indicating a critical role of sustained antigen presentation (Extended Data Fig. 3b).

To identify potential mediators of this differential response, we next focused the analysis of our scRNA-seq dataset on LN APC populations. Broadly, we identified varying numbers of differentially expressed genes within immune cell clusters between the MPS and bolus vaccine conditions (Fig. 4c). The most dramatic transcriptional changes were in the LN-resident cDC2 and cDC1 compartments, more so than in migratory DCs and Langerhans cells. The cDC2s showed the greatest number of differentially expressed protein coding genes between the two vaccine strengths (Fig. 4c,d), and consensus non-negative matrix factorization (cNMF) analysis[26] identified a cDC2-specific programme (CNMF_X14) enriched with MPS vaccination (Supplementary Fig. 10a–d). This programme included genes involved in inflammation (*Il1r2m*, *Cd86*), immune regulation (*Clec4a2*, *Sirpa*, *Lst1*), cell migration machinery (*Rasgef1b*, *Elmo1*) and smooth muscle contraction (*Ppp1r14a*) (Supplementary Fig. 10d). Furthermore, the gene encoding lymphotoxin beta (*Ltb*), another member of CNMF_X14, was strongly upregulated in cDC2s with MPS vaccination relative to the bolus condition (Fig. 4d,e and Supplementary Fig. 10d,e). The involvement of mechanosensing genes and *Ltb*, involved in lymphoid organogenesis, suggests that cDC2s may both respond and contribute to the changing LN microenvironment during expansion. Despite robust LN expansion and immune activation, *Cd274* (PD-L1) was not notably upregulated on myeloid cell subsets 20 days after immunization (Supplementary Fig. 11a,b). MPS immunization also increased the frequency of *CD19*− plasma cells and directed gene expression towards more mature immunoglobulin (*Ighg1* versus *Ighm*) expression (Supplementary Fig. 12a–d).

Inflammatory monocytes demonstrated significant transcriptional changes between the MPS and bolus vaccine groups (Fig. 4c), and the greatest expansion by both total number and relative proportion following MPS vaccination (Fig. 4f and Supplementary Fig. 8n). Therefore, we were interested in how MPS vaccination affected their gene expression profile. Monocytes, similar to DCs, can present antigen to T cells in LNs[27], and particular attention was paid to potential T cell interactions. Monocyte-specific clustering identified three subpopulations of inflammatory monocytes (Fig. 4g,h). Of these, c0 formed the predominant monocyte phenotype in LNs with sustained expansion (MPS condition) relative to naïve or bolus-vaccinated mice (Fig. 4h,i). Gene set enrichment analysis identified pathways associated with antigen processing and presentation, IFNγ response and inflammatory signalling that differentiated monocytes in the strong and weak vaccine LNs (Fig. 4j). These gene alterations position inflammatory monocytes as a potential stimulatory, APC type involved in sustained LN expansion.

### Inflammatory monocytes accompany robust LN expansion

To confirm the impact of vaccine strength on antigen-presenting, inflammatory monocytes, LNs of mice vaccinated with the MPS vaccine (with or without antigen) were collected and further compared to LNs of mice given bolus or PBS controls (Fig. 5a). Consistent with the scRNA-seq analysis, Ly6C$^{hi}$ inflammatory monocytes[27,28] comprised the majority (~60–70%) of LN monocytes in the MPS group over time, significantly higher than the PBS and bolus groups (~40–50%) by day 20 (Supplementary Fig. 13a,b). Inflammatory monocytes were also significantly expanded in terms of number and proportion in the LNs of MPS-vaccinated mice at day 20 compared with the PBS and bolus, and were visualized in LNs through CCR2 expression (Fig. 5b and Supplementary Fig. 13c,d)[29]. Inflammatory monocyte responses were abrogated at later timepoints when the MPS vaccine was delivered without antigen, equivalent to the PBS or bolus controls by day 20, suggesting a relationship between long-lived antigen presentation, LN expansion and monocyte responses (Fig. 5b). Consistent with scRNA-seq data and previous investigation on the MPS vaccine system, MPS immunization elicited robust and persistent germinal centre B cell responses, also dependent on the presence of antigen in the vaccine (Extended Data Fig. 4a–c).

Unlike LNs, spleens did not demonstrate superior cellular expansion after MPS vaccination compared with other vaccine groups (Extended Data Fig. 5a). Although total numbers of splenic immune cells including B cells and DCs were largely unaffected by vaccination, transient increases in T cells and macrophages were detected (Extended Data Fig. 5b–e). Notably, significantly higher numbers and proportions of inflammatory monocytes were found in MPS-vaccinated mouse spleens compared with all other conditions on day 20 (Extended Data Fig. 5f,g). These cells also remained elevated in circulation at the latest timepoint (Extended Data Fig. 5h).

Inflammatory monocytes in the MPS group displayed characteristics of antigen presentation; MHCII expression significantly increased in the MPS vaccine group compared with all others several weeks after vaccination (Fig. 5c,d). Numbers of monocyte-derived DCs (CD11c and MHCII-expressing Ly6C$^{hi}$ monocytes) were also significantly increased in the MPS-vaccinated dLN at this time compared with PBS-treated mice, or any condition in the spleen (Extended Data Fig. 6a). In the spleen, MHCII expression on inflammatory monocytes was unaltered with vaccination (Extended Data Fig. 6b). These results indicate that Ly6C$^{hi}$ monocytes induced by MPS vaccination may engage in antigen presentation, specifically within the LN compartment.

### Inflammatory monocyte depletion impacts vaccine response

To further discern the impact of inflammatory monocytes on lymph-node expansion and vaccine response, specific depleting reagents were next employed. MPS-vaccinated mice were treated with the CCR2-targeting MC-21 monocolonal antibody (mAb)[30–32] either early (days 1–5, LN expansion phase) or later (days 10–14, LN maintenance phase) after immunization (Fig. 5e). MC-21 mAb effectively depleted Ly6C$^{hi}$ monocytes in the blood, LN and MPS scaffold during the treatment course, although numbers in the blood rebounded within days (Fig. 5f and Supplementary Fig. 14a–c). Early depletion of Ly6C$^{hi}$ monocytes delayed the effector CD8$^+$ T cell response to vaccination, which peaked later, after monocytes had been restored, relative to the MPS vaccine group (Fig. 5g). Furthermore, only the MPS-vaccinated group treated early with MC-21 antibody had significantly elevated tetramer-specific CD8$^+$ T cells by day 20,

**Fig. 4 | MPS vaccination elicits long-term transcriptional changes in APCs.** **a,b**, Mice were immunized on day 0 with a full MPS vaccine (containing GM-CSF, CpG and OVA protein) or an MPS vaccine without antigen (GM-CSF and CpG only). LN volume was tracked using HFUS imaging. *n* = 5 biologically independent animals per group. **a**, Representative HFUS images of vaccine-draining LNs. Scale bar, 2 mm. **b**, Quantification of LN volume over time. Statistical analysis was performed using two-tailed *t*-tests. For **a** and **b**, *n* = 5 biologically independent animals per group. **c–j**, Mice were injected with MPS or bolus vaccines (GM-CSF, CpG, OVA) and dLNs were collected at a late timepoint (days 20 and 21). Naïve mice were included as controls. *n* = 5 biologically independent animals per group, barcoded and pooled for sequencing. **c**, Numbers of differentially expressed protein coding genes by cell type between the MPS and bolus conditions. *P* value calculated using DESeq2. **d**, Volcano plot displaying differentially expressed cDC2 genes between the MPS and bolus conditions. **e**, *Ltb* (lymphotoxin β) expression among DC subtypes in the different conditions. **f**, Proportion of inflammatory monocytes (cluster 6 from Fig. 3h) in LNs. **g**, UMAP of 1,468 inflammatory monocytes coloured by cluster membership. **h**, UMAP as in **g**, here coloured by cell density. Red indicates high cell density, blue low density. **i**, Proportion of cluster c0 among inflammatory monocytes. **j**, Pathway analysis for inflammatory monocyte cluster c0. For **c** and **d**, statistical analysis was performed with DESeq2. For **f** and **i**, statistical analysis was performed using ANOVA with Tukey's post hoc test.

after the monocyte rebound, compared with the PBS controls (Fig. 5h). Administration of MC-21 mAb in the later phase of the LN response (days 10–14) had no discernible impact on the T cell response. These results further suggest a role of inflammatory monocytes in effector

CD8+ T cell responses to MPS vaccination, potentially through direct antigen presentation or inflammatory stimulation.

LN expansion kinetics in the absence of inflammatory monocytes or other immune cell subsets were next assessed. MC-21 mAb

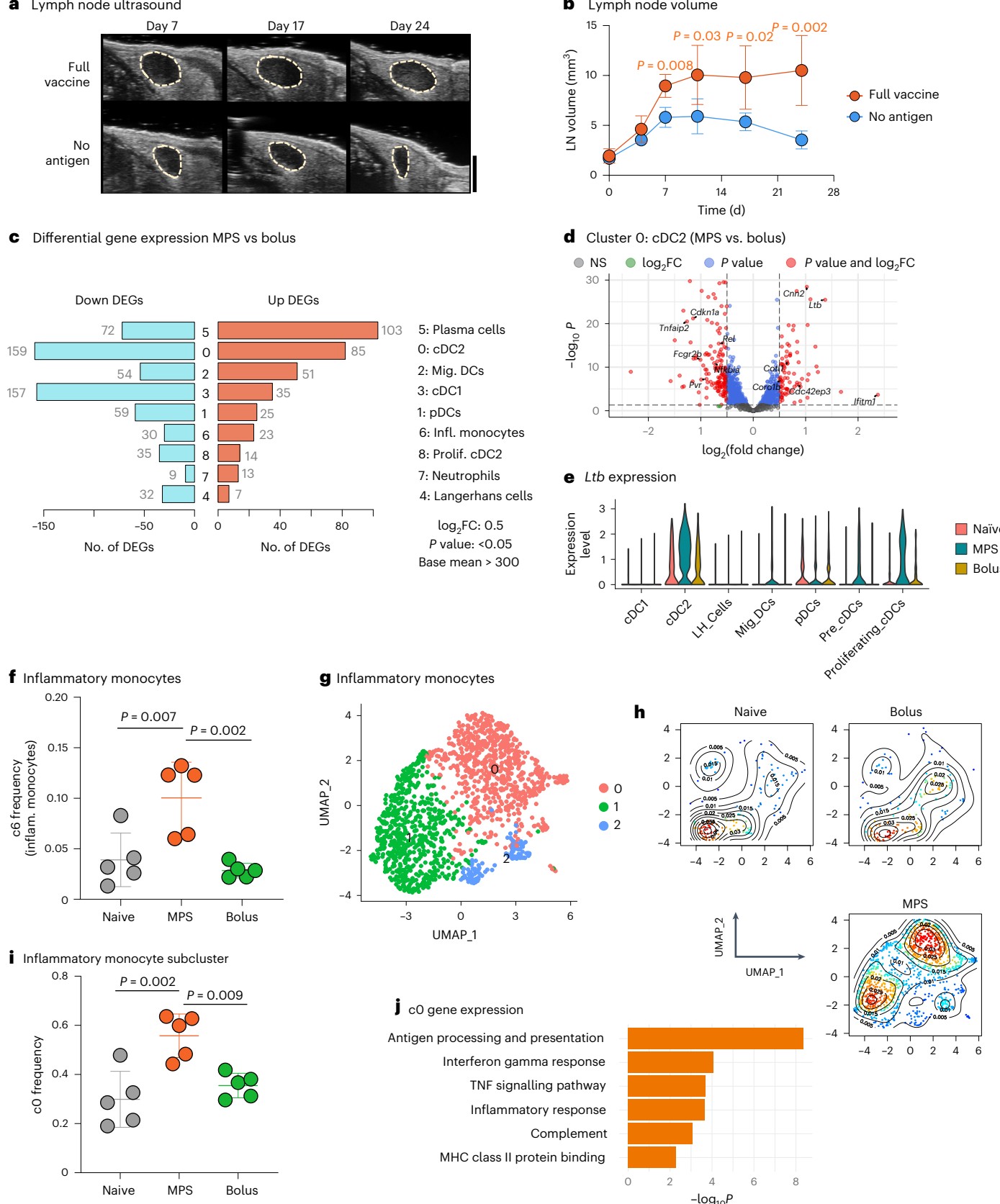

**a** Lymph node ultrasound

**b** Lymph node volume

**c** Differential gene expression MPS vs bolus

5: Plasma cells
0: cDC2
2: Mig. DCs
3: cDC1
1: pDCs
6: Infl. monocytes
8: Prolif. cDC2
7: Neutrophils
4: Langerhans cells

log₂FC: 0.5
P value: <0.05
Base mean > 300

**d** Cluster 0: cDC2 (MPS vs. bolus)

**e** *Ltb* expression

**f** Inflammatory monocytes

**g** Inflammatory monocytes

**h**

**i** Inflammatory monocyte subcluster

**j** c0 gene expression

Antigen processing and presentation
Interferon gamma response
TNF signalling pathway
Inflammatory response
Complement
MHC class II protein binding

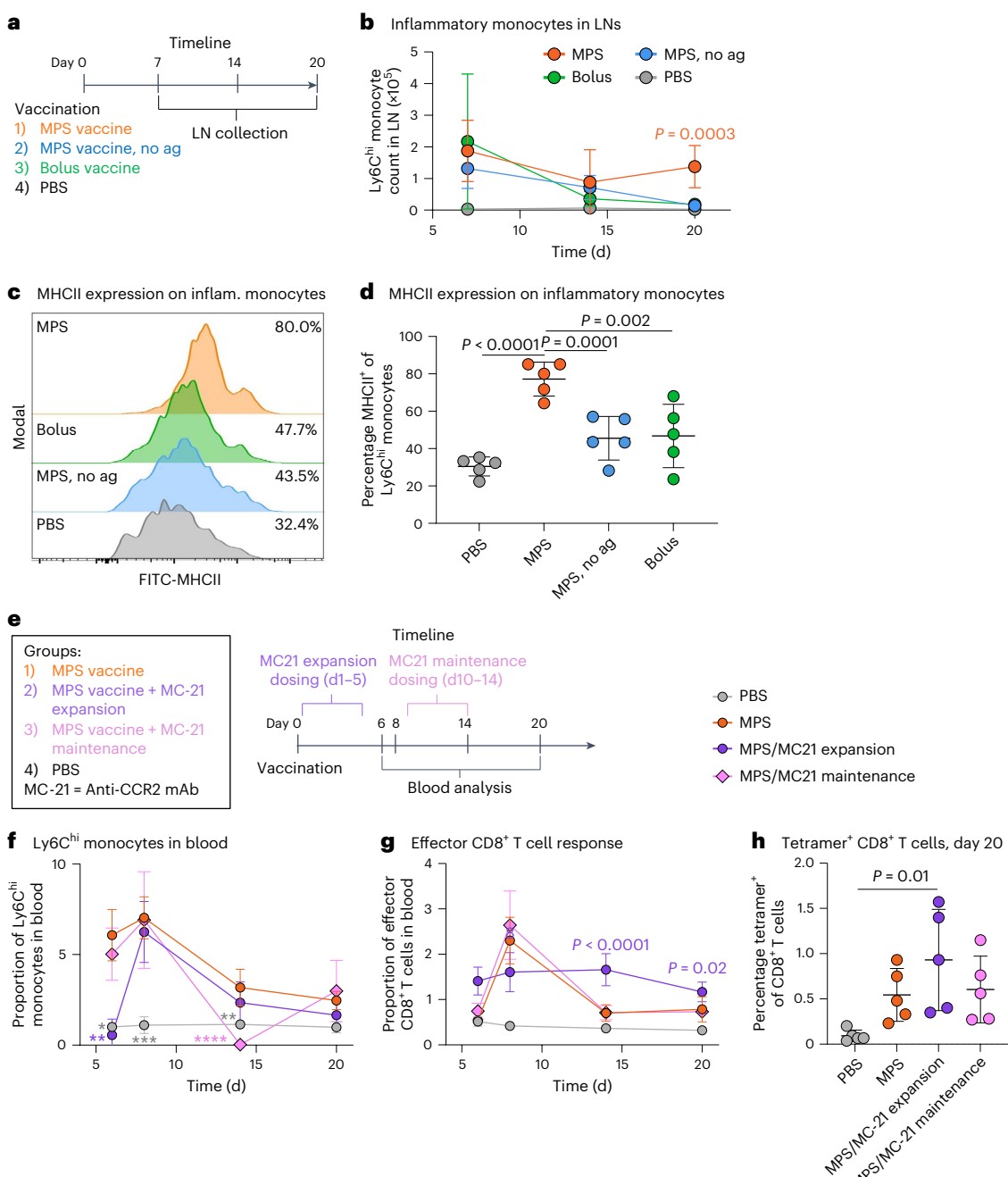

**Fig. 5 | Prolonged LN enlargement is associated with inflammatory monocyte expansion.** Mice were treated with MPS or bolus vaccines (containing GM-CSF, CpG, OVA), MPS vaccine without antigen (GM-CSF, CpG only) or PBS, and LNs were collected on days 7, 14 and 20 for cellular analysis. $n = 5$ biologically independent animals per group per timepoint. **a**, Experimental timeline and conditions. **b**, Inflammatory monocyte number in LNs over time. Statistical analysis was performed using ANOVA with Tukey's post hoc test. **c**, Representative flow cytometry histograms depicting MHCII expression on Ly6$^{hi}$ inflammatory monocytes. Median percentage MHCII expression in each group is listed on the right. **d**, MHCII expression on Ly6$^{hi}$ inflammatory monocytes in the LN at day 20. Statistical analysis was performed using ANOVA with Tukey's post hoc test. For **b** and **d**, means ± s.d. **e–h**, Mice were administered MPS vaccines (containing GM-CSF, CpG, OVA) or PBS. One group of MPS-vaccinated mice was treated with MC-21 CCR2-depleting mAb daily from days 1–5 ('MC-21 expansion') and one group was treated daily from days 10–14 ('MC-21 maintenance'). Peripheral blood was collected on days 6, 8, 14 and

20 for cellular analysis. $n = 5$ biologically independent animals per group. **e**, Experimental timeline and conditions. **f**, Inflammatory monocyte proportion in blood over time. Differences between groups are statistically significant (day 6 MPS versus MPS/MC-21 expansion, $P = 0.005$; day 6 MPS versus PBS, $P = 0.03$; day 8 MPS versus PBS, $P = 0.001$; day 14 MPS versus MPS/MC-21 maintenance, $P < 0.0001$; day 14 MPS versus PBS, $P = 0.002$). Significant differences between the MPS group and other groups are indicated on the figure (*$P < 0.05$, **$P < 0.01$, ***$P < 0.001$, ****$P < 0.0001$). **g**, Proportion of effector CD8$^+$ T cells (CD44$^+$ CD62L$^-$) in blood over time. **h**, Proportion of OVA-tetramer$^+$ of CD8$^+$ T cells in peripheral blood 20 days after vaccination. Statistical analysis was performed using ANOVA with Tukey's post hoc test. For **f–h**, means ± s.d. For **f** and **g**, statistical analysis was performed using Kruskal–Wallis test with Dunn's post hoc test (day 6 timepoint) or ANOVA with Tukey's post hoc test (days 8, 14, 20). For **b**, **f** and **h**, only differences between one group and all other groups are shown (*$P < 0.05$, **$P < 0.01$, ***$P < 0.001$, ****$P < 0.0001$).

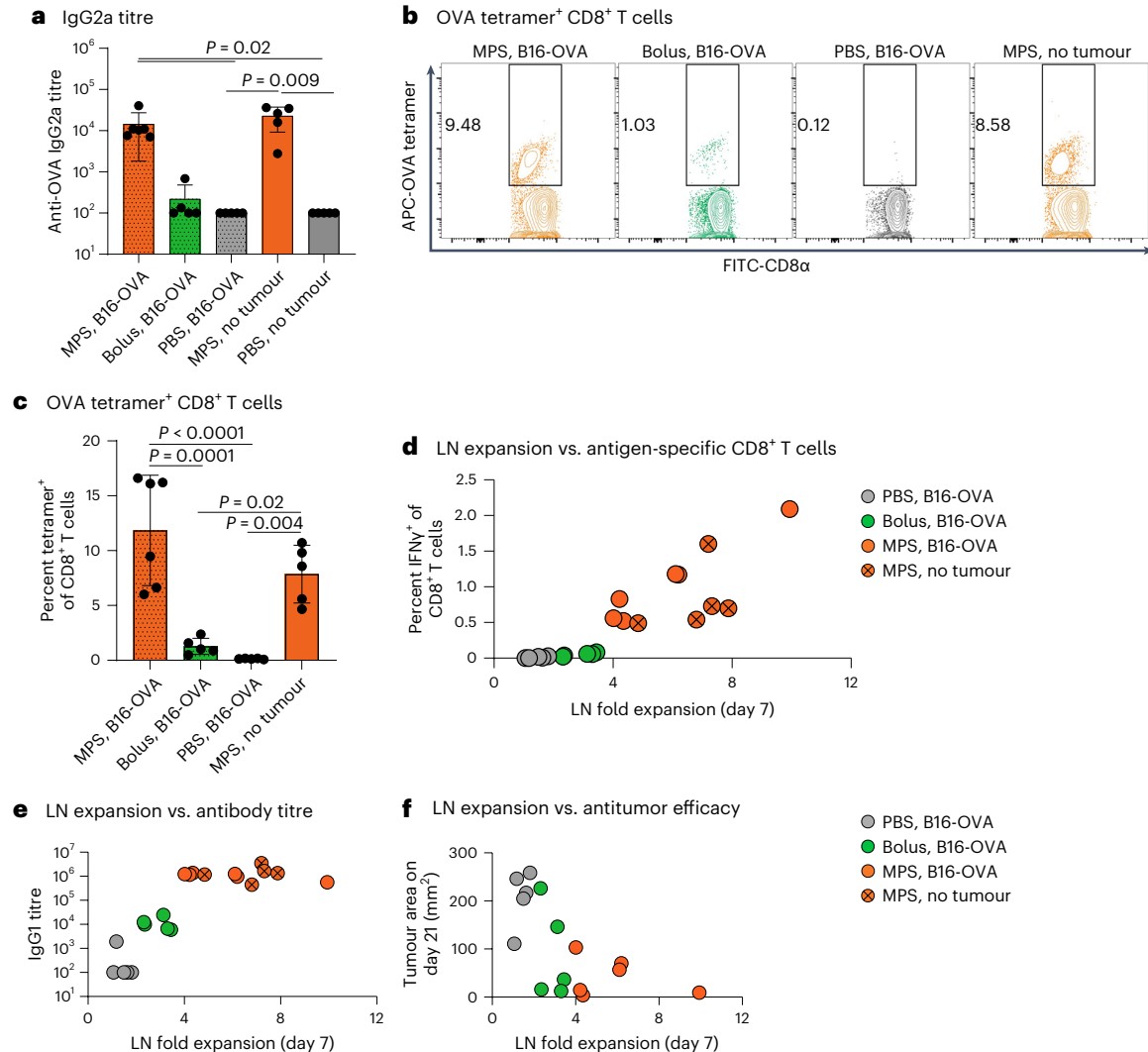

**Fig. 6 | Lymph-node expansion correlates with adaptive immunity and therapeutic outcomes of vaccination.** Mice were inoculated with B16-OVA melanoma tumours and 3 days later treated with MPS or bolus vaccines containing GM-CSF, CpG and OVA protein, and compared to PBS-injected controls. A fourth group of tumour-free mice was treated with MPS vaccines (called 'MPS, no tumour'). Inguinal dLNs were imaged using HFUS at multiple timepoints, and blood was collected 8 and 21 days after vaccination to assess T cell responses and serum antibody titres, respectively. $n = 6$ (MPS, B16-OVA) or 5 (all other groups) biologically independent animals per group. **a**, Serum anti-OVA IgG2a antibody titre 21 days after immunization. Statistical analysis was performed using Kruskal–Wallis test with Dunn's post hoc test. **b,c**, T cell analysis in the peripheral blood 8 days after immunization. **b**, Representative flow cytometry plots of OVA-tetramer binding to CD8+ T cells. **c**, Proportion OVA-tetramer+ of CD8+ T cells in blood. Statistical analysis was performed using ANOVA with Tukey's post hoc test. For **a** and **c**, means ± s.d. **d**, LN fold expansion 7 days after vaccination versus blood IFNγ+ CD8+ T cell response to SIINFEKL restimulation 8 days after vaccination. **e**, LN fold expansion 7 days after vaccination versus anti-OVA IgG1 titres 21 days after vaccination. **f**, LN fold expansion 7 days after vaccination versus tumour area at the latest timepoint with all mice surviving (day 21).

and/or clodronate liposomes were used to deplete Ly6C^hi monocytes and macrophages, respectively (Extended Data Fig. 7a). Lymphocyte (anti-CD4, CD8 and B220) and neutrophil (anti-Ly6G) antibodies were also tested. No differences were observed in the magnitude or kinetics of LN expansion with depletion of any immune cell subset alone (Extended Data Fig. 7b–g). However, depleting both inflammatory monocytes and macrophages together restrained the maintenance of LN expansion (Extended Data Fig. 7h). Taken together, these data indicate a stimulatory and antigen-presenting role of inflammatory monocytes, and that these cells in association with macrophages may be required for sustained LN expansion.

## LN expansion determines vaccine outcomes

Finally, we considered whether durable LN expansion could indicate functional outcomes of vaccination. In a therapeutic model of mouse melanoma, LN expansion after vaccination against a

tumour-expressed antigen was not affected by tumour presence (Supplementary Fig. 15a–c). The MPS vaccine generated stronger adaptive immune responses than the bolus vaccine, leading to therapeutic benefit (Fig. 6a–c and Supplementary Figs. 15d–g and 16a–c). Importantly, LN expansion associated positively with antibody titres, CD8+ T cell responses and antitumour efficacy of cancer vaccine formulations (Fig. 6d–f and Supplementary Fig. 17a–c). The degree of LN expansion also correlated strongly with effector CD8+ T cell proportions following vaccination across experiments (Supplementary Fig. 17d). In a tumour-free setting, MPS vaccination also enhanced long-term antibody production (day 90) and splenic CD8+ T cell (day 103) responses as compared with the bolus vaccine, and responses associated with earlier degree of LN expansion (Extended Data Fig. 8a–f). Sustained inflammatory cytokine expression in splenic CD8+ T cells suggested a long-lived adaptive immune response in multiple lymphoid organs.

We next assessed potential indicators of toxicity or T cell dysfunction that could result from sustained LN expansion. In MPS-vaccinated mice, serum HMGB-1 levels, indicative of inflammatory cytokine responses and/or cellular death[33,34], were comparable to PBS controls (Extended Data Fig. 8g). Long-term (day 103) PD-1 expression on splenic T cells was also not different between the MPS vaccine group and PBS controls (Extended Data Fig. 8h,i). Mice monitored for 485 days after MPS vaccination did not display changes in weight, LN or spleen cell counts, or proportions of immune cell subsets in blood or secondary lymphoid organs, although elevated OVA-specific CD8[+] T cells remained detectable in all immune compartments investigated (Extended Data Fig. 9a–i). Furthermore, MPS vaccine-generated T cells retained functional, antigen-specific antitumour response when challenged 50 days after immunization (Extended Data Fig. 10a–c). Altogether, these results suggest that enduring LN expansion is associated with immune memory and antitumour efficacy, without indications of T cell dysfunction.

To explore whether LN expansion could directly improve vaccine efficacy, the MPS vaccine without antigen (Fig. 4a,b) was employed to 'jump-start' LN expansion before administration of a full, antigen-containing bolus vaccine (Fig. 7a). LNs of mice given the antigen-free MPS jump-start expanded over the first week and continued to increase in size after administration of the bolus vaccine, becoming significantly enlarged compared with all other groups (Fig. 7b). The jump-start plus bolus vaccine broadly improved vaccine responses compared with the traditional bolus vaccine. The proportion of OVA-tetramer[+] CD8[+] T cells in blood was significantly increased in this condition (Fig. 7c,d). Blood CD8[+] T cells restimulated ex vivo with SIINFEKL peptide had superior cytokine production (IFNγ and TNFα) with the jump-start (Supplementary Fig. 18a–c), and the jump-start also increased the proportion of effector CD8[+] T cells and decreased the blood CD4/CD8 T cell ratio relative to mice given the bolus vaccine alone (Supplementary Fig. 18d,e). The combination treatment improved short- and long-term IgG2a antibody titres, with 10/10 (versus 6/10 with the bolus only) detectable IgG2a titres after 100 days (Fig. 7e and Supplementary Fig. 18f). In these experiments, the jump-start was dosed 7 days before the bolus vaccine to match the peak of LN enlargement (Supplementary Fig. 19a). Spacing the jump-start closer to bolus vaccination (4 days) tended to increase antigen-specific cytokine expression (IFNγ and TNFα) and OVA-tetramer binding; however, increasing the dose separation (11 days) increased granzyme B and reduced PD-1 expression, suggesting that the timing of jump-start and bolus vaccination can alter functional T cell outcomes, and the day 7 timepoint balances both sets of outcomes (Supplementary Fig. 19b–h). All additional experiments were conducted with a 7-day spacing. In treating B16-OVA tumour-bearing mice, the jump-start strategy (Supplementary Fig. 20a,b) elicited prolonged tumour regressions compared with the bolus vaccine, which induced only transient tumour regressions, with all mice in this condition eventually succumbing to tumour burden within 50 days. In the jump-start plus bolus group, 25% of mice survived at 200 days, a significant improvement over all other groups (Fig. 7f,g). In summary, jump-starting LN expansion before vaccine administration improved T cell responses and antitumour efficacy in a model antigen tumour model.

### A prime-boost aids LN expansion and improves immune response

Finally, we assessed the impact of a booster vaccine format on LN expansion kinetics and adaptive immune responses. Following the MPS prime vaccine, dLN volume increased over the following 1–2 weeks and declined by day 42 (Supplementary Fig. 21a,b). On day 43, a booster MPS vaccine was delivered, and this led to more immediate LN expansion, reaching peak volumes within 4 days, compared with day 7 with the initial vaccine. Seven days after the booster vaccine, peripheral blood was collected and compared to mice that had received only prime vaccination at the same timepoint as the boost in the prime-boost group.

No differences in the IFNγ[+] proportion of CD8[+] T cells after OVA peptide restimulation were detected; however, the IFNγ[+] proportion of CD4[+] T cells was significantly increased relative to both naïve control mice and mice that had received only prime vaccination (Supplementary Fig. 21c,d). The proportion of effector-phenotype (CD44[+] CD62L[−]) CD8[+] T cells was elevated with the MPS prime and further increased after the booster (Supplementary Fig. 21e). Both IgG1 and IgG2a titres against OVA were increased after the booster dose compared with either the same mice on day 21 (pre-boost) or the prime-only mice at the same timepoint (Supplementary Fig. 21f,g). These results indicate that a booster vaccine may elicit more rapid LN expansion along with a stronger adaptive immune response.

## Discussion

Our results indicate that, whereas adjuvant and biomaterials engage early LN expansion, antigen-specific responses are key to durable LN enlargement. The MPS vaccine used here as a model of robust, persistent vaccination elicited a ~7-fold increase in dLN volume that was maintained for weeks. This is a distinctive outcome compared with reported vaccine formulations (~4–5-fold increase in mass or size, and contracting from days 7–14)[2,35–40]. In a recent study, popliteal LNs expanded ~10-fold in response to vaccination with complete Freund's adjuvant, although this response was not explored beyond 14 days[10]. Although clinically successful vaccines such as those against SARS-CoV-2 have induced lymphadenopathy in a subset of patients, presenting potential discomfort, these responses largely resolved naturally and could point to a productive adaptive immune response[16,17,41]. In our experiments, the MPS vaccine promoted significantly greater LN expansion than commonly used vaccine delivery methods including alum and MF59, an emulsion-based vaccine. The MPS vaccine also drove stronger LN enlargement than cryogel-based scaffold vaccines fabricated from alginate, an interesting result given the identical vaccine components in these two vaccines, suggesting that the selection of biomaterial itself can impact immune outcomes. This result is not unexpected given that silica can activate inflammasome activity, while alginate is widely used for its biocompatibility and lack of immunogenicity[42,43]. The prolonged LN expansion observed here likely derives from persistence of the MPS vaccine and its sustained release of antigen and adjuvant[18]. Here, MPS vaccines without sustained antigen presentation, either antigen-free or with antigen rapidly released as a bolus, did not maintain LN expansion relative to when antigen was released in a sustained manner from MPS. Removing either CpG or GM-CSF from the vaccine also diminished LN expansion. Sustained antigen release can prolong germinal centre activity in the LNs and may maximize LN expansion[24,25,35,40], and it was previously shown that early explant of the MPS scaffold site impaired antibody titres[22].

The MPS vaccine elicited strong and persistent humoral and cellular immune responses, consistent with previous findings[18,22]. By day 20, the MPS formulation, in contrast to the bolus vaccine, led to mature Ig expression by LN plasma cells. In addition, GL7[+] B cells were observed to expand in MPS-vaccinated mouse LNs, dependent on the presence of antigen. Our finding of heightened antibody titres for months after vaccination additionally supports these results. Antigen-specific CD8[+] T cells were also detectable in blood, spleens and LNs at 8, 20, 103 and up to 485 days after vaccination. Altogether, these results position the MPS vaccine as a potent inducer of both adaptive and humoral arms of immunity, also associated with strong LN expansion.

Although robust and maintained immune engagement could have the potential to induce T cell exhaustion or anergy, as previously suggested[44], the data in this paper do not suggest that this occurs with the MPS vaccine. Here, although PD-1 was upregulated on CD8[+] T cells within 8 days after vaccination, these levels returned to baseline within weeks. In addition, 20 days after vaccination, no evidence of PD-L1 gene upregulation was detected on myeloid cells. Splenic T cells stimulated ex vivo 103 days after immunization remained functional, capable of antigen-specific cytokine secretion, and evidence of tetramer binding

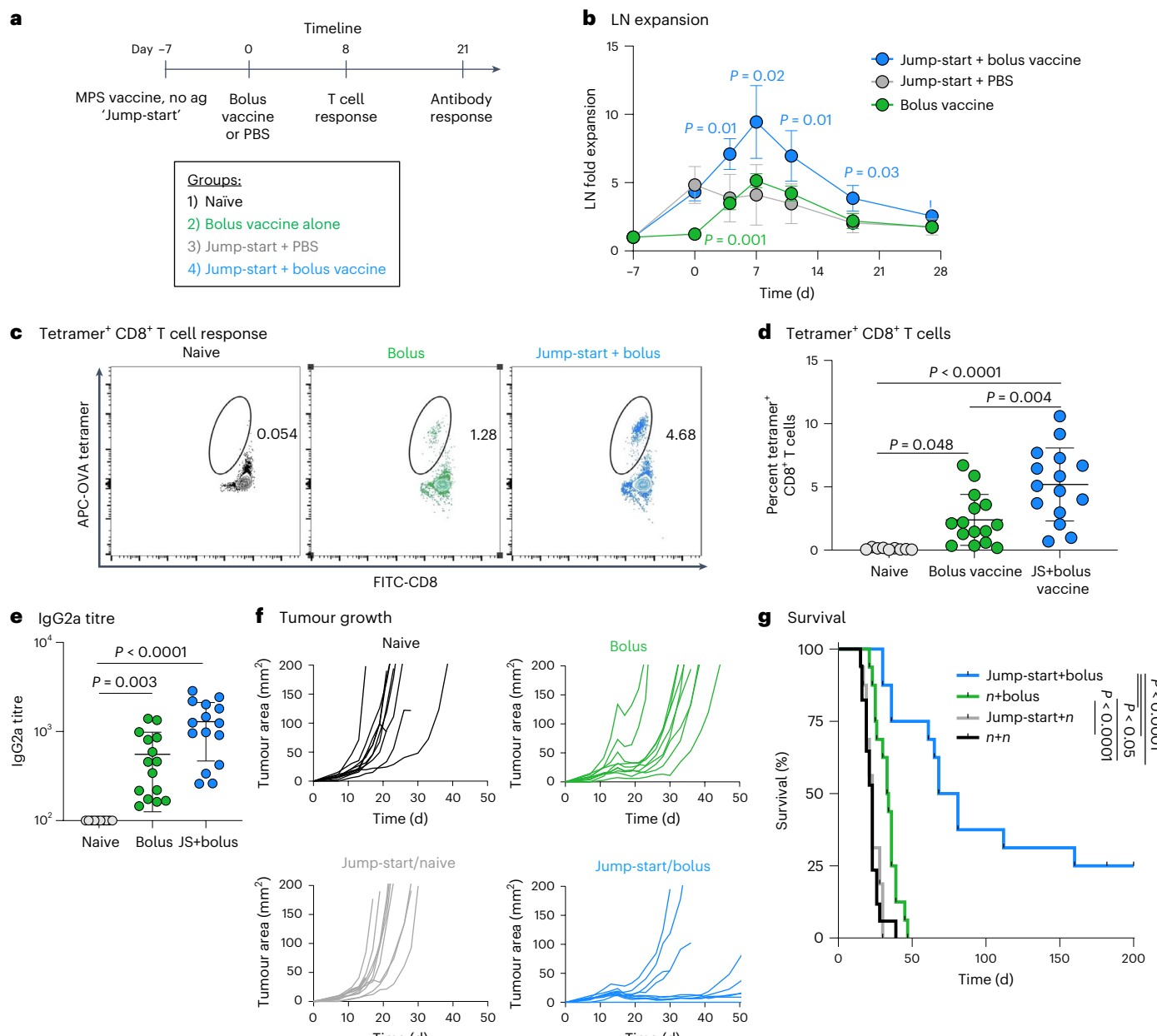

**Fig. 7 | Augmenting lymph-node expansion improves the magnitude and efficacy of the immune response. a**, Experimental timeline for **b**−**e**; mice were injected with PBS or a bolus vaccine on day 0, or injected with an MPS no-antigen 'jump-start' on day −7 followed by PBS or a bolus vaccine (GM-CSF, CpG and OVA protein) on day 0. Mice were bled after 8 and 21 days for T cell analysis and serum antibody titres, respectively. **b**, LN expansion measured by HFUS imaging. Values are normalized to the baseline volume for each individual LN. $n = 5$ biologically independent animals per group; only differences between one group and all other groups are shown. **c**, Representative flow cytometry plots depicting CD8+ T cell OVA-tetramer binding in cells derived from blood on day 8. **d**, OVA-tetramer+ proportion of CD8+ T cells. **e**, Anti-OVA IgG2a titre on day 21. Statistical analysis was performed using Kruskal–Wallis test with Dunn's post hoc test. **f**,**g**, Mice bearing B16-OVA tumours (inoculated on day −8) were treated starting at day −7 as per studies in **a**−**e** with an MPS 'jump-start' (MPS material, GM-CSF and

CpG without antigen) or left untreated, and then injected with a bolus vaccine (GM-CSF, CpG and OVA) or left untreated at day 0. Tumour growth and survival were tracked. **f**, Tumour growth curves. $n = 10$ biologically independent animals per group. **g**, Kaplan–Meier curves depicting survival. $n$ indicates naive/non-vaccinated ($n$+bolus, naive + bolus vaccine; jump-start+$n$, jump-start + naive; $n$+$n$, naive + non-vaccinated). Statistical analysis was performed using log-rank (Mantel–Cox) test, correcting for multiple comparisons. $n = 17$ ($n/n$) or 16 (all other groups) biologically independent animals per group; results are combined from two independent experiments, the second performed in a blinded manner. For **b** and **d**, statistical analysis was performed using ANOVA with Tukey's post hoc test. For **c**−**e**, $n = 9$ (PBS) or 15 (all other groups) biologically independent animals per group; results are combined from two independent experiments. For **b**, **d** and **e**, means ± s.d.

was found up to 485 days. The functional capacity of the immune system following vaccination was further supported by the ability of vaccine-induced T cells to restrain tumour challenge after 50 days. These results align with efforts to extend vaccine delivery and LN targeting to improve adaptive immune responses[45–47] and indicate that vaccine-induced T cells retain functional memory potential for

months to years. Notably, spleens did not expand to the degree of LNs, and broad immune cell populations in the spleen were largely unchanged after MPS vaccination. Furthermore, MHCII upregulation was only observed on inflammatory monocytes in the LN, suggesting that the MPS vaccine may exert specific influence on the local draining LN versus systemic compartments.

Our characterization of the spatial mechanical properties of LNs after vaccination reveals that vaccine-draining LNs show location-specific alterations in ECM and in tissue mechanical properties with immunization. HA provides both tissue scaffolding and cell signalling cues, and is regularly recycled through LNs. Its increased quantity in the LN periphery after vaccination may suggest altered recycling dynamics and could suggest a potential role in inflammatory signalling during the vaccine response. F-actin plays a key role in cytoskeletal maintenance and cell motility, and the observed pattern of F-actin localization indicates that the tissue-level mechanical changes in LNs may translate down to individual cell behaviour, such as immune cell migration in the expanded node. Our findings are in general agreement with a previous study suggesting LN softening during podoplanin blockade[8]. The consistent cell density and spatial arrangement found here in LNs during enlargement suggests that LNs expand coordinately with the degree of internal cellular expansion, and their expansion is not due to changes in cell density. The relaxation of individual FRCs, increasing gap size during LN expansion or altered ECM distribution could instead be responsible for this phenomenon[2,6,10,48]. Previously, parallel plate compression of explanted LNs recorded a modest increase in Young's modulus following vaccination, potentially due to this technique capturing internal proliferation and pressure, or to fibrosis of the LN capsule[10].

Inflammatory monocytes are strongly implicated as key mediators of the persistence of the MPS vaccine response. During LN expansion following MPS vaccination, immune and stromal cell numbers increased in dLNs, while weaker vaccination (that is, bolus) led to minimal cellular expansion. The patterns of LN cellular responses observed here are consistent with previous immunological characterizations[2,7,9]. For example, immune cell numbers peaked earlier than stromal cells (days 4–7 versus day 11). Immune cells themselves responded with the natural progression of infection response, with innate/myeloid cells responding most rapidly followed by adaptive immune cells. Ly6C[hi] inflammatory monocytes rapidly increased in LNs following 'strong' vaccination and remained elevated throughout LN expansion. These monocytes upregulated antigen-presentation-related genes in the MPS relative to the bolus vaccine group, supporting a role in antigen presentation[27]. Monocytes can also affect the quality of T cell responses (for example, effector phenotype) through cytokine secretion and T cell co-localization, and mobilize in response to type I adjuvants including CpG[49]. Notably, early depletion of inflammatory monocytes delayed effector CD8[+] T cell responses and antigen specificity, supporting these results and suggesting a functional role of these cells in the vaccine response. Although depletion of monocytes alone did not affect the kinetics or magnitude of LN expansion, co-depletion with macrophages prevented long-term maintenance of LN enlargement, additionally implicating these APC subsets in sustained vaccine responses.

The upregulation of genes associated with LN development and maintenance (*Ltb*) in cDC2s with MPS vaccination also suggests a role of cDC2s in prolonging LN expansion. *Ltb* encodes lymphotoxin β, with known involvement in lymphogenesis and expansion[9]. Although DCs were previously associated with driving LN expansion, the exact subtype was not defined, relying on a mouse model with broad CD11c[+] cell modification[2]. Here, cDC2s showed extensive transcriptional differences between 'strong' and 'weak' vaccine conditions, and significantly upregulated *Ltb*. Because the most dramatic changes in gene expression occurred with LN-resident APC populations (cDC1s, cDC1s, plasma cells), these results suggest that the MPS vaccine strongly and specifically affects the LN environment to produce strong protective immunity. To our knowledge, our single-cell analyses of LN myeloid populations at a late time (day 20) of vaccine-mediated LN expansion are the latest timepoint reported in the literature, providing insight into features regulating durable LN expansion[50].

Importantly, LN expansion was associated with therapeutic vaccine outcomes, and expansion of LNs before bolus vaccination significantly improved efficacy. The timing of LN jump-start before bolus vaccination had implications on vaccine-responding T cell functionality, as spacing the jump-start and bolus vaccine closer together increased T cell expression of inflammatory cytokines IFNγ and TNFα, but at the cost of decreased granzyme B and increased PD-1 expression. Previously, in a model of E7 peptide vaccination in C3 HPV-E7 tumour-bearing mice, vaccine response was reported to correlate with LN expansion (assessed by magnetic resonance imaging (MRI))[35]. Accordingly, minimally invasive LN imaging technologies such as HFUS or MRI may indicate the strength of vaccine immune responses in mouse models. In humans, certain vaccines such as those against SARS-CoV-2 can lead to lymphadenopathy in a subset of patients[16,51]. It remains to be shown whether this expansion indicates initiation of a successful immune response. Our studies made use of OVA, a commonly used model antigen, and exploration of more representative antigens (such as melanoma neoantigens, viral targets or peptides) can supplement this work. The jump-start vaccine concept developed here comprises generalized and off-the-shelf components, and could be applied to broadly improve subsequent antigen-specific therapy.

## Methods

### Fabrication and characterization of the MPS vaccines

MPS vaccines were prepared as previously described[18,22]. Briefly, 4 g of Pluronic P123 surfactant (average $M_n$ ~ 5,800, Sigma) was dissolved in 150 g of 1.6 M HCl, mixed with 8.6 g tetraethyl orthosilicate (TEOS, Sigma) at 40 °C for 20 h and stirred at 100 °C for another 24 h to fabricate MPS. Pluronic P123 was cleared in 1% ethanol in HCl at 80 °C for 18 h, and MPS was filtered and dried at 65 °C. To prepare MPS vaccines, MPS was suspended in pH 7.4 sterile DPBS, and 2 mg each was incubated with OVA (200 µg per vaccine, unless otherwise indicated in experiments using MPS vaccines without antigen) or CpG (100 µg per vaccine) at room temperature for ~7 h before freezing and overnight lyophilisation. The next day, a separate 1 mg aliquot of MPS was suspended in DPBS and loaded with GM-CSF (1 µg per vaccine) for 1 h at 37 °C while shaking. The three MPS populations (loaded with antigen, CpG and GM-CSF) were combined in 150 µl of sterile deionized $H_2O$. Each MPS vaccine mixture was injected through an 18G needle into the left flank of the mouse. GM-CSF was purchased from PeproTech. CpG-ODN 1826 5′-TCCATGACGTTCCTGACGTT-3′ was synthesized by Integrated DNA Technologies. OVA was purchased from InvivoGen (InvivoGen vac-pova). Vaccine components (MPS rods, CpG, GM-CSF and OVA) were endotoxin tested (<0.01 EU per vaccine). Control MPS vaccines lacking either antigen or adjuvant were prepared using the same total amount of MPS material (5 mg) with either antigen, CpG or GM-CSF excluded from the vaccine combination. For bolus antigen injection experiments, the MPS vaccine containing CpG and GM-CSF alone was followed by injection of OVA antigen alone adjacent to the vaccine site. A low-dose MPS vaccine was prepared using 20 µg OVA, 20 µg CpG and 1 µg GM-CSF delivered in the same 5 mg MPS material per vaccine. MPS particle morphology was assessed using brightfield microscopy and scanning electron microscopy, and surface area, pore size and pore volume were measured using $N_2$ adsorption/desorption isotherms[18].

### Bolus vaccines

OVA (200 µg per vaccine), CpG (100 µg per vaccine) and GM-CSF (1 µg per vaccine) were mixed in pH 7.4 sterile DPBS to obtain a total volume of 150 µl per dose.

### Ultrasound imaging

Mice were anaesthetized before imaging. Hair on the left and right abdomen was shaved and fully removed using Nair hair removal lotion. Mice were placed on a heated stage (37 °C) and their limbs were gently

secured using tape (3 M Transpore). Signa creme electrode cream (Parker Laboratories) was applied on the paws to detect respiration. The stage was rotated about the long axis of the animal (~30°) to expose the inguinal region, and Aquasonic ultrasound transmission gel (Parker Laboratories) was applied. Inguinal LNs were imaged using a Vevo 3100 scanner and 50 MHz transducer on a Vevo 3100 preclinical imaging system (Visualsonics). Respiration gating was enabled to avoid respiratory artefacts, and LNs were scanned using a step size of 0.1 mm. LN volumes were quantified using Vevo LAB software (Visualsonics). This procedure was also applied to image the vaccine site. To check reliability of data, the same mouse LNs were imaged by two researchers on the same day and the same researcher on different days. This resulted in consistent volume measurements, indicating that this a replicable technique. To account for the effect of circadian rhythms in lymphocyte trafficking[52,53], LNs were imaged at the same time of day (±1 h) for the duration of each experiment.

## Alternative vaccine formulations

Alum and emulsion-based vaccines were formulated following manufacturer protocols. Briefly, alum-based vaccines were prepared by mixing 2% (20 mg ml$^{-1}$) Alhydrogel adjuvant (Invivogen vac-alu-50) in a 1:1 volume ratio with 200 µg OVA protein antigen to reach a total volume of 100 µl per vaccine dose. The MF59 oil-in-water emulsion vaccine was prepared by mixing AddaVax (Invivogen vac-adx-10) in a 1:1 volume ratio with 200 µg OVA protein antigen to reach a total volume of 100 µl per vaccine dose. Both vaccines were injected subcutaneously (s.c.) through an insulin syringe.

## Cellular analysis of lymph nodes

At days 4, 7, 11, 17 and 51, or 7, 14 and 20 post-vaccination, mice were euthanized and vaccine-draining and non-draining LNs were explanted and digested in RPMI-1640 (Corning) containing 0.8 mg ml$^{-1}$ Dispase II, 0.2 mg ml$^{-1}$ Collagenase P and 0.1 mg ml$^{-1}$ DNase I (all Roche, procured from Sigma) until no visible LN pieces remained, following established protocols[54,55]. Cell suspensions were strained through a 70 µm filter, counted using a Countess II FL (ThermoFisher) and stained using standard flow cytometry protocols with antibodies in Supplementary Table 1. Viability was assessed using a Zombie Yellow stain (BioLegend) and samples were run on an Aurora Spectral Analyzer (Cytek) or LSRII flow cytometer (BD Biosciences) and analysed using FlowJo (v.10.4) and Graphpad Prism (v.9) software. Flow cytometry gating strategies for immune cell populations are indicated in Supplementary Fig. 7.

## Peripheral blood analysis

Blood was collected from mice retro-orbitally using heparinized capillary tubes (Fisherbrand) and stored in heparinized collection tubes (BD Biosciences) on ice. Red blood cells were lysed using ACK lysing buffer (Quality Biological). To detect OVA-specific CD8$^+$ T cells, samples were incubated with 7 µg ml$^{-1}$ allophycocyanin-conjugated H-2Kb-SIINFEKL tetramer (National Institutes of Health tetramer core facility) at 37 °C. For cytokine analysis, cells were instead incubated with 2 µg ml$^{-1}$ OVA257–264 SIINFEKL OVA CD8 peptide for 1.5 h at 37 °C. GolgiStop (BD Biosciences) was then added and samples incubated for another 4 h at 37 °C. Samples were stained with anti-mouse CD3, CD8 and CD4 antibodies and treated with a Cytofix/Cytoperm kit (BD Biosciences) following manufacturer protocol. Permeabilized samples were stained for IFNγ, IL-2 and TNFα and analysed by standard flow cytometry protocols. For general cellular analysis, blood samples were stained with antibodies listed in Supplementary Table 1.

## Antibody titres

Blood was collected from mice and allowed to coagulate for 30 min at room temperature. Tubes were centrifuged at 2,200 × g for 10 min and serum collected and transferred into low-binding tubes for storage at −80 °C. Anti-OVA antibody titres were quantified using ELISA.

Briefly, high-binding plates (Costar 2592, Cole Palmer) were coated at 4 °C overnight with 10 µg ml$^{-1}$ OVA in DPBS with gentle rocking. Serum samples were diluted (1:10$^2$–1:10$^8$) in 1x ELISA diluent in pH 7.4 DPBS and added to the washed plate for 2 h at room temperature. Plates were washed and biotinylated anti-mouse IgG1 and IgG2a antibodies (BD Biosciences, each diluted 1:100 in 1x ELISA diluent) were added for 2 h at room temperature. Plates were washed and streptavidin-HRP (BD Biosciences, diluted 1:1,000 in 1x ELISA diluent) was added for 15 min at room temperature. Plates were again washed and TMB substrate solution (BioLegend) was added for 15 min at room temperature before addition of stop solution (2 M hydrochloric acid). Absorbance was read at λ = 450 nm, subtracting background λ = 540 nm. Anti-OVA titre was defined as the lowest serum dilution with optical density (OD) = 0.2.

## Immunohistochemistry of LNs

Mice were euthanized and inguinal LNs were collected into 0.25 ml PBS on ice and cleaned of surrounding tissue. LNs were fixed in 1% paraformaldehyde in PBS for 1 h at 4 °C, rinsed 3× with PBS and transferred into 30% sucrose in PBS overnight at 4 °C. LNs were subsequently transferred to a 1:1 solution of 30% sucrose in PBS and Tissue-Tek O.C.T. compound (OCT, VWR) for 45 min at room temperature. LNs were blotted on a Kimwipe, added to OCT-containing cryomolds on dry ice and stored at −20 °C. Sections (15 µm) were prepared using a cryostat (Leica). For staining, samples were placed in PBS for 2 min to dissolve OCT and blocked in 3% normal goat serum and 1% BSA in PBS at room temperature. When biotin-containing stains were used, samples were also blocked in 0.01% avidin and 0.001% biotin in subsequent steps, both for 15 min at room temperature. Primary antibody staining solution (Supplementary Table 2) was added overnight at 4 °C. The next day, secondary and nuclear stains including streptavidin-AF594 conjugate (Invitrogen, 1:100), AlexaFluor 647 goat anti-rabbit IgG (Invitrogen, 1:100) and Hoechst 33342 (Thermo Scientific, 1:500) were added for 2 h at room temperature. Samples were mounted using ProLong Gold antifade reagent (Invitrogen) and imaged using a Zeiss 710 confocal system with Zen Black (v.2.1, Zeiss) software. Images were analysed using ImageJ (v.1.53 m) software and custom code in MATLAB (v.R2020a, Mathworks).

## Nanoindentation of LNs

Mice were injected with an MPS vaccine or PBS control on day 0 and euthanized on days 4, 7 or 11. Inguinal vaccine-draining LNs were collected, embedded in 4% agarose in cryomolds, sectioned into 500 µm thick sections using a vibratome (Leica) and collected on glass slides[56–58]. Sections were kept hydrated in PBS and stored on ice for the duration of the procedure. Mechanical testing was performed using a G200 nanoindenter (Keysight Technologies) with a 400 µm spherical tip[59]. Sample points were defined in the centre and periphery of each section, and PBS was periodically added to maintain hydration. For each LN, outlier data points were identified through ROUT test (Q = 1%) in GraphPad Prism v.9 software and excluded from analysis. To generate mechanical maps of LN tissue, LN slices were first photographed and imported into Adobe Illustrator v.22.1 software to trace the perimeter. The LN edge was identified visually using the lens on the G200 nanoindenter; sample points spanning the LN were then selected, and x−y coordinates were collected and point location recorded in Illustrator. Indentation points were spaced to avoid tip overlap (minimum spacing between nearest points ~240 ± 46 µm) and the agarose mould (peripheral points selected ~150 µm from the LN edge). Visual maps were prepared by colouring the space closest to each sample point on a gradient corresponding to its stiffness measurement.

## Jump-start experiments

Mice were injected with MPS loaded with CpG and GM-CSF but no antigen. Seven days later, mice were administered a bolus vaccine or a PBS control. This vaccine strategy was applied to naïve mice or mice

bearing B16-OVA tumours (injected 1 day before jump-start). Peripheral blood was collected at 8 and 21 days after vaccination to assess antigen-specific CD8[+] T cell and antibody responses, respectively, or tumour growth and animal survival were monitored. When testing the timing of jump-start relative to bolus vaccination, the jump-start was administered at 11, 7 or 4 days before bolus vaccination and blood was analysed 8 days later.

## Tumour studies

The B16-OVA cell line was obtained from Prof. Kai Wucherpfennig's laboratory (Dana-Farber Cancer Institute, Boston, Massachusetts). B16-OVA cells were cultured in DMEM (Gibco) containing 10% FBS (Sigma) and 0.4 mg ml[−1] G418 selection antibiotic (Gibco). Cells were collected, counted using a haemocytometer and resuspended in cold PBS. B16-OVA cells ($2.5 \times 10^5$ for MPS tumour experiments or $1.25 \times 10^5$ for jump-start tumour experiments) were injected s.c. into the upper flank of mice using a 25G needle. Beginning on day 7, tumour sizes were measured externally using calipers and the area calculated as $A = (\pi/4) \times$ length × width. Mice were tracked and euthanized following Institutional Animal Care and Use Committee (IACUC) protocol according to a cumulative score incorporating body condition, weight loss and tumour size (tumours ≥17 mm in any two dimensions).

## Immune cell depletion experiments

Mice were administered MPS vaccines (containing GM-CSF, CpG, OVA) and treated with immune cell-depleting reagents. To deplete inflammatory monocytes, 20 μg MC-21 antibody was injected intraperitoneally (i.p.) daily for 5 total doses beginning at 0 (days 0–4), 1 (days 1–5) or 10 (days 10–14) days after vaccination. To deplete macrophages, 200 μl of clodronate liposome solution (Liposoma) was injected i.p. on days 0, 3 and 5 (early depletion) or 9, 12, 15 and 18 (late depletion) after vaccination. To deplete lymphocytes, a mixture of 250 μg each of anti-CD4, anti-CD8 and anti-B220 antibodies (Invivogen) was injected i.p. on days 9, 12, 15 and 18 after vaccination. To deplete neutrophils, 25 μg anti-Ly6G antibody was injected i.p. every day from days 9–15 and then increased to 50 μg from days 16–20 after vaccination. To combat host antibody responses and extend neutrophil depletion, 50 μg anti-rat kappa Ig (Invivogen) was injected i.p. every other day from days 9–19. Peripheral blood and/or immune organs were collected to verify depletions.

## Single-cell sequencing setup and sorting

Mice were treated with an MPS or bolus vaccine (containing GM-CSF, CpG and OVA) and dLNs were collected on days 20–21 and compared to naïve controls. LNs were digested in RPMI-1640 medium (Corning) containing 0.8 mg ml[−1] Dispase II, 0.2 mg ml[−1] Collagenase P and 0.1 mg ml[−1] DNase I (all Roche, procured from Sigma) until no visible LN pieces remained and passed through a 70 μm filter. Samples were incubated in TruStain FcX (Fc block, 1:200, BioLegend) before incubation with staining antibodies (PE/Cy7 anti-mouse CD45, BV711 anti-mouse CD3, BV421 anti-mouse CD19, FITC anti-mouse CD31 and allophycocyanin-conjugated anti-mouse podoplanin, all BioLegend) and one Hashtag antibody per replicate using BioLegend TotalSeq-B03 (155831, 155833, 155835, 155837, 155839; barcode sequences ACCCAC-CAGTAAGAC (-B0301), GGTCGAGAGCATTCA (-B0302), CTTGCCGCAT-GTCAT (-B0303), AAAGCATTCTTCACG (-B0304), CTTTGTCTTTGTGAG (-B0305), all diluted 1:100 in FACS buffer). Cell viability was assessed using eBioscience Fixable Viability Dye eFluor 780 (ThermoFisher). Cells were gently washed and resuspended in PBS + 2% FBS before sorting. Myeloid cells (viable, CD45[+] CD3[−] CD19[−]) were enriched using a BD FACS Aria cell sorter into RPMI + 30% FBS on ice. Cells were counted and resuspended at an adequate concentration for loading. Samples were prepared using a Chromium Single Cell 3′ reagent kit (10x Genomics) and run on a NovaSeq S4-200 cell (Illumina).

## Single-cell sequencing analysis

Single-cell RNA-seq data for each library from each cell type were processed with CellRanger 'count' (10x Genomics) using a custom reference gene annotation based on mouse reference genome GRCm38/mm10 and GENCODE gene models. DecontX was used to correct for ambient RNA contamination[60]. Corrected unique molecular identifier (UMI) count tables were read in R with the Seurat[61] package and counts were normalized to (CPM/100 + 1) and log transformed. We performed demultiplexing by hashing UMI counts with DemuxEM[62]. Individual libraries were merged into a Seurat object. Cells with less than 500 measured genes or over 5% mitochondrial counts were removed. Only cells labelled as 'singlets' on the basis of hashtag oligo counts were included. The top 2,000 most variable genes were selected via variance stabilizing transformation (FindVariableFeatures) and their expression was scaled (ScaleData). Principal component analysis was then performed in this gene space (RunPCA). Clustering was carried out on the basis of shared nearest neighbour between cells (FindNeighbors) and graph-based clustering (FindClusters). For graph-based clustering and generation of a uniform manifold approximation and projection (UMAP) reduction (RunUMAP), the same number of principal components was used as input[61].

Following exclusion of contaminating T cells, B cells, NK cells and stromal cells, 20,858 cells were used to identify clusters. A total of 30 PCs and a resolution of 0.1 were used to identify clusters. Clusters were annotated as DC2 cells (Sirpa, H2-Ab1), plasmacytoid DCs (Bst2, Siglech), migratory DCs (Ccr7), DC1 cells (Xcr1), Langerhans cells (Cd207), plasma cells (Ighg2b), inflammatory monocytes (Csf1r, Ly6c2), neutrophils (Cx3cr1) and proliferating DC2s (Mki67, Top2a). Data from individual populations (for example, inflammatory monocytes, DCs) were reprocessed with Seurat (DCs: 30 PCs, 0.5 resolution; monocytes: 30 PCs, 0.3 resolution; plasma cells: 30 PCs, 0.3 resolution). Markers for each cluster were detected using the FindAllMarkers function in Seurat with default parameters. The average expression of markers was calculated for each cluster using the AverageExpression function in Seurat, and per-gene $z$-scores were calculated for visualization using the pheatmap package. Density plots of cells were generated using the UMAP coordinates of cells from each condition using the LSD R package (https://cran.r-project.org/web/packages/LSD/index.html).

Cluster population frequency changes for each cluster were evaluated using Dunn's test, with $P$ values adjusted via the Benjamini–Hochberg procedure following a positive Kruskal–Wallis test. Pseudo-bulk differential expression analysis was performed using DESeq2 (ref. 63) and pooling counts across all cells within each replicate. Results were filtered for protein coding genes using abs(log₂FC) > 0.5 and basemean expression >300. cNMF programmes were identified using cNMF[26] with raw counts as input. cNMF was run with 20 iterations and 2,000 genes. Optimal $K$ was chosen from between 4 and 20 as the $K$ with a local maximum stability.

## Studies involving animals

Female C57BL/6J mice aged 6–8 weeks upon initiation of each study were purchased from Jackson Laboratory. Mice were housed with food and water provided ad libitum and light in 14:10 h light:dark cycles. Mice were housed at ambient 22 °C (±1 °C) temperature at 30–70% humidity. All animal procedures were compliant with relevant ethics regulations established by the National Institutes of Health and institutional guidelines with the approval of Harvard University's IACUC.

## Reporting summary

Further information on research design is available in the Nature Portfolio Reporting Summary linked to this article.

## Data availability

All datasets used in this study are included in the paper and its Supplementary Information, and are available via the Harvard Dataverse

repository at https://doi.org/10.7910/DVN/BB8OSJ (ref. 64). The scRNA-seq datasets generated during this study are available from the ArrayExpress database under accession code E-MTAB-13698 (ref. 65). Source data are provided with this paper.

## Code availability

The custom code for quantifying lymph node immunohistochemistry images is available in the Harvard Dataverse repository with the identifier https://doi.org/10.7910/DVN/BB8OSJ (ref. 64). For scRNA-seq, no new algorithms were developed for this project. Analysis code is available through the Open Science Framework (OSF) project with ID b5rcz (https://osf.io/b5rcz/?view_only=f15ba3fd86af40ddbb6ef6486f486a11).

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

## Acknowledgements

We thank C. Gerhardinger and Z. Niziolek for assistance with scRNA-seq and FACS; J. Alvarenga for support with nanoindentation; and K. Adu-Berchie, D. Zhang, S. Lee, M. Janes and A. Li for valuable scientific discussions. The MC-21 antibody was kindly provided by M. Mack (Universität Regensburg, Regensburg, Germany). We thank the Harvard Catalyst for biostatistician consultation. D.J.M. discloses support for the research described in this study from the National Institutes of Health/National Cancer Institute (U54 CA244726; R01 CA223255). This work was supported, in whole or in part, by the Bill and Melinda Gates Foundation (INV-055756). Under the grant conditions of the Foundation, a Creative Commons Attribution 4.0 Generic License has already been assigned to the author accepted manuscript version that might arise from this submission. A.J.N. recognizes a Graduate Research Fellowship from the National Science Foundation. B.R.F. recognizes support from the NIH/NIA (K99 AG065495). A.E.-A. received funding for this work from the European Union's Horizon 2020 research and innovation programme through Marie Sklodowska-Curie grant agreement no. 798504. G.B. acknowledges funding from the Swiss National Science Foundation via a Postdoc Mobility grant (P500PN_210721).

## Author contributions

A.J.N. and D.J.M. conceived the research. A.J.N., R.S.L., M.C.S., G.B., S.K., B.R.F., J.G.E., A.E.-A., C.M.T., M.O.D., K.W., H.I. and S.M. performed the research. D.J.M. and S.J.T. supervised the findings of this work. A.J.N. and D.J.M. wrote the paper, and R.S.L., K.W., S.J.T., B.R.F., A.E-A., C.M.T. and M.C.S. edited the paper.

## Competing interests

D.J.M. declares the following competing interests: Novartis, sponsored research, licensed IP; Immulus, equity; IVIVA, SAB; Attivare, SAB, equity; Lyell, licensed IP, equity. R.S.L., K.W., S.M. and S.J.T. are employed by Genentech, Inc. The other authors declare no competing interests.

## Additional information

**Extended data** is available for this paper at https://doi.org/10.1038/s41551-024-01209-3.

**Correspondence and requests for materials** should be addressed to David J. Mooney.

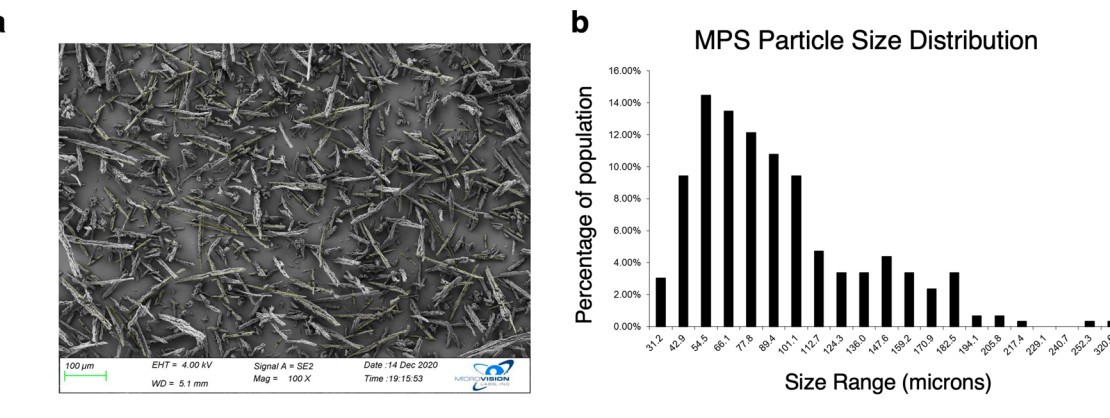

**a**

**b** MPS Particle Size Distribution

**c**

| Particle size (μm) | Surface area (m²/g) | Pore size (A) | Pore volume (cm³/g) |
|---|---|---|---|
| 85.9 | 530.4336 | 64.706 | 0.556031 |

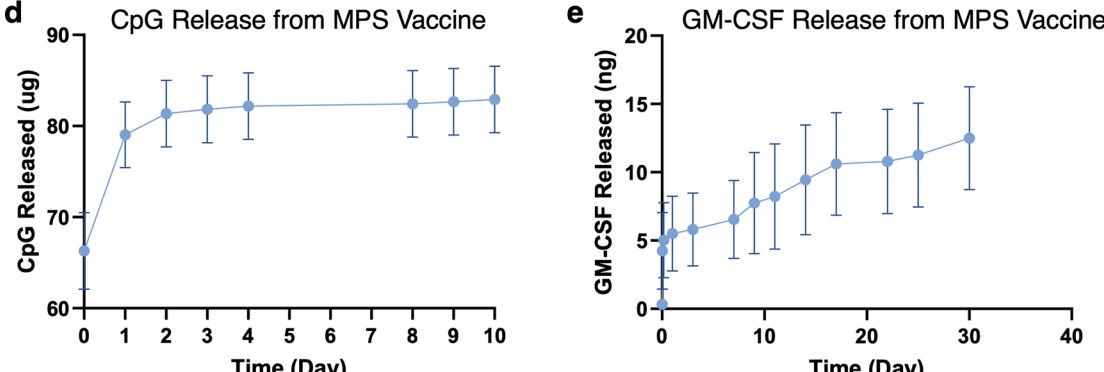

**d** CpG Release from MPS Vaccine

**e** GM-CSF Release from MPS Vaccine

**Extended Data Fig. 1 | MPS vaccine characterization. a**, Scanning electron microscopy imaging of MPS particles. **b**, MPS particle size distribution. **c**, Size and volume characterization of MPS particles. *In vitro* release curves of **d** CpG and **e** GM-CSF from MPS vaccines. n = 3 biologically independent replicates. For **d-e**, means depicted; error bars, s.d.

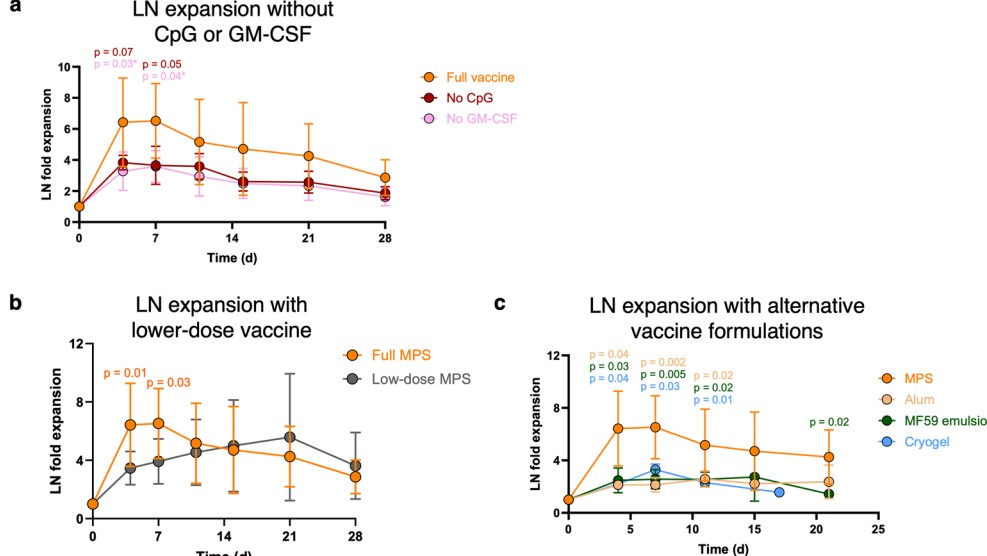

**Extended Data Fig. 2 | Comparing lymph-node expansion with MPS and alternative vaccine formulations.** Mice were immunized with MPS vaccines (GM-CSF, CpG, and OVA protein), MPS vaccines without CpG (GM-CSF and OVA only), or MPS vaccines without GM-CSF (CpG and OVA only). Additional mice were immunized with the same quantities of GM-CSF, CpG, and OVA in alum, emulsion (MF59), or alginate cryogel-based vaccines. A final group was dosed with an MPS vaccine containing a log-fold lower dose each of CpG and OVA. Vaccine-draining LNs were longitudinally imaged using high frequency ultrasound. **a**, Quantification of vaccine draining LN fold expansion over time, comparing the full MPS vaccine with formulations lacking either CpG or GM-CSF. Statistical analysis was performed using a one-way ANOVA with Dunnett's post hoc test. n = 4 (full MPS) or 5 (other groups) biologically independent animals per group. **b**, Vaccine draining LN fold expansion over time, comparing the full MPS vaccine with a low-dose MPS vaccine. n = 4 (full MPS) or 10 (low-dose MPS) biologically independent animals per group. The low-dose MPS group displays the combined results of two separate experiments. Statistical analysis was performed using a two-tailed t test. **c**, Vaccine draining LN fold expansion comparing MPS, alum, MF59 emulsion, and cryogel formulations. Statistical analysis was performed using a one-way ANOVA with Dunnett's post hoc test (t = 7, 11, 21 days) or a Kruskal-Wallis test with Dunn's post hoc test (t = 4 days). P values for the comparison between MPS and other vaccine systems are displayed on the plot. n = 4 (full MPS) or 5 (other groups) biologically independent animals per group. For **a-c**, values are normalized to the baseline volume for each individual LN; means depicted; error bars, s.d.

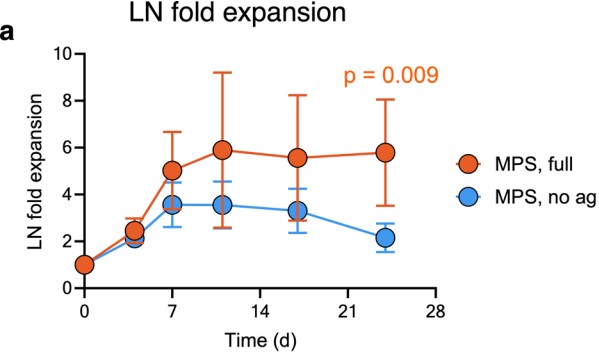

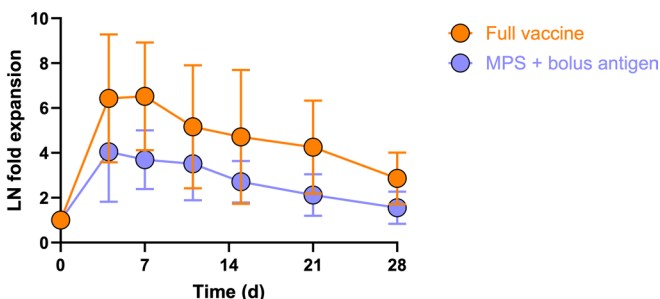

**Extended Data Fig. 3 | Comparing lymph-node expansion with MPS and vaccines lacking sustained antigen release.** Mice were immunized with MPS vaccines (GM-CSF, CpG, and OVA protein), MPS vaccines without antigen (GM-CSF and CpG alone), or MPS vaccines with bolus antigen (GM-CSF and CpG loaded into MPS, with OVA injected separately as a bolus). Vaccine-draining LNs were longitudinally imaged using high frequency ultrasound. **a**, LN fold expansion over time, normalized to volume at the day 0 timepoint. n = 5 biologically independent animals per group. **b**, Vaccine draining LN fold expansion over time, comparing the full MPS vaccine (antigen loaded in MPS material) to an MPS vaccine with a separate bolus injection of antigen. n = 4 (full MPS) or 5 (bolus antigen) biologically independent animals per group. For **a-b**, values are normalized to the baseline volume for each individual LN; means depicted; error bars, s.d.

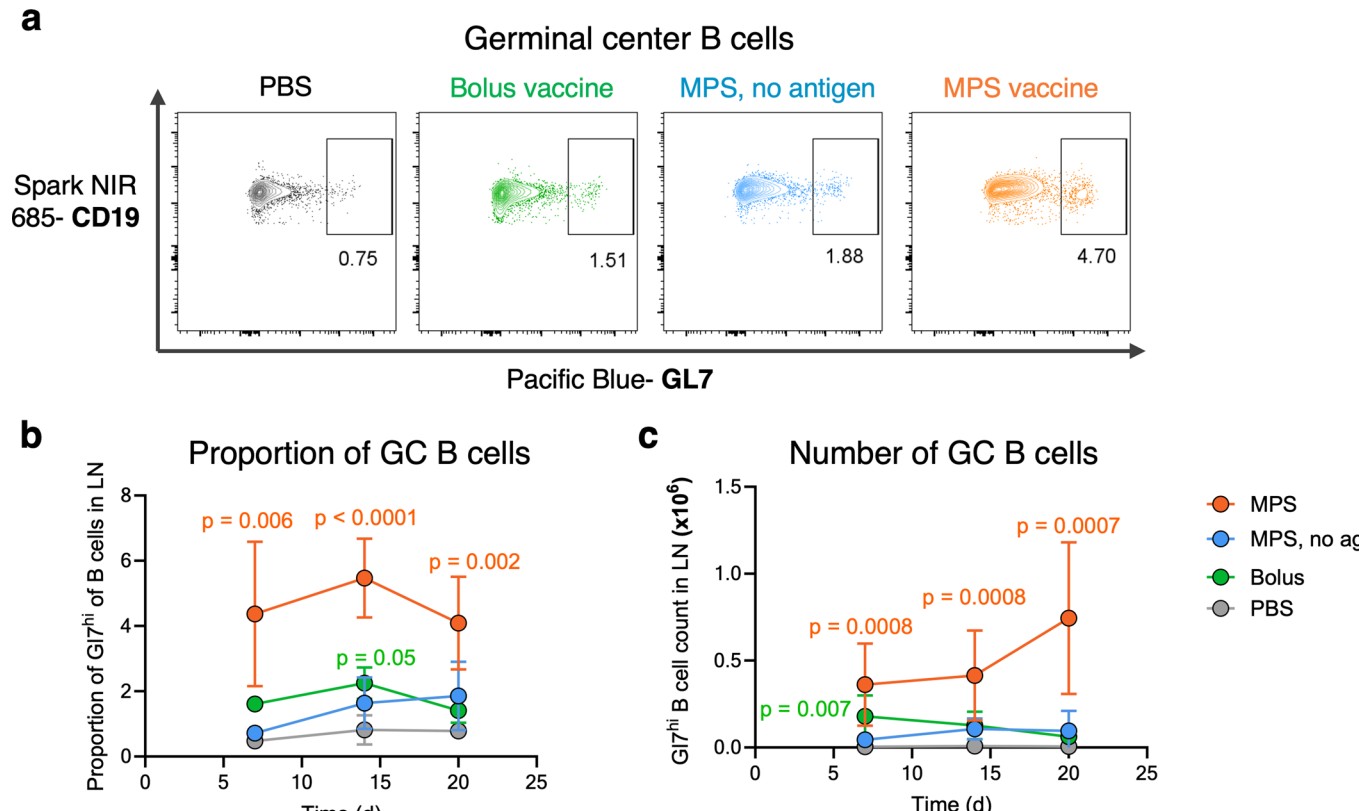

**Extended Data Fig. 4 | MPS vaccination induces potent germinal centre B cell responses in the LN.** Mice were treated with MPS or bolus vaccines (containing GM-CSF, CpG, OVA), MPS vaccine without antigen (GM-CSF, CpG only), or PBS, and LNs were collected on days 7, 14, and 20 for cellular analysis. n = 5 biologically independent animals per group per timepoint. **a**, Representative flow cytometry plots depicting CD3⁻CD19⁺ B cells from LNs on day 20. GL7hi B cells (germinal centre B cells) are gated. **b**, Proportion GL7hi of B cells in LNs over time. Statistical analyses were performed using analysis of variance (ANOVA) with Tukey's post hoc test (days 7 and 14) and Kruskal-Wallis test with Dunn's post hoc test (day 20). **c**, Number of GL7hi B cells (that is germinal centre B cells) in LNs over time. Statistical analyses were performed using analysis of variance (ANOVA) with Tukey's post hoc test (day 14) and Kruskal-Wallis test with Dunnett's post hoc test (days 7 and 20). For **b-c**, means depicted; error bars, s.d. Differences present between any group and the PBS control group are depicted.

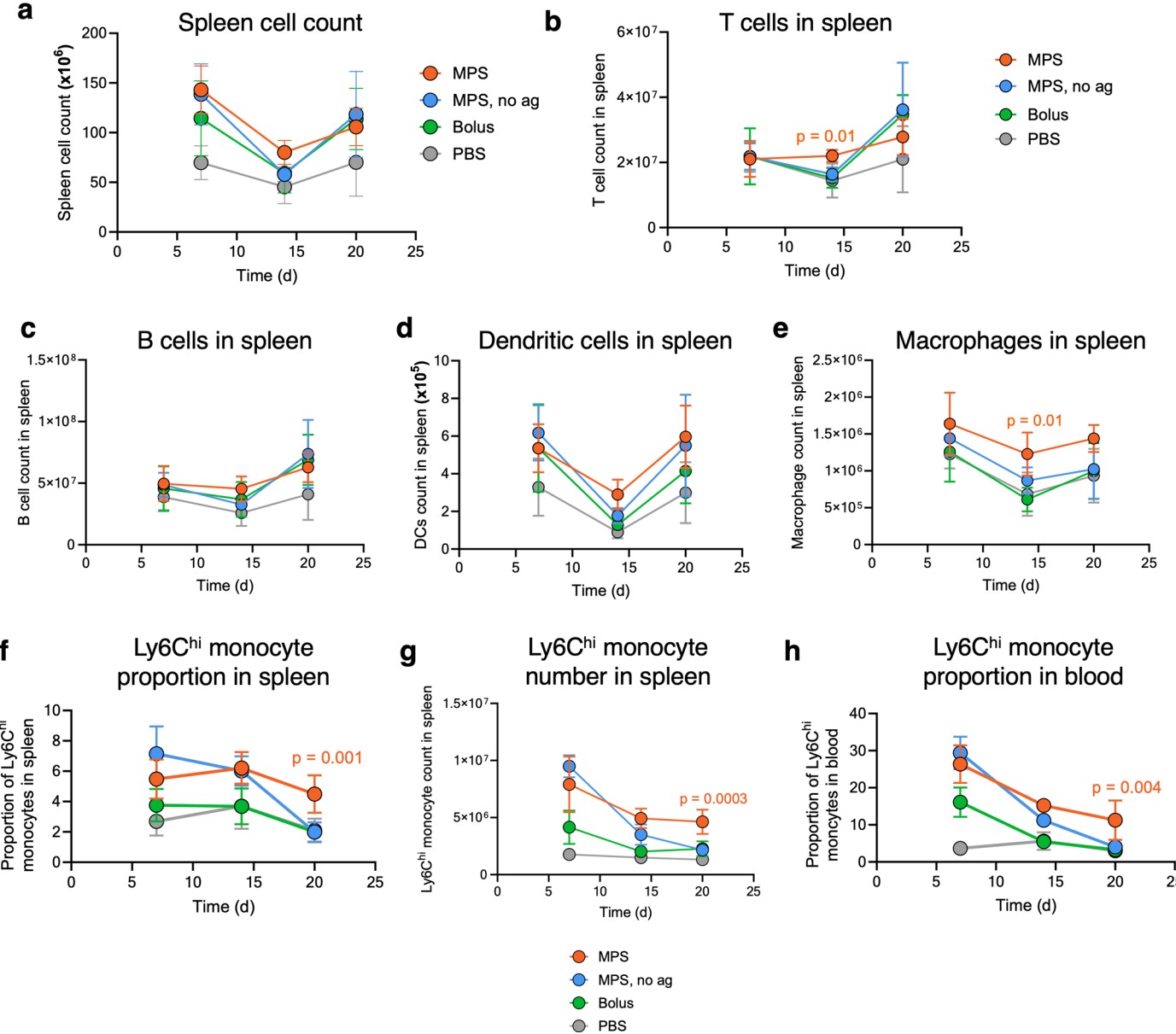

**Extended Data Fig. 5 | MPS vaccination minimally impacts splenic immune cells, with the exception of inflammatory monocytes.** Mice were treated with MPS or bolus vaccines (containing GM-CSF, CpG, OVA), MPS vaccine without antigen (GM-CSF and CpG only), or PBS, and spleens and blood were collected on days 7, 14, and 20 for cellular analysis. **a**, Total cell counts in the spleen. **b**, Number of T cells (CD3$^+$CD19$^-$) in the spleen. Statistical analysis was performed using a one-way ANOVA with Tukey's post hoc test. **c**, Number of B cells (CD3$^-$CD19$^+$) in the spleen. **d**, Numbers of dendritic cells (CD3$^-$CD19$^-$Ly6C$^-$F4/80$^-$CD11c$^+$MHCII$^+$) in the spleen. **e**, Number of macrophages (CD3$^-$CD19$^-$Ly6C$^-$CD11b$^+$F4/80$^+$) in the spleen. Statistical analysis was performed using a one-way

ANOVA with Tukey's post hoc test. **f**, Proportion of inflammatory monocytes (CD3$^-$CD19$^-$Ly6C$^{hi}$CD11b$^+$) of total cells in the spleen. Statistical analysis was performed using a one-way ANOVA with Tukey's post hoc test. **g**, Number of inflammatory monocytes in the spleen. Statistical analysis was performed using a one-way ANOVA with Tukey's post hoc test. **h**, Proportion of inflammatory monocytes in the blood. Statistical analysis was performed using a one-way ANOVA with Tukey's post hoc test. For **a-h**, n = 5 biologically independent animals per group per timepoint; means depicted; error bars, s.d.; statistical significance is shown between the MPS dLN group and all other groups.

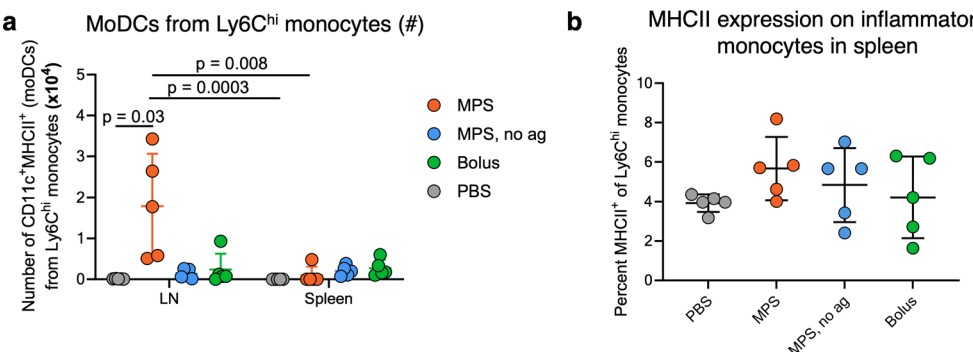

**Extended Data Fig. 6 | Monocyte-derived DCs in LNs and spleens.** Mice were treated with MPS or bolus vaccines (containing GM-CSF, CpG, OVA), MPS vaccine without antigen (GM-CSF and CpG only), or PBS, and LNs and spleens were collected on day 20 for cellular analysis. **a**, Number of monocyte-derived DCs from Ly6$^{hi}$ inflammatory monocytes (CD3$^-$CD19$^-$CD11b$^+$Ly6C$^{hi}$CD11c$^+$MHC II$^+$ cells) in the LN and spleen at day 20. Statistical analysis was performed using a Kruskal-Wallis test with Dunn's post hoc test. **b**, Proportion of inflammatory monocytes expressing MHCII in the spleen on day 20. Differences between groups are not significant. For **a-b**, n = 5 biologically independent animals per group; means depicted; error bars, s.d.

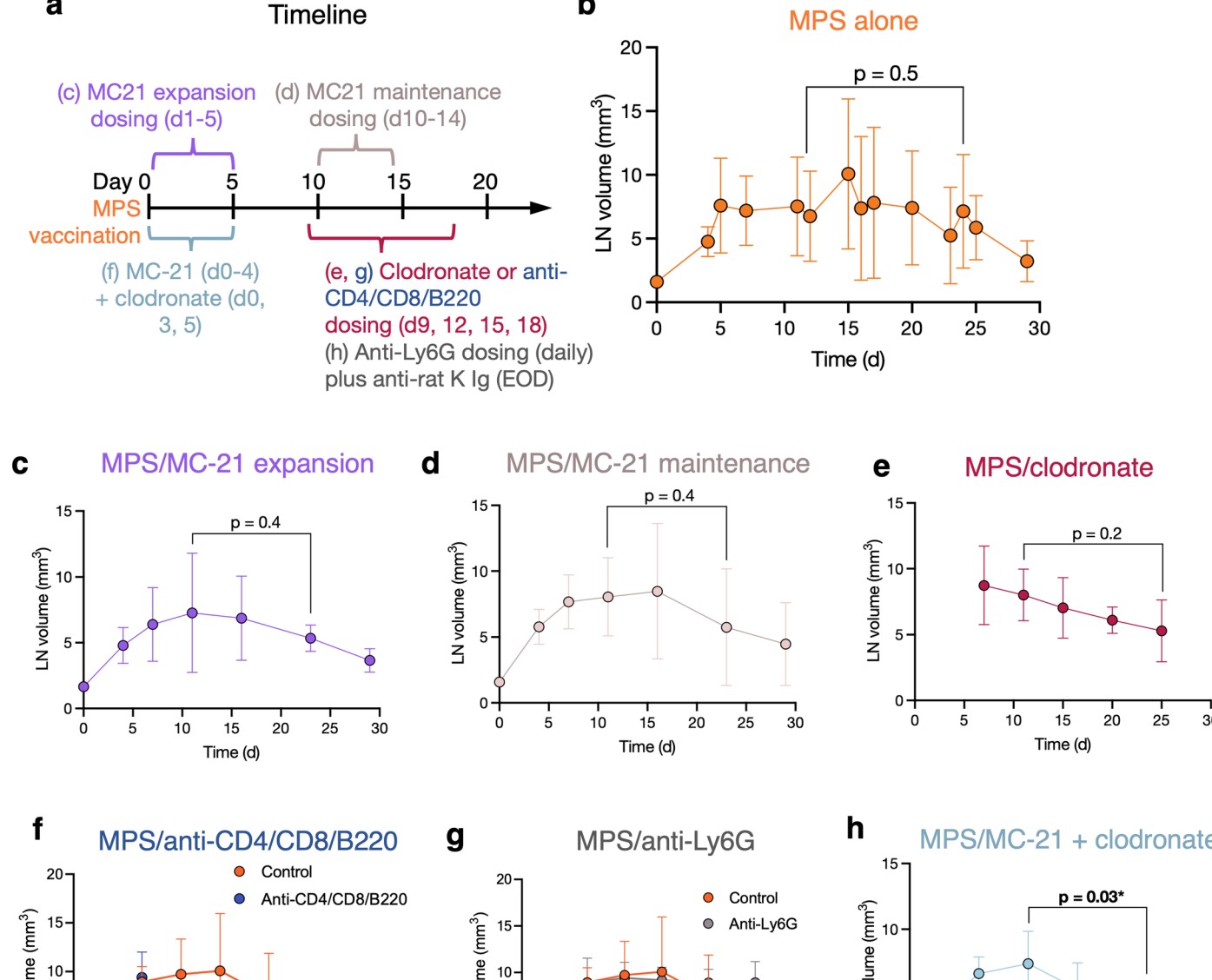

**Extended Data Fig. 7 | Immune intervention alters maintenance of LN expansion after vaccination.** Mice were treated with MPS vaccines (containing GM-CSF, CpG, and OVA) on day 0 and administered depleting reagents: MC-21 antibody (to deplete Ly6C$^{hi}$ inflammatory monocytes), clodronate liposomes (to deplete macrophages/monocytes), anti-CD4/CD8/B220 (to deplete T and B lymphocytes), and anti-Ly6G/rat κ Ig (to deplete neutrophils). Lymph node expansion kinetics were tracked using HFUS imaging. **a**, Timeline of treatments. LN expansion curves are shown for mice treated with **b** MPS vaccine alone (n = 4 biologically independent animals on days 15 and 20, 10 on day 0, 14 on day 7, and 5 at all other timepoints), **c** MPS vaccine plus MC-21 (days 1–5; n = 5 biologically independent animals), **d** MPS vaccine plus MC-21 (days 10–14; n = 5 biologically independent animals), **e** MPS vaccine plus clodronate liposomes (days 9, 12, 15, 18; n = 4 biologically independent animals on days 7, 11, 15, 20 and 2 on day 25), **f** MPS plus anti-CD4/CD8/B220 antibodies (n = 3 biologically independent animals), **g** MPS plus anti-Ly6G and anti-rat κ Ig (n = 3 biologically independent animals), or **h** MPS vaccine plus MC-21 (days 0–4) and clodronate liposomes (days 0, 3, 5) (n = 5 biologically independent animals). Statistical analyses were performed using a two-tailed t test; significance is shown between LN volumes on day 11–12 and 23–25. Means depicted; error bars, s.d.

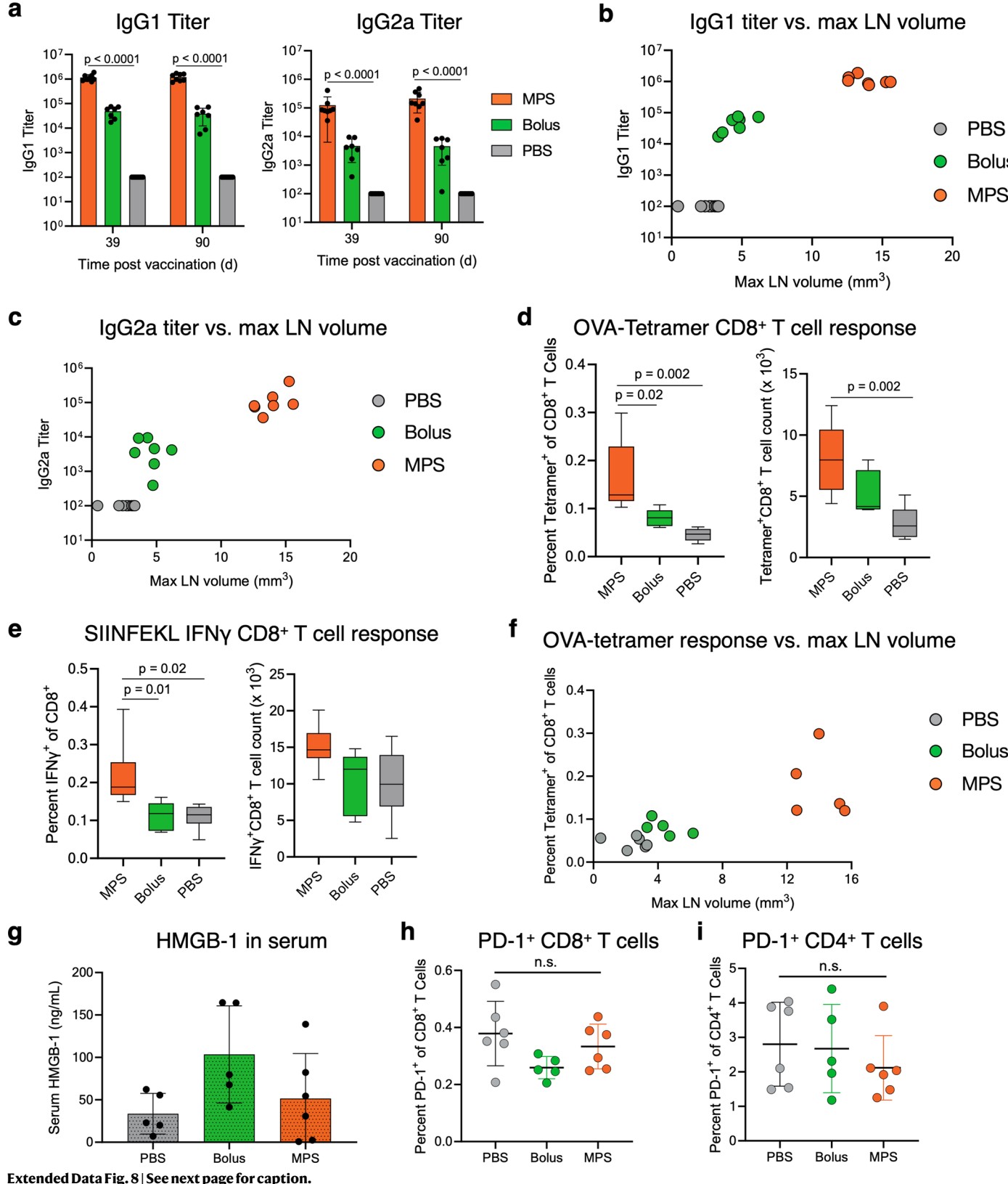

**Extended Data Fig. 8 | See next page for caption.**

**Extended Data Fig. 8 | Long-lived vaccine responses and LN expansion.** Mice were immunized with MPS or bolus vaccines delivering GM-CSF, CpG, and OVA protein. 39 and 90 days after vaccination, blood was collected for OVA-specific serum antibody titre analysis using ELISA. 103 days after vaccination, spleens were collected for T cell analysis. **a**, Serum IgG1 (left) and IgG2a (right) titres against OVA. n = 7 (bolus) or 8 (MPS and PBS) biologically independent animals per group; means depicted; error bars, s.d. Statistical analysis was performed using a Kruskal-Wallis test with Dunn's post hoc test. **b**, IgG1 titre and **c** IgG2 titre versus the maximum LN volume in each mouse. Proportion (left) and number (right) of **d** OVA-tetramer⁺ CD8⁺ T cells and **e** IFNγ⁺ CD8⁺ T cells in spleens. Whiskers extend min-to-max with box displaying 25th to 75th percentiles. For

**d** and **e** (right), statistical analysis was performed using analysis of variance (ANOVA) with Tukey's post hoc test (d, e right). For **e** (left), statistical analysis was performed using Kruskal-Wallis test with Dunn's post hoc test. **f**, OVA-tetramer⁺ CD8⁺ T cell proportion versus maximum LN volume. **g**, HMGB-1 concentration in mouse serum 21 days after vaccination. n = 5 (PBS and bolus) or 6 (MPS) biologically independent animals per group. **h**, PD-1 expression on CD8⁺ T cells. **i**, PD-1 expression on CD4⁺ T cells. For **h-i**, statistical analysis was performed using analysis of variance (ANOVA) with Tukey's post hoc test; no significant differences were found between groups. For **d, e, h**, and **i**, n = 5 (bolus) or 6 (MPS and PBS) biologically independent animals per group; means depicted; error bars, s.d.

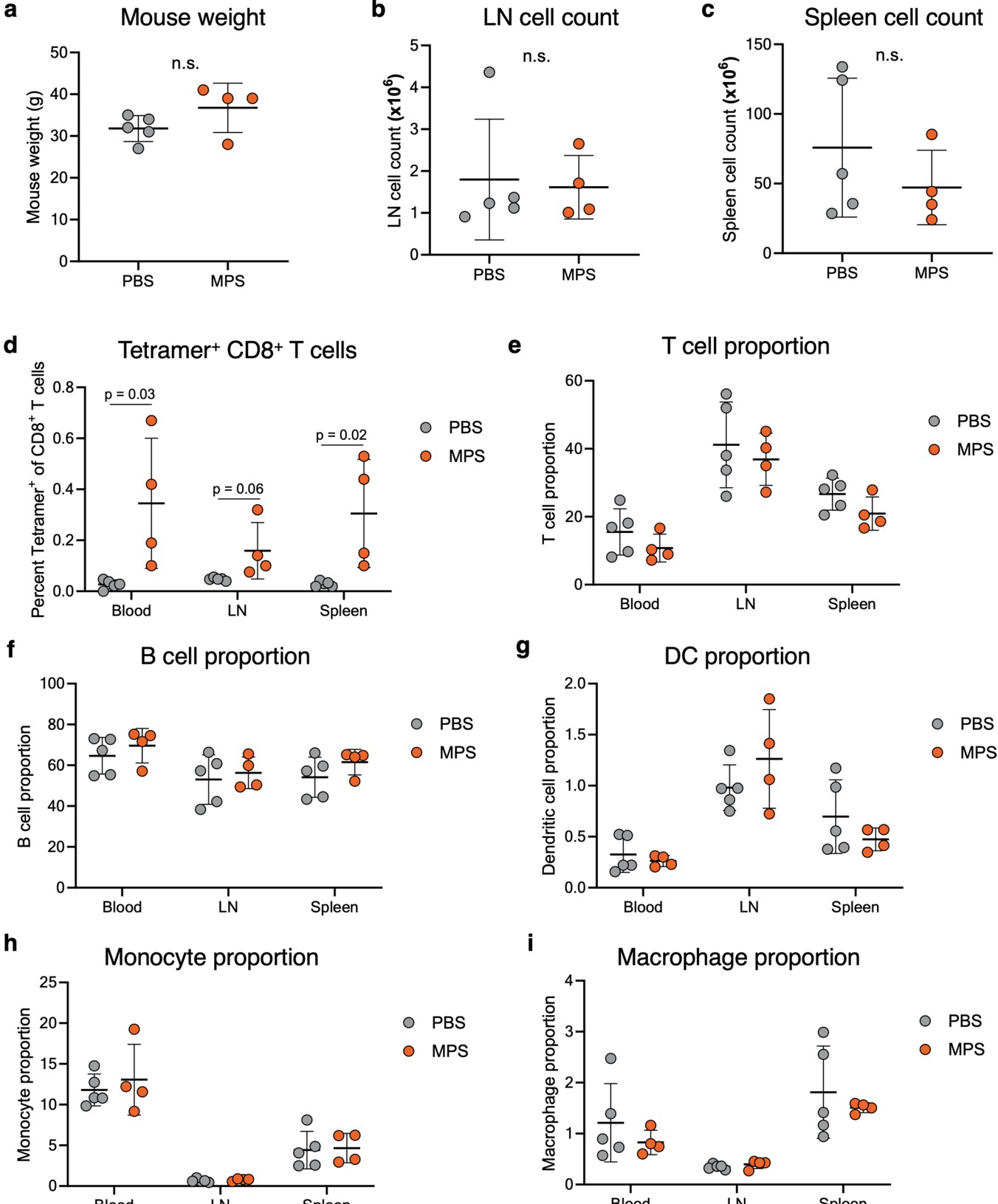

**Extended Data Fig. 9 | Long-term response to MPS vaccination.** Mice were injected on day 0 with PBS or an MPS vaccine consisting of GM-CSF, CpG, and OVA protein. 485 days later, mice were euthanized and blood, LNs, and spleens were collected for analysis. **a**, Mouse weight at 485 days. Total cell counts in draining **b** LN and **c** spleen. **d** OVA-tetramer binding proportion of CD8⁺ T cells. Proportions of **e** T cells (CD19⁻CD3⁺), **f** B cells (CD19⁺CD3⁻), **g** DCs (CD19⁻CD

3⁻Ly6C⁻F4/80⁻CD11c⁺MHCII⁺), **h** monocytes (CD19⁻CD3⁻CD11b⁺Ly6C⁺), and **i** macrophages (CD19⁻CD3⁻Ly6C⁻CD11b⁺F4/80⁺) in the blood, LNs and spleens. For **a-i**, statistical analyses were performed using two-tailed t tests; n = 4 (MPS) or 5 (PBS) biologically independent animals per group; means depicted; error bars, s.d.

**a**

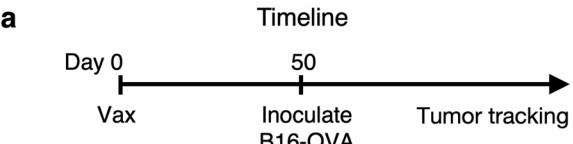

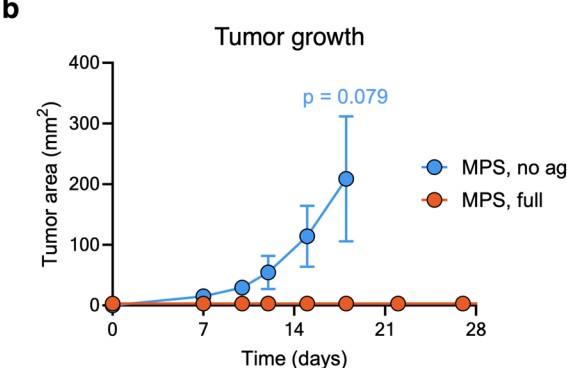

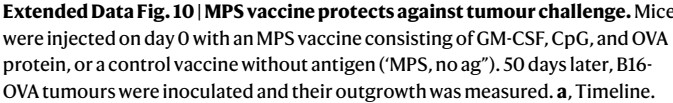

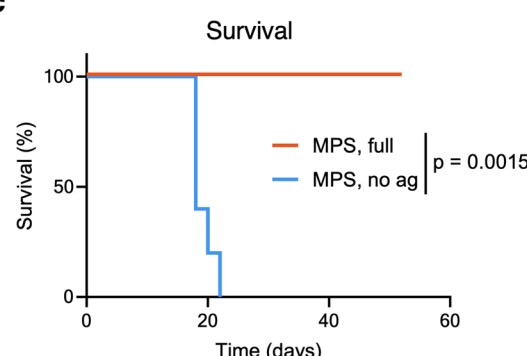

**Extended Data Fig. 10 | MPS vaccine protects against tumour challenge.** Mice were injected on day 0 with an MPS vaccine consisting of GM-CSF, CpG, and OVA protein, or a control vaccine without antigen ('MPS, no ag"). 50 days later, B16-OVA tumours were inoculated and their outgrowth was measured. **a**, Timeline.

**b**, Tumour growth curves. Statistical analysis was performed using a Mann-Whitney test. Means depicted; error bars, s.d. **c**, Kaplan-Meier curves depicting survival of groups. Statistical analysis was performed using a log-rank (Mantel-Cox) test. For **b-c**, n = 5 biologically independent animals per group.

# Reporting Summary

## Statistics

For all statistical analyses, confirm that the following items are present in the figure legend, table legend, main text, or Methods section.

| n/a | Confirmed | |
|---|---|---|
| ☐ | ☒ | The exact sample size (*n*) for each experimental group/condition, given as a discrete number and unit of measurement |
| ☐ | ☒ | A statement on whether measurements were taken from distinct samples or whether the same sample was measured repeatedly |
| ☐ | ☒ | The statistical test(s) used AND whether they are one- or two-sided<br>*Only common tests should be described solely by name; describe more complex techniques in the Methods section.* |
| ☒ | ☐ | A description of all covariates tested |
| ☐ | ☒ | A description of any assumptions or corrections, such as tests of normality and adjustment for multiple comparisons |
| ☐ | ☒ | A full description of the statistical parameters including central tendency (e.g. means) or other basic estimates (e.g. regression coefficient) AND variation (e.g. standard deviation) or associated estimates of uncertainty (e.g. confidence intervals) |
| ☐ | ☒ | For null hypothesis testing, the test statistic (e.g. *F*, *t*, *r*) with confidence intervals, effect sizes, degrees of freedom and *P* value noted<br>*Give P values as exact values whenever suitable.* |
| ☒ | ☐ | For Bayesian analysis, information on the choice of priors and Markov chain Monte Carlo settings |
| ☒ | ☐ | For hierarchical and complex designs, identification of the appropriate level for tests and full reporting of outcomes |
| ☒ | ☐ | Estimates of effect sizes (e.g. Cohen's *d*, Pearson's *r*), indicating how they were calculated |

*Our web collection on statistics for biologists contains articles on many of the points above.*

## Software and code

Policy information about availability of computer code

| Data collection | MATLAB (vR2020a, Mathworks) was used to quantify actin distribution in lymph nodes, SpectroFlo (v3.0, Cytek) was used for cytometry data acquisition, and Zen Black (v2.1, Zeiss) was used for imaging data collection. The custom code for quantifying lymph-node immunohistochemistry images is available at the Harvard Dataverse repository with the identifier https://doi.org/10.7910/DVN/BB8OSJ (ref. 64). For scRNA-seq, no new algorithms were developed for this project. Analysis code is available through the Open Science Framework (OSF) project with ID b5rcz (https://osf.io/b5rcz/?view_only=f15ba3fd86af40ddbb6ef6486f486a11). |
|---|---|
| Data analysis | FlowJo (v10.4) was used for flow-cytometry data analysis, VevoLab (v 5.6.1, VisualSonics) was used for volume quantification of lymph-node images, Prism (v9 and v10, Graphpad) were used for statistical analysis and plotting, ImageJ (v1.53m) was used for image analysis, and Adobe Illustrator (v22.1, Adobe) was used for generating maps of lymph-node mechanical properties. |

For manuscripts utilizing custom algorithms or software that are central to the research but not yet described in published literature, software must be made available to editors and reviewers. We strongly encourage code deposition in a community repository (e.g. GitHub). See the Nature Portfolio guidelines for submitting code & software for further information.

## Data

Policy information about availability of data

All manuscripts must include a data availability statement. This statement should provide the following information, where applicable:

- Accession codes, unique identifiers, or web links for publicly available datasets
- A description of any restrictions on data availability
- For clinical datasets or third party data, please ensure that the statement adheres to our policy

All datasets used in this study are included in the paper and its Supplementary information, and are available from the Harvard Dataverse repository with the identifier https://doi.org/10.7910/DVN/BB8OSJ (ref. 64). The scRNA-seq datasets generated during this study are available from the ArrayExpress database under accession code E-MTAB-13698 (https://www.ebi.ac.uk/biostudies/arrayexpress/studies/E-MTAB-13698) (ref. 65).

## Research involving human participants, their data, or biological material

Policy information about studies with human participants or human data. See also policy information about sex, gender (identity/presentation), and sexual orientation and race, ethnicity and racism.

| | |
|---|---|
| Reporting on sex and gender | The study did not involve human participants. |
| Reporting on race, ethnicity, or other socially relevant groupings | – |
| Population characteristics | – |
| Recruitment | – |
| Ethics oversight | – |

Note that full information on the approval of the study protocol must also be provided in the manuscript.

# Field-specific reporting

Please select the one below that is the best fit for your research. If you are not sure, read the appropriate sections before making your selection.

☒ Life sciences ☐ Behavioural & social sciences ☐ Ecological, evolutionary & environmental sciences

For a reference copy of the document with all sections, see nature.com/documents/nr-reporting-summary-flat.pdf

# Life sciences study design

All studies must disclose on these points even when the disclosure is negative.

| | |
|---|---|
| Sample size | Sample sizes for the in vivo experiments were determined empirically on the basis of results from prior publications, selecting appropriate numbers to achieve statistical significance while minimizing the number of mice used in accordance with the 3 Rs of animal use (Kim, J., et al. Nature Biotechnology 2015, doi: 10.1038/nbt.3071; Dellacherie, M., et al. Advanced Functional Materials, https://doi.org/10.1002/adfm.202002448) and with input and approval from Harvard University's Institutional Animal Care and Use Committee. |
| Data exclusions | In analysing nanoindentation data of lymph nodes, outlier data points (such as when the nanoindenter tip struck the sample mould rather than the sample itself) were excluded based on outlier analyses using GraphPad Prism v9.0. Mice with undetectable or inconsistent LN measurements (rare, 1–2 across hundreds in studies) were excluded from further analysis. |
| Replication | Replication information is included in the figure legends. Major experiments were conducted two to three times independently, to confirm results. |
| Randomization | The animals were randomized prior to initiation of the in vivo studies. In tumour studies, mouse tumours were measured prior to treatment and subsequently randomized into treatment groups. |
| Blinding | The majority of the data acquired were not blinded owing to quantifiable and unbiased outputs, feasibility and personnel limitations, or because the initial studies were exploratory. The repeat experiment confirming efficacy of the jump-start vaccine was performed in a blinded fashion. |

# Reporting for specific materials, systems and methods

We require information from authors about some types of materials, experimental systems and methods used in many studies. Here, indicate whether each material, system or method listed is relevant to your study. If you are not sure if a list item applies to your research, read the appropriate section before selecting a response.

## Materials & experimental systems

| n/a | Involved in the study |
|-----|------------------------|
| ☐ | ☒ Antibodies |
| ☐ | ☒ Eukaryotic cell lines |
| ☒ | ☐ Palaeontology and archaeology |
| ☐ | ☒ Animals and other organisms |
| ☒ | ☐ Clinical data |
| ☒ | ☐ Dual use research of concern |
| ☒ | ☐ Plants |

## Methods

| n/a | Involved in the study |
|-----|------------------------|
| ☒ | ☐ ChIP-seq |
| ☐ | ☒ Flow cytometry |
| ☒ | ☐ MRI-based neuroimaging |

## Antibodies

Antibodies used

The following antibodies were used, and are listed in Supplementary Table 1 and Supplementary Table 2 of the manuscript:

B220 BV510 RA3-6B2 BioLegend 1.5
B220 BV570 RA3-6B2 BioLegend 2
CD3 APC/Fire 810 17A2 BioLegend 1.25
CD3 PerCP/Cy5.5 17A2 BioLegend 2
CD3 Pacific Blue 17A2 BioLegend 1.25
CD3 BV570 17A2 BioLegend 2
CD3 PE/Cy5 145-2C11 BioLegend 1.25
CD4 PerCP/Cy5.5 GK1.5 BioLegend 1
CD4 BV711 RM4-5 BioLegend 1.5
CD4 PE/Dazzle 594 RM4-5 BioLegend 1.25
CD8 BV 605 53-6.7 BioLegend 1.25
CD8 FITC 53-6.7 BioLegend 0.5
CD11b PerCP/Cy5.5 M1/70 BioLegend 1
CD11b BV421 M1/70 BioLegend 1
CD11c PE/Cy5 N418 BioLegend 1.25
CD11c PE/Cy7 N418 BioLegend 1
CD19 Spark NIR 685 6D5 BioLegend 1
CD24 BV421 M1/69 BioLegend 1
CD26 PE/Cy7 H194-112 BioLegend 1.8
CD31 FITC MEC13.3 BioLegend 1.5
CD35/21 (CR1/2) APC 7E9 BioLegend 1.5
CD44 BV 510 IM7 BioLegend 1.25
CD44 PE IM7 BioLegend 1.25
CD44 APC/Fire 750 IM7 BioLegend 1.25
CD45 APC/Fire 750 30-F11 BioLegend 1
CD45 PerCP/Cy5.5 30-F11 BioLegend 1
CD45 PE/Cy7 30-F11 BioLegend 1
CD49b PE/Dazzle 594 DX5 BioLegend 1
CD54 (ICAM-1) Pacific Blue YN1/1.7.4 BioLegend 1.5
CD62L BV 785 MEL-14 BioLegend 1.25
CD62L PE/Cy7 MEL-14 BioLegend 1.25
CD62L FITC MEL-14 BioLegend 1.25
CD64 BV 711 X54-5/7.1 BioLegend 1.8
CD68 PE FA-11 BioLegend 1
CD103 BV785 2E7 BioLegend 2
CD301b PE/Dazzle 594 URA-1 BioLegend 2
CLEC-2 PE 17D9 BioLegend 1.25
F4/80 APC/Fire 750 BM8 BioLegend 1
F4/80 APC BM8 BioLegend 1
Gl7 Pacific Blue GL7 BioLegend 0.5
Granzyme B APC/Fire 750 QA16A02 BioLegend 5
IFNγ APC XMG1.2 BioLegend 1.5
IFNγ PE XMG1.2 BioLegend 1.5
IL-2 PE JES6-5H4 BioLegend 3
Ly6C AF 700 HK1.4 BioLegend 0.5
Ly6C APC HK1.4 BioLegend 1
Ly6G BV785 1A8 BioLegend 1.5
Ly6G BV570 1A8 BioLegend 2
Ly6G PE/Cy7 1A8 BioLegend 1.5
MAdCAM1 PE MECA-367 BioLegend 1.5
MHCII FITC M5/114.15.2 BioLegend 0.5
NK1.1 PE/Cy5 PK136 BioLegend 2
NK1.1 BV570 PK136 BioLegend 2

PD-1 PerCP/Cy5.5 RMP1-30 BioLegend 2
PD-1 BV510 29F.1A12 BioLegend 2
Podoplanin PE/Cy7 8.1.1 BioLegend 2.5
Podoplanin APC 8.1.1 BioLegend 1.25
Siglec H PE 551 BioLegend 1
SIINFEKL-H-2Kb PE/Dazzle 594 25-D1.16 BioLegend 2
TNFα PE/Cy7 MP6-XT22 BioLegend 1.25

B220 AF594 RA3-6B2 BioLegend 1:50
CCR2 n/a EPR20844 Abcam 1:50
CD3 AF647 17A2 BioLegend 1:100
CD11b AF488 M1/70 BioLegend 1:100
Collagen I n/a AB765P Millipore Sigma 1:40
Collagen VI AF488 ER-TR7 Santa Cruz Biotechnology 1:50
F-actin (phalloidin) AF488 A12379 Thermo Fisher Scientific 1:100
Hyaluronic acid Biotin 385911 Millipore Sigma 1:50
Ly6C Biotin ER-MP20 Abcam 1:200

| | |
|---|---|
| Validation | Antibodies were used according to manufacturer recommendations and gated based on known negative (such as FMO or secondary only) controls. At first use, the antibody volumes used in experiments were tested on control mouse samples to verify reactivity. The validation statement included on the manufacturer's (BioLegend's) website, states "Each lot of this antibody is quality control tested by intracellular immunofluorescent staining with flow cytometric analysis. For flow cytometric staining, the suggested use of this reagent is ≤0.25 µg per million cells in 100 µl volume. It is recommended that the reagent be titrated for optimal performance for each application." Initial antibody volume choices were based on this recommendation. |

## Eukaryotic cell lines

Policy information about cell lines and Sex and Gender in Research

| | |
|---|---|
| Cell line source(s) | The B16-OVA cell line was obtained from Kai Wucherpfennig's laboratory (Dana Farber Cancer Institute, Boston, MA). This cell line is derived from the commercially available B16-F10 line (ATCC) and can be purchased commercially from Biocytogen. |
| Authentication | The cell line was not authenticated. |
| Mycoplasma contamination | The cell line was not tested for mycoplasma contamination. |
| Commonly misidentified lines (See ICLAC register) | No commonly misidentified cell lines were used. |

## Animals and other research organisms

Policy information about studies involving animals; ARRIVE guidelines recommended for reporting animal research, and Sex and Gender in Research

| | |
|---|---|
| Laboratory animals | Female C57BL/6J mice, aged 6–8 weeks on initiation of each study, were purchased from Jackson Laboratory (Bar Harbor, ME). Mice were housed with food and water ad libitum, and light was provided in 14-h-light–10-h-dark cycles. Mice were housed at ambient temperature 22 deg (+/− 1 deg) at 30–70% humidity. |
| Wild animals | The study did not involve wild animals. |
| Reporting on sex | Female mice were used in all experiments. |
| Field-collected samples | The study did not involve samples collected from the field. |
| Ethics oversight | All animal procedures were compliant with relevant ethical regulations established by the National Institutes of Health and institutional guidelines with the approval of Harvard University's Institutional Animal Care and Use Committee. |

Note that full information on the approval of the study protocol must also be provided in the manuscript.

# Flow Cytometry

## Plots

Confirm that:

☒ The axis labels state the marker and fluorochrome used (e.g. CD4-FITC).

☒ The axis scales are clearly visible. Include numbers along axes only for bottom left plot of group (a 'group' is an analysis of identical markers).

☒ All plots are contour plots with outliers or pseudocolor plots.

☒ A numerical value for number of cells or percentage (with statistics) is provided.

## Methodology

| | |
|---|---|
| Sample preparation | Blood was collected from mice retro-orbitally using heparinized capillary tubes (Fisherbrand) and stored in heparinized collection tubes (BD Biosciences) on ice. Red blood cells were lysed using ACK Lysing Buffer (Quality Biological). LNs were explanted and digested in RPMI-1640 (Corning) containing 0.8mg/mL Dispase II, 0.2mg/mL Collagenase P, and 0.1mg/mL DNase I (all Roche, procured from Sigma) until no visible LN pieces remained, following established protocols. |
| Instrument | Samples were run on an Aurora Spectral Analyzer (Cytek) or LSRII flow cytometer (BD Biosciences). |
| Software | Data were acquired using SpectroFlo software (Cytek) and analysed using Flowjo v10 software. |
| Cell population abundance | Abundance of cell populations within flow-cytometry results are specified throughout the manuscript in reference to specific experiments. |
| Gating strategy | Cells were gated on (1) SSC-A vs. FSC-A, selecting the major cell population in accordance with published datasets, (2) fSC-H vs. FSC-A, selecting single cells based on the visible population distribution, (3) live cells based on negative staining for a viability dye, and then (4) subsequent stains dependent on the panel of interest. Sample gating strategies are included throughout the manuscript and particularly in Supplementary Fig. 7. |

☒ Tick this box to confirm that a figure exemplifying the gating strategy is provided in the Supplementary Information.

