## [Peer Review File · Nature Biomedical Engineering]

Durable lymph-node expansion is associated with the efficacy of therapeutic vaccination

Corresponding author: David Mooney

Editorial note

This document includes relevant written communications between the manuscript's corresponding author and the editor and reviewers of the manuscript during peer review. It includes decision letters relaying any editorial points and peer-review reports, and the authors' replies to these (under 'Rebuttal' headings). The editorial decisions are signed by the manuscript's handling editor, yet the editorial team and ultimately the journal's Chief Editor share responsibility for all decisions.

Any relevant documents attached to the decision letters are referred to as **Appendix #**, and can be found appended to this document. Any information deemed confidential has been redacted or removed. Earlier versions of the manuscript are not published, yet the originally submitted version may be available as a preprint. Because of editorial edits and changes during peer review, the published title of the paper and the title mentioned in below correspondence may differ.

Correspondence

Wed 22 Mar 2023

Decision on Article nBME-23-0221

Dear Dr Mooney,

Thank you again for submitting to *Nature Biomedical Engineering* your manuscript, "Durable lymph node expansion strengthens vaccine efficacy". The manuscript has been seen by 4 experts, whose reports you will find at the end of this message.

You will see that the reviewers appreciate aspects of the work. However, they articulate concerns about the degree of support for some of the claims and about the advance that the work represents over relevant published studies, and provide useful suggestions for improvement. We hope that with significant further effort you can address the criticisms, increase the level of significance of the study, and convince the reviewers of its merits. In particular, we would expect that a revised version of the manuscript provides:

- * Incorporation of antigen-free and adjuvant-free control groups, and evidence of whether a vaccine using alum as adjuvant also leads to durable lymph-node expansion.
- * Extended evidence of the immune responses to the vaccine, as per the various comments from Reviewers #1, #2 and #3.
- * Thorough characterization of the vaccine formulation.
- * Extended discussion of the results, to aid their interpretability.

When you are ready to resubmit your manuscript, please upload the revised files, a point-by-point rebuttal to the comments from all reviewers, the reporting summary, and a cover letter that explains the main improvements included in the revision and responds to any points highlighted in this decision.Please follow the following recommendations:

- * Clearly highlight any amendments to the text and figures to help the reviewers and editors find and understand the changes (yet keep in mind that excessive marking can hinder readability).
- * If you and your co-authors disagree with a criticism, provide the arguments to the reviewer (optionally, indicate the relevant points in the cover letter).
- * If a criticism or suggestion is not addressed, please indicate so in the rebuttal to the reviewer comments and explain the reason(s).
- * Consider including responses to any criticisms raised by more than one reviewer at the beginning of the rebuttal, in a section addressed to all reviewers.
- * The rebuttal should include the reviewer comments in point-by-point format (please note that we provide all reviewers will the reports as they appear at the end of this message).
- * Provide the rebuttal to the reviewer comments and the cover letter as separate files.

We hope that you will be able to resubmit the manuscript within 25 weeks from the receipt of this message. If this is the case, you will be protected against potential scooping. Otherwise, we will be happy to consider a revised manuscript as long as the significance of the work is not compromised by work published elsewhere or accepted for publication at *Nature Biomedical Engineering*.

We hope that you will find the referee reports helpful when revising the work. Please do not hesitate to contact me should you have any questions.

Best wishes,

Filipe

Dr Filipe Almeida
Associate Editor, Nature Biomedical Engineering

Reviewer #1 (Report for the authors (Required)):

In this manuscript, Dr. Mooney and co-authors try to answer the question of durable lymph node (LN) response and its implications on adaptive immunity and vaccine efficacy. They used high-frequency ultrasound imaging to assess the LN responses to 'strong' and 'weak' vaccines, and found cellular expansion was greater and more sustained in MPS-vaccinated mice while enhanced numbers and altered transcriptional features of conventional dendritic cells and inflammatory monocytes were only found in long-term expanded LNs. These results revealed the missing fact that durable LN expansion is an important signal for robust vaccine responses, which has guiding significance for the design of next generation vaccines. However, some writing and experimental details remain to be refined before the publication. I have some comments and suggestions for the authors to consider in revision.

1. Many experimental results are just stated without interpretation and further explanation, which is not easy for readers to understand the meaning of these results. For example,

a) in the second section, what could be inferred from the changes of hyaluronic acid (HA) localization and cellular F-actin signal, or what physiological activities are there two substances usually involved in?

b) in the third section and Supplementary Fig. 7p, it was found that different immune cells reached their peak expansion at different time points, is it related to the time cycle of the immune response (immune cells have

different activation times in the process of immune response because of their different functions)?

2. Sustained release of MPS vaccine is the basis of this study. However, there is completely no characterization on the MPS vaccine. More introduction or characterizations about MPS vaccine should be supplemented.

3. In the fourth section, MPS vaccine without antigen was set as the control group to assess the influence of no sustained antigen presentation. But no antigen presentation is different from no sustained antigen presentation, and results from MPS vaccine without antigen could only reveal that sustained adjuvants release was not able to maintain durable LN expansion. To construct a control group of no sustained antigen presentation, MPS vaccine without antigen should co-administrated with free antigens.

4. The title of the sixth section (Harnessing LN expansion improves vaccine outcomes) seems have some logical mistakes, and so does the title of this manuscript (Durable lymph node expansion strengthens vaccine efficacy). LN expansion is a signal of vaccine responses, but not a designed condition that could be 'harnessed' or a reason for vaccine efficacy. It is the sustained adjuvant release and immune stimulation leads to strengthened vaccine efficacy and durable lymph node expansion but not the other way around.

5. Page 2, second paragraph, "with either vaccine (Fig.1c)" should be (Fig.1d).

6. In the method section "MPS vaccine fabrication", "The three MPS populations (loaded with antigen, CpG, and OVA)", OVA should be "GM-CSF".

Reviewer #2 (Report for the authors (Required)):

In this manuscript, the authors propose that robust and maintained LN expansion strongly correlated with vaccine efficacy, and direct manipulation of LN expansion could generate more potent immunization outcomes. By using novel high-frequency ultrasound (HFUS) technology to visualize superficial internal structures of LNs, they established the magnitude and kinetics of LN response between mesoporous silica (MPS) rod-based vaccine and traditional bolus (liquid) vaccine. Meanwhile, LN expansion was analyzed in comparison to outcomes of immunization, tissue-scale mechanical and matrix properties, cellular composition, and transcriptional signatures. Overall, the manuscript describes some novel findings and it is well-written with high-quality figures. However, this reviewer thinks this manuscript has obvious limitations. The data are not comprehensive to support their claims and the doses of the antigen and adjuvants are way too high. Moreover, it is not convincing by only using MPS vaccines to demonstrate their main points and there is a lack of some deeper mechanism exploration.

Major comments:

1. Although the MPS vaccine formulation was selected as a model of "strong" vaccination that resulted in a persistent LN expansion. But is it representative or universal? Does the most commonly alum adjuvant (or MF-59 adjuvant, AS01...) have this property? What properties do formulations need to induce persistent LN expansion? In addition, the authors should provide more details about MPS (size, TEM...).

2. How accurate is this HFUS device to quantify the volume of lymph nodes? Has there ever been a case like this? How to ensure the reliability of data? Additionally, the authors should show more photographs of explanted LNs at more time points than just day 7 after vaccination.

3. The authors should add the antigen-free and adjuvant-free of MPS control to exclude the influence of the carrier itself against LN expansion. Moreover, the dose of OVA (200 μ g/vaccine), CPG (100 μ g/vaccine), and GM-CSF (1 μ g/vaccine) in mice is way too high. In the "MPS vaccine without antigen" and "Jump-start + bolus vax" studies, whether such a high dose of adjuvants induces significant side effects, such as cytokine storms.

4. In the sections "Cellular activity in the lymph node is regulated by vaccine formulation" and "MPS vaccination alters long-term gene expression of antigen-presenting cells in LNs", after determining the critical cells in the process of LN expansion (such as LN-resident cDC2, cDC1, monocytes, etc.), blocking these cells with specific antibodies to investigate whether similar phenomena still exist will make the conclusion more convincing.

5. The authors should consider introducing more different antigens and designing more reasonable tumor models to verify the universality of “jump-start” strategy. In the “Jump-start + bolus vax” strategy, the optimal timing of bolus vax administration should be demonstrated and discussed.
6. The immunological data provided by the paper should also include the responses of the spleen, and the data of ELISPOT are suggested to fully support their conclusions.
7. All the data about the correlation analysis in this manuscript, R2 is generally low, such as Fig. 2j, Fig. 3b, Fig. 6d, 6e, 6f, etc. Is such a linear or nonlinear relationship still accurate? Please provide a more accurate analysis.
8. In clinical practice, the subunit vaccine is usually administrated in two or more doses, while all data in this paper were obtained on the basis of a single dose. Do the changes in LN expansion after multiple doses show similar characteristics? Please provide relevant data.

Minor issues:

1. In the section “Vaccine formulation alters the magnitude and duration of lymph node expansion”, the data mentioned in the sentence “NdLNs did not change in volume with ether vaccine (Fig. 1c)”, should be Fig. 1d.
2. Fig. 3c, 3e and Fig. 7b, 7c show almost the same data. Please check the full text for similar errors.
3. Please adjust the alphabetical order in Fig.6.

Reviewer #3 (Report for the authors (Required)):

This paper investigates the effect of the MPS vaccine adjuvant system, previously developed by the authors, on the expansion of the draining lymph node. While this study contains an impressive body of work and reports interesting findings, it lacks proper control groups to support solid claims.

More specific comments:

- The authors compare their own MPS formulation to a soluble mixture of GM-CSF/CpG without providing a clear rationale for this choice. Figure 6A suggests that the adjuvant effect of the soluble mixture of OVA+GM-CSF/CpG is very low, as it induces similar IgG2 levels as OVA+PBS. It would be more relevant to compare MPS to established depot systems such as alum and an o/w emulsion, and investigate whether the latter also induce long-term lymph node expansion.
- The authors should also provide longitudinal data on antibody and T-cell responses against OVA or a more relevant antigen, as well as the extent of GC B cell formation in the dLN.
- From an immunological standpoint, I wonder whether the observed long-term response in the lymph nodes is actually desirable. Could prolonged innate activation in the dLN stimulate T-cell anergy, and could excessive upregulation of immune checkpoints such as PD-L1 be a concern? Thsi should be addressed by new experimental data.
- From a translational standpoint, is there any risk of patient discomfort?

Reviewer #4 (Report for the authors (Required)):

Najibi et al. demonstrate that a formerly described mesoporous silica (MPS) rod-based vaccine induces a more potent and persistent LN expansion than standard immunization with the same antigen and adjuvants in liquid form, termed “bolus” vaccine. They also describe changes in the mechanics and cellularity of the LN that accompany the expansion. The authors then use the comparison between the two vaccine formulations to examine the impact of LN expansion on the quality and duration of the induced immune response. Data is presented that demonstrates that conventional dendritic cells and inflammatory monocytes were mainly

found in long-term expanded LN. These factors correlated with an increased adaptive immune response and enhanced magnitude and efficacy of a therapeutic cancer vaccination.

The approach used by the authors allows to investigate how LN expansion is important for immune responses and efficacy of vaccines, as they compare different formulations of the same vaccine, with no change in antigen, adjuvant, dose or application route. The data presented is very extensive and convincing, with state-of-the-art techniques to characterize in depth the changes in LN morphology and function and the impact on immune cell phenotype and function. Although LN expansion has been well characterized, including factors responsible for the expansion, the authors show novel findings with respect to how LN expansion may contribute to vaccine efficacy. This may have a therapeutic impact on vaccine development.

Minor comments:

1. Page 2: Fig1c does not show ndLN, please correct in text.
2. To compensate for differences in LN size between individual mice, the authors could consider expressing the LN size as a percentage of the non-draining lymph nodes.
3. In fig 2 the authors show several parameters relating to the mechanics of the LN (actin, collagen and HA). However, the text does not mention what are the implied consequences of these changes, in particular with respect to immune cell function and migration. Please explain.
4. In fig 3H/I/J the authors show that they can distinguish 9 cellular populations using single cell RNAseq. However, the more important conclusion of their RNAseq follows in SI 9c, where the authors show that after MPS vaccination cell populations change in comparison to a bolus vaccination. I suggest to include SI 9c in the main figure.
5. In figure 4 the authors discuss the change in DC subpopulations but then quickly move to monocytes. The change between cell types is unexpected and lacks explanation in the text.
6. In figure 6D/E/F the authors show a correlation between LN expansion and different immunological effects. As they are comparing very different subgroups, this method of correlation can suggest a false relationship. (see for example: <https://www.ncbi.nlm.nih.gov/pmc/articles/PMC5079093/>)

Mon 16 Oct 2023

Decision on Article nBME-23-0221A

Dear Dr Mooney,

Thank you for your revised manuscript, "Durable lymph node expansion signals vaccine efficacy". Having consulted with the original reviewers (whose comments you will find at the end of this message), I am pleased to write that we shall be happy to publish the manuscript in *Nature Biomedical Engineering*.

We will be performing detailed checks on your manuscript, and in due course will send you a checklist detailing our editorial and formatting requirements. You will need to follow these instructions before you upload the final manuscript files.

Best wishes,

Filipe

Dr Filipe Almeida
Associate Editor, Nature Biomedical Engineering

Reviewer #1 (Report for the authors (Required)):

In the revision, the authors responded well to the questions in the comments. Specifically, the main improvements including:

- 1, The morphological characterization of MPS vaccines as well as their adjuvant sustained-releasing results were complemented.
- 2, The control group of free antigens combined with MPS was supplemented in the fourth section.
- 3, Some specific analysis of the experimental results was supplemented.
- 4, The logical error of the article title had been corrected.
- 5, Previous writing errors had been corrected.

While for the comments from other reviewers, the main improvements in the revision including:

- 1, Supplemented experiments showed that MPS vaccine could induce stronger LN expansion than other commonly used adjuvants, thus demonstrating that MPS vaccine can indeed be used as a "strong" vaccine.
- 2, Immune cell depletion experiments were supplemented and indicated the stimulatory and antigen-presenting role of inflammatory monocytes, which associating with macrophages may be required for sustained LN expansion.
- 3, Supplemented experiments indicated that the spleen was less affected by MPS vaccine.
- 4, Optimizing the discussions of data trend changes to make them more reasonable

There are no problems with the supplementary experimental data and the text description in the revision, and there are no further comments to this article.

Reviewer #2 (Report for the authors (Required)):

In this revised version, the authors have addressed my previous comments point by point. They have supplemented some details and they have done new experiments and analysis to better support their conclusion.

Reviewer #3 (Report for the authors (Required)):

I am satisfied with the revisions implemented by the authors. They did a substantial effort to present new experimental data, additional figures and textual changes in response to my comments. Congratulations with this interesting work!

Reviewer #4 (Report for the authors (Required)):

Thank you for this detailed revision. All my comments have been addressed.

Rebuttal 1

Point-by-point responses to reviewers' comments (nBME-23-0221).

We thank the reviewers for their insightful and constructive comments on the manuscript. Our response to each comment is included below, with reviewer comments in bold typeface and the replies in regular typeface.

Reviewer #1 (Report for the authors):

In this manuscript, Dr. Mooney and co-authors try to answer the question of durable lymph node (LN) response and its implications on adaptive immunity and vaccine efficacy. They used high-frequency ultrasound imaging to assess the LN responses to 'strong' and 'weak' vaccines, and found cellular expansion was greater and more sustained in MPS-vaccinated mice while enhanced numbers and altered transcriptional features of conventional dendritic cells and inflammatory monocytes were only found in long-term expanded LNs. These results revealed the missing fact that durable LN expansion is an important signal for robust vaccine responses, which has guiding significance for the design of next generation vaccines. However, some writing and experimental details remain to be refined before the publication. I have some comments and suggestions for the authors to consider in revision.

We thank the reviewer for their evaluation of the manuscript and for their comments, which reflect the key elements of the study. We have responded to specific comments below.

1. Many experimental results are just stated without interpretation and further explanation, which is not easy for readers to understand the meaning of these results. For example,

We appreciate the reviewer's suggestion and have now added additional text to the results section to summarize our key findings, supporting interpretation and guiding readers through the manuscript. Examples are provided below.

Taken together, these results suggest that LN tissue encompasses a range of mechanical properties, dependent on location within the node, and these parameters change as LNs expand.

This indicates that long-term antigen presentation at the vaccine site is important for sustained LN expansion.

The involvement of mechanosensing genes and *Ltb*, involved in lymphoid organogenesis, suggests that cDC2s may both respond and contribute to the changing LN microenvironment during expansion.

These gene alterations position inflammatory monocytes as a potential inflammatory, antigen-presenting cell type during sustained LN expansion.

Inflammatory monocyte responses were abrogated at later timepoints when the MPS vaccine was delivered without antigen, equivalent to the PBS or bolus controls by day 20, suggesting a relationship between long-lived antigen presentation, LN expansion, and monocyte responses (Fig. 5b).

These results further suggest a role of inflammatory monocytes in effector CD8⁺ T cell responses to MPS vaccination, potentially through direct antigen presentation or inflammatory stimulation.

Taken together, these data indicate a stimulatory and antigen-presenting role of inflammatory monocytes, and that these cells in association with macrophages may be required for sustained LN expansion.

In summary, jump-starting LN expansion prior to vaccine administration associated with improved T cell responses and antitumor efficacy in a model antigen tumor model.

a) in the second section, what could be inferred from the changes of hyaluronic acid (HA) localization and cellular F-actin signal, or what physiological activities are there two substances usually involved in?

We have now added additional text in the both results and discussion sections to interpret the HA and F-actin signal in the context of our broader data, provided below.

Results section: These changes suggest that LN expansion may be accompanied by changes in tissue mechanical properties, as both HA and F-actin are involved in cellular mechanotransduction and signaling pathways.

Discussion section: HA provides both tissue scaffolding and cell signaling cues, and is regularly recycled through LNs. Its increased quantity in the LN periphery after vaccination may suggest altered recycling dynamics and could suggest a potential role in inflammatory signaling during the vaccine response. F-actin plays a key role in cytoskeletal maintenance and cell motility, and the observed pattern of F-actin localization indicates that the tissue-level mechanical changes in LNs may translate down to individual cell behavior, such as immune cell migration in the expanded node.

b) in the third section and Supplementary Fig. 7p, it was found that different immune cells reached their peak expansion at different time points, is it related to the time cycle of the immune response (immune cells have different activation times in the process of immune response because of their different functions)?

We have now more carefully explained our experimental results in the context of previously understood immune cell response kinetics, as reproduced below.

Results section: These results indicate that vaccine-induced LN expansion engages the temporal dynamics of a pathogen-induced immune response, with innate immune cells rapidly responding followed by lymphocytes at later times.

Discussion section: The patterns of LN cellular responses observed here are consistent with prior immunological characterizations^{2,7,9}. For example, immune cell numbers peaked earlier than stromal cells (day 4-7 versus day 11). Immune cells themselves responded with the natural progression of infection response, with innate/myeloid cells responding most rapidly followed by adaptive immune cells.

2. Sustained release of MPS vaccine is the basis of this study. However, there is completely no characterization on the MPS vaccine. More introduction or characterizations about MPS vaccine should be supplemented.

We have now included data from experiments characterizing the MPS vaccine. Notably, we performed experiments to characterize (1) MPS particle size and structure (surface area, pore volume, pore size, and SEM imaging) and (2) release kinetics of vaccine components (e.g. CpG and GM-CSF) from the MPS material. In addition, we have expanded the text introducing the MPS vaccine in the context of

prior work characterizing this vaccine formulation. The new data have been incorporated into the manuscript in the form of a new Supplementary figure and additional text, included below.

Methods: MPS particle morphology was assessed using brightfield microscopy and scanning electron microscopy, and surface area, pore size, and pore volume were measured using N₂ adsorption/desorption isotherms ¹⁸.

Results: Mesoporous silica (MPS) rod-based vaccines, previously found to elicit strong cellular and humoral responses against diverse antigen targets ^{15–18} compared to a traditional bolus (liquid) vaccine, were explored. These high-aspect ratio, silica-based nanoparticles can adsorb vaccine antigens and adjuvants for sustained release, and form a three-dimensional scaffold promoting antigen-presenting cell recruitment in mouse models. MPS vaccines previously induced potent and long-lived germinal center responses dependent on sustained antigen release from the vaccine site. Here, MPS rods used in vaccine formulation had an average length of 85.9µm and released vaccine components CpG and GM-CSF in a sustained manner (**Supplementary Fig. 1a-e**).

Supplementary Fig. 1. MPS vaccine characterization. (a) Scanning electron microscopy imaging of MPS particles. (b) MPS particle size distribution. (c) Size and volume characterization of MPS particles. *In vitro* release curves of CpG (d) and GM-CSF (e) from MPS vaccines.

3. In the fourth section, MPS vaccine without antigen was set as the control group to assess the influence of no sustained antigen presentation. But no antigen presentation is different from no sustained antigen presentation, and results from MPS vaccine without antigen could only reveal

that sustained adjuvants release was not able to maintain durable LN expansion. To construct a control group of no sustained antigen presentation, MPS vaccine without antigen should co-administrated with free antigens.

To decouple the variables of antigen presence and antigen release rate, we have now performed a new experiment in which the MPS vaccine without antigen was co-administered with free antigen. This was achieved through a separate injection of free antigen not loaded into the MPS, although draining to the same lymph node. We observed that draining LNs in the group with bolus antigen both did not enlarge to the same extent as the full vaccine, and began decreasing in size more quickly. This pattern was, interestingly, similar to the MPS vaccine without antigen, suggesting that sustained antigen release contributes to sustained LN volume expansion. These results are included in a new Supplementary Figure, and we have added additional text to our methods, results, and discussion sections considering this new data.

Methods:

For bolus antigen injection experiments, the MPS vaccine containing CpG and GM-CSF alone was followed by injection of OVA antigen alone adjacent to the vaccine site.

Results: Injecting the antigen separately as a bolus (i.e. not delivered from the MPS scaffold) similarly reduced the degree and duration of expansion, indicating a critical role of sustained antigen presentation (**Supplementary Fig. 12b**).

Supplementary Figure 12. Comparing lymph node expansion with MPS and alternative vaccine formulations. Mice were immunized with MPS vaccines (GM-CSF, CpG, and OVA protein) or MPS vaccines with bolus antigen (GM-CSF and CpG loaded into MPS, with OVA injected separately as a bolus). (b) Vaccine draining LN fold expansion over time, comparing the full MPS vaccine (antigen loaded in MPS material) to an MPS vaccine with a separate bolus injection of antigen. n = 4 (full MPS) or 5 (bolus antigen) biologically independent animals per group.

Discussion:

The prolonged LN expansion observed here likely derives from persistence of the MPS vaccine and its sustained release of antigen and adjuvant¹⁵. Here, MPS vaccines without sustained antigen presentation, either antigen-free or with antigen rapidly released as a bolus, did not maintain LN expansion relative to when antigen was released from MPS.

4. The title of the sixth section (Harnessing LN expansion improves vaccine outcomes) seems have some logical mistakes, and so does the title of this manuscript (Durable lymph node

expansion strengthens vaccine efficacy). LN expansion is a signal of vaccine responses, but not a designed condition that could be ‘harnessed’ or a reason for vaccine efficacy. It is the sustained adjuvant release and immune stimulation leads to strengthened vaccine efficacy and durable lymph node expansion but not the other way around.

To clarify our meaning, we have modified the language in the titles to this section (“LN expansion signals vaccine outcomes”) and to our manuscript (“Durable lymph node expansion signals vaccine efficacy”). We believe these changes more accurately reflect the data presented and specify the focus of our work.

5. Page 2, second paragraph, “with either vaccine (Fig.1c)” should be (Fig.1d).

We have now corrected our text to reflect the appropriate figure (Fig. 1d).

6. In the method section “MPS vaccine fabrication”, “The three MPS populations (loaded with antigen, CpG, and OVA)”, OVA should be “GM-CSF”.

We have now corrected our text to indicate the proper groups: “The three MPS populations (loaded with antigen, CpG, and GM-CSF) were combined...”

Reviewer #2 (Report for the authors (Required)):

In this manuscript, the authors propose that robust and maintained LN expansion strongly correlated with vaccine efficacy, and direct manipulation of LN expansion could generate more potent immunization outcomes. By using novel high-frequency ultrasound (HFUS) technology to visualize superficial internal structures of LNs, they established the magnitude and kinetics of LN response between mesoporous silica (MPS) rod-based vaccine and traditional bolus (liquid) vaccine. Meanwhile, LN expansion was analyzed in comparison to outcomes of

immunization, tissue-scale mechanical and matrix properties, cellular composition, and transcriptional signatures. Overall, the manuscript describes some novel findings and it is well-written with high-quality figures. However, this reviewer thinks this manuscript has obvious limitations. The data are not comprehensive to support their claims and the doses of the antigen and adjuvants are way too high. Moreover, it is not convincing by only using MPS vaccines to demonstrate their main points and there is a lack of some deeper mechanism exploration.

We thank the reviewer for their careful evaluation of the manuscript and for their specific comments, to which we have responded below. We have now collected additional data to support our claims, and modified our text and language with intention to more accurately reflect the data depicted and support its interpretation. We believe these changes encouraged by the reviewer's comments have strengthened our message and the impact of our manuscript.

Major comments:

1. Although the MPS vaccine formulation was selected as a model of “strong” vaccination that resulted in a persistent LN expansion. But is it representative or universal? Does the most commonly alum adjuvant (or MF-59 adjuvant, AS01...) have this property? What properties do formulations need to induce persistent LN expansion? In addition, the authors should provide more details about MPS (size, TEM...).

We have now conducted additional experiments comparing the MPS vaccine to more commonly used formulations (Alum and emulsion-based vaccines) and an alternative controlled-release vaccine formulation (a polymer-based cryogel explored in our group) to contextualize these results. Interestingly, the MPS vaccine led to a significantly higher LN expansion than these other formulations, lending support to our selection of this system as a ‘strong’ vaccine with which to explore LN expansion. These data are represented in new figures and text in the methods, results, and discussion sections. Furthermore, we have devoted a portion of our discussion section to address the reviewer's specific questions regarding which properties are shared among vaccine formulations leading to persistent LN expansion.

Methods:

Alternative vaccine formulations. Alum and emulsion-based vaccines were formulated following manufacturer protocols. Briefly, alum-based vaccines were prepared by mixing 2% (20mg/mL) Alhydrogel adjuvant (Invivogen) in a 1:1 volume ratio with 200µg OVA protein antigen to reach a total volume of 100µL per vaccine dose. The MF59 oil-in-water emulsion vaccine was prepared by mixing AddaVax (Invivogen) in a 1:1 volume ratio with 200µg OVA protein antigen to reach a total volume of 100µL per vaccine dose. Both vaccines were injected *s.c.* through an insulin syringe.

Results:

While other published depot-based vaccine formulations including alum, MF59 emulsion, and cryogel-based scaffolds also induced LN expansion, expansion was notably lower than with the MPS vaccine (**Supplementary Fig. 3c**).

Supplementary Figure 3. Comparing lymph node expansion with MPS and alternative vaccine formulations. Mice were immunized with MPS vaccines (GM-CSF, CpG, and OVA protein). Additional mice were immunized with the same quantity of OVA antigen in alum, emulsion (MF59), or alginate cryogel-based vaccines (cryogel vaccines also contained GM-CSF and CpG in equivalent amounts to the MPS vaccine). Vaccine-draining LNs were longitudinally imaged using high frequency ultrasound. (c) Vaccine draining LN fold expansion comparing MPS, alum, MF59 emulsion, and cryogel formulations. Statistical analysis was performed using a one-way ANOVA with Dunnett's post hoc test (t= 7, 11, 21 days) or a Kruskal-Wallis test with Dunn's post hoc test (t= 4 days). P values for the comparison between MPS and other vaccine systems are displayed on the plot. n = 4 (full MPS) or 5 (other groups) biologically independent animals per group.

Discussion:

The MPS vaccine used here as a model of robust, persistent vaccination elicited a ~7-fold increase in dLN volume that was maintained for weeks. This is a striking outcome compared to reported vaccine formulations (~4-5-fold increase in mass or size, and contracting from day 7-14)^{2,29-34}. In a recent study, popliteal LNs expanded ~10-fold to vaccination with complete Freund's adjuvant, although this response was not explored beyond 14 days¹⁰. Although clinically successful vaccines such as those against SARS-CoV-2 have induced lymphadenopathy in a subset of patients, presenting potential discomfort, these responses largely resolved naturally, and could point to a productive adaptive immune response^{16,17,35}. In our experiments, the MPS vaccine promoted significantly greater LN expansion than commonly used vaccine delivery methods including Alum and MF59, an emulsion-based vaccine. The MPS vaccine also drove stronger LN enlargement than cryogel-based scaffold vaccines fabricated from alginate, an interesting result given the identical vaccine components in these two vaccines, suggesting that the selection of biomaterial itself can impact immune outcomes. This result is not unexpected given that silica can activate inflammasome activity, while alginate is widely used for its biocompatibility and lack of immunogenicity^{36,37}. The prolonged LN expansion observed here likely derives from persistence of the MPS vaccine and its sustained release of antigen and adjuvant¹⁸. Here, MPS vaccines without sustained antigen presentation, either antigen-free or with antigen rapidly released as a bolus, did not maintain LN expansion relative to when antigen was released in a sustained manner from MPS.

In addition, we have now included data from experiments characterizing the MPS vaccine. Notably, we performed experiments to characterize (1) MPS particle size and structure (surface area, pore volume, pore size, and SEM imaging) and (2) release kinetics of vaccine components (e.g. CpG and GM-CSF) from the MPS material. In addition, we have expanded the text introducing the MPS vaccine in the context of prior work characterizing this vaccine formulation. The new data have been

incorporated into the manuscript in the form of a new Supplementary figure and additional text, included below.

Methods: MPS particle morphology was assessed using brightfield microscopy and scanning electron microscopy, and surface area, pore size, and pore volume were measured using N₂ adsorption/desorption isotherms ¹⁸.

Results: Mesoporous silica (MPS) rod-based vaccines, previously found to elicit strong cellular and humoral responses against diverse antigen targets ^{15–18} compared to a traditional bolus (liquid) vaccine, were explored. These high-aspect ratio, silica-based nanoparticles can adsorb vaccine antigens and adjuvants for sustained release, and form a three-dimensional scaffold promoting antigen-presenting cell recruitment in mouse models. MPS vaccines previously induced potent and long-lived germinal center responses dependent on sustained antigen release from the vaccine site. Here, MPS rods used in vaccine formulation had an average length of 85.9µm and released vaccine components CpG and GM-CSF in a sustained manner (**Supplementary Fig. 1a-e**).

Supplementary Fig. 1. MPS vaccine characterization. (a) Scanning electron microscopy imaging of MPS particles. (b) MPS particle size distribution. (c) Size and volume characterization of MPS particles. *In vitro* release curves of CpG (d) and GM-CSF (e) from MPS vaccines.

2. How accurate is this HFUS device to quantify the volume of lymph nodes? Has there ever been a case like this? How to ensure the reliability of data? Additionally, the authors should show more photographs of explanted LNs at more time points than just day 7 after vaccination.

The Vevo ultrasound device operated with a high-frequency (50 MHz) transducer enables axial resolution down to 20 μ m, sufficient to visualize and quantify bulk LN properties (on the order of mm in each dimension). To our knowledge, this is the first reported use of HFUS for longitudinal measurement of murine LNs; however, HFUS has previously been used to image and assess human LNs with diagnostic capacity [Mehta, N. *et al. Clinical Imaging* 2021 <https://doi.org/10.1016/j.clinimag.2021.01.016>; Heaven, C.L. *et al. ANZ J Surg* 2022 <https://doi.org/10.1111/ans.17808>; Ahuja, A., *et al. American Journal of Roentgenology* 2005, doi: 10.2214/ajr.184.5.01841691]. We have amended our introduction of this technique to clarify its resolution and prior use in human LN imaging.

Introduction: Vaccine-draining LNs (dLNs) were imaged longitudinally using high frequency ultrasound (HFUS), a simple, noninvasive technique for visualizing bulk properties of superficial internal structures with high (~20 μ m) axial resolution, to establish the magnitude and kinetics of LN response¹⁴. HFUS has previously been employed to image and assess human LNs with potential diagnostic capacity¹⁵⁻¹⁷.

The reliability of HFUS measurement was established by early investigative studies with two researchers imaging the same mouse LN, and the same researcher imaging the same mouse LN on separate days, achieving consistent volume measurements. Furthermore, our longitudinal imaging of naïve mouse LNs over 100 days demonstrated a reliable baseline for imaging without notable fluctuations. We now include text in our Methods section describing this process to ensure data reliability.

Methods: To check reliability of data, the same mouse LNs were imaged with two researchers on the same day, and the same researcher on different days. This resulted in consistent volume measurements, indicating this a replicable technique.

We now include additional photographs of explanted LNs at an additional timepoint (day 20) following vaccination, showcasing the continued expansion in the MPS group. A major benefit of noninvasive HFUS imaging is to avoid euthanizing mice for organ explant, and so we attempted to minimize repetitive euthanasia for organ collection whenever possible in our studies. This imaging is now part of a Supplementary figure and referenced in the text.

Results: While PBS injection did not affect LN volume, both vaccine formulations induced LN expansion, but with markedly different durability (**Fig. 1a,b, Supplementary Fig. 2a-c**).

Supplementary Figure 2. MPS vaccination induces robust, prolonged LN expansion. (c) Photograph of LNs explanted 20 days after vaccination. Each LN was derived from a unique mouse.

3. The authors should add the antigen-free and adjuvant-free of MPS control to exclude the influence of the carrier itself against LN expansion. Moreover, the dose of OVA (200µg/vaccine), CPG (100µg/vaccine), and GM-CSF (1µg/vaccine) in mice is way too high. In the “MPS vaccine without antigen” and “Jump-start + bolus vax” studies, whether such a high dose of adjuvants induces significant side effects, such as cytokine storms.

We have now conducted additional experiment assessing the impact of different vaccine components on LN expansion. Notably, we compared the full MPS vaccine to adjuvant-free controls including an MPS groups without either CpG or GM-CSF. We observed that LNs of mice treated with the CpG or GM-CSF-free vaccines expanded to a significantly lesser extent than those of mice treated with the full vaccine. These data are now included in a new Supplementary figure and discussed in the manuscript text.

Methods:

Control MPS vaccines lacking either antigen or adjuvant were prepared using the same total amount of MPS material (5mg) with either antigen, CpG, or GM-CSF excluded from the vaccine combination.

Results:

The removal of either CpG or GM-CSF from the vaccine formulation diminished the magnitude of dLN expansion (Supplementary Fig. 3a).

Supplementary Figure 3. Comparing lymph node expansion with MPS and alternative vaccine formulations. Mice were immunized with MPS vaccines (GM-CSF, CpG, and OVA protein), MPS vaccines without CpG (GM-CSF and OVA), MPS vaccine without GM-CSF (CpG and OVA). (a) Quantification of vaccine draining LN fold expansion over time, comparing the full MPS vaccine with formulations lacking either CpG or GM-CSF. Statistical analysis was performed using a one-way ANOVA with Dunnett’s post hoc test. $n = 4$ (full MPS) or 5 (other groups) biologically independent animals per group.

Discussion: The prolonged LN expansion observed here likely derives from persistence of the MPS vaccine and its sustained release of antigen and adjuvant¹⁵. Here, MPS vaccines without sustained antigen presentation, either antigen-free or with antigen rapidly released as a bolus, did not maintain LN expansion relative to when antigen was released from MPS. Removing either CpG or GM-CSF from the vaccine also diminished LN expansion.

In addition, we have now investigated MPS vaccines delivering an order-of-magnitude lower dose of antigen and adjuvant. In the literature, model vaccines typically deliver 10-50µg CpG [1-3], although other vaccine studies have tested as low as 2.5µg [4] or up to 400µg [5]. OVA doses range can range up to 500µg [2, 6-9]. Our initial selection of 100µg CpG and 200g OVA was based on previous biomaterial scaffold-based vaccine development [10]. In a new experiment, we have now tested MPS vaccines delivering 20µg OVA and 20µg CpG. Our selection of 1µg GM-CSF was not altered, as it was based on prior studies in our group determining the optimal amount of GM-CSF to support antigen-presenting cell recruitment and expansion in scaffold vaccines [10]. The new data indicate that the low-dose MPS vaccine also promoted robust and maintained LN expansion, although with a different expansion kinetic, peaking at a later timepoint. This experiment was repeated to confirm this result. These results suggest that LN expansion is mediated by the MPS vaccine at a range of doses and formulations, and these results are now included in our text and a Supplementary Figure.

[1] Nanishi, E., *et al. Sci. Transl. Med.* 2021, doi: 10.1126/scitranslmed.abj5305
 [2] Moser, B.A., *et al. Science Advances* 2020, doi: 10.1126/sciadv.aaz8700
 [3] Lin, Y., *et al. Scientific Reports* 2018, <https://doi.org/10.1038/s41598-018-28281-5>
 [4] Ben-Akiva, E., *et al. PNAS* 2023, <https://doi.org/10.1073/pnas.2301606120>
 [5] Bode, C., *et al. Expert Rev Vaccines* 2011, doi: 10.1586/erv.10.174
 [6] Kim, KS, *et al. Advanced Functional Materials* 2016, doi: 10.1002/adfm.201504879
 [7] Garulli, B., *et al. Clinical and Vaccine Immunology* 2008, doi: <https://doi.org/10.1128/CVI.00166-08>
 [8] Mittal, A., *et al. Nanomedicine* 2015, doi: 10.1016/j.nano.2014.08.009
 [9] Moser, B.A., *et al. Science Advances* 2020, doi: 10.1126/sciadv.aaz8700
 [10] Kim, J., *et al. Nat. Biotech* 2015 doi: 10.1038/nbt.3071

Methods:

A low-dose MPS vaccine was prepared using 20µg OVA, 20µg CpG, and 1µg GM-CSF delivered in the same 5mg MPS material per vaccine.

Results:

An MPS vaccine with 10-fold lower doses of OVA and CpG also induced long-term LN expansion (Supplementary Fig. 3b).

Supplementary Figure 3. Comparing lymph node expansion with MPS and alternative vaccine formulations. Mice were immunized with MPS vaccines (GM-CSF, CpG, and OVA protein). A final group was dosed with an MPS vaccine containing a log-fold lower dose each of CpG and OVA.

Vaccine-draining LNs were longitudinally imaged using high frequency ultrasound. (b) Vaccine draining LN fold expansion over time, comparing the full MPS vaccine with a low-dose MPS vaccine. n = 4 (full MPS) or 10 (low-dose MPS) biologically independent animals per group. The low-dose MPS group displays the combined results of two separate experiments.

Regarding the safety of high-dose vaccine formulations, the “MPS vaccine without antigen” and “jump-start plus bolus vax” groups were repeated multiple times throughout our studies, and no signs of toxicity were observed. Mice were monitored throughout these studies and displayed no changes in weight or other safety indications. Importantly, despite the selection of high OVA, CpG, and GM-CSF doses, these components are largely maintained within the MPS vaccine and released slowly. We have now included additional release data in our MPS characterization (included in response to an above reviewer comment) to demonstrate that a benefit of the scaffold-based delivery method is that it can reduce the burst release of a single bolus dose.

In addition, we have now added new data to the manuscript tracking MPS-vaccinated mice over 16 months, during which no long-term changes in mouse weight, behavior, or spleen or LN cellularity were observed, suggesting no long-term detrimental impacts on immune compartments. Furthermore, we have measured HMGB-1 levels in serum 21 days after vaccination. HMGB-1 in serum can indicate inflammation or cellular damage and has been associated with cytokine release syndrome [Yang, H. & Tracey, K.J. *Biochimica et Biophysica Acta* 2010, doi:10.1016/j.bbagr.2009.11.019; Wulandari S., *et al. Immunology* 2023, doi: 10.1111/imm.13623]. Serum HMGB-1 levels did not significantly differ between MPS or PBS-injected mice. These data are included in our text and in the form of new Supplementary Figures, and reproduced below.

Results: In MPS-vaccinated mice, serum HMGB-1 levels, indicative of inflammatory cytokine responses and/or cellular death^{32,33}, were comparable to PBS controls (**Supplementary Fig. 25g**). Mice tracked out to 485 days after MPS vaccination did not display changes in weight, LN or spleen cell counts, or proportions of immune cell subsets in blood or secondary lymphoid organs, although elevated OVA-specific CD8+ T cells remained detectable in all immune compartments investigated (**Supplementary Fig. 26a-i**).

Supplementary Figure 25. Long-lived vaccine responses and LN expansion. (g) HMGB-1 concentration in mouse serum 21 days after vaccination.

Supplementary Figure 26. Long-term response to MPS vaccination. Mice were injected on day 0 with PBS or an MPS vaccine consisting of GM-CSF, CpG, and OVA protein. 485 days later, mice were euthanized and blood, LNs, and spleens collected for analysis. (a) Mouse weight at 485 days. Total cell counts in draining LN (b) and spleen (c). (d) OVA-tetramer binding proportion of CD8⁺ T cells. Statistical analysis was performed using a two-tailed t test. Proportions of (e) T cells (CD19⁻CD3⁺), (f) B cells (CD19⁺CD3⁻), (g) DCs (CD19⁻CD3⁻Ly6C⁺F4/80⁻CD11c⁺MHCII⁺), (h) monocytes (CD19⁻CD3⁻CD11b⁺Ly6C⁺), and (i) macrophages (CD19⁻CD3⁻Ly6C⁺CD11b⁺F4/80⁺) in the blood, LNs and spleens. n = 4 (MPS) or 5 (PBS) biologically independent animals per group; means depicted; error bars, s.d.

4. In the sections “Cellular activity in the lymph node is regulated by vaccine formulation” and “MPS vaccination alters long-term gene expression of antigen-presenting cells in LNs”, after determining the critical cells in the process of LN expansion (such as LN-resident cDC2, cDC1, monocytes, etc.), blocking these cells with specific antibodies to investigate whether similar phenomena still exist will make the conclusion more convincing.

To test the contributions of distinct immune cell types to LN expansion and the vaccine response, we have now conducted experiments utilizing specific depleting reagents. From our single-cell analysis, inflammatory monocytes were highlighted as a key antigen-presenting cell type engaged by MPS vaccination. The anti-CCR2 MC-21 antibody was used to deplete inflammatory monocytes. This antibody is functional for ~5-6 days in mouse models until host immune responses restrict depletion, so short dosing regimes were selected in either the early or late periods of LN expansion. In addition, neutrophils (anti-Ly6G antibody), macrophages (clodronate liposomes), and lymphocytes (anti-CD4, anti-CD8, and anti-B220 antibodies) were depleted and the impact on LN expansion was assessed.

Depletion of inflammatory monocytes impacted effector and antigen-specific CD8⁺ T cell responses to vaccination, in a manner dependent on the timing of the depletion; only depleting monocytes in the early LN expansion phase had this effect. Notably, the depletion of any single immune cell subset did not affect the magnitude or kinetics of LN expansion assayed through ultrasound. However, the combined depletion of Ly6C^{hi} monocytes and macrophages reduced the duration of LN expansion, suggesting a role of both of these antigen-presenting cell types in the maintenance of LN enlargement. These data have now been incorporated into the manuscript in the methods, results, and discussion sections, and in both additions to main figures and new supplementary figures. These changes are included below.

Methods:

Immune cell depletion experiments. Mice were administered MPS vaccines (containing GM-CSF, CpG, OVA) and treated with immune cell-depleting reagents. To deplete inflammatory monocytes, 20µg MC-21 antibody was injected *i.p.* daily for 5 total doses beginning 0 (days 0-4), 1 (days 1-5), or 10 (days 10-14) days after vaccination. To deplete macrophages, 200µL of clodronate liposome solution (Liposoma) was injected *i.p.* on days 0, 3, and 5 (early depletion) or 9, 12, 15, and 18 (late depletion) after vaccination. To deplete lymphocytes, a mixture of 250µg each of anti-CD4, anti-CD8, and anti-B220 antibodies (Invivogen) was injected *i.p.* on days 9, 12, 15, and 18 after vaccination. To deplete neutrophils, 25µg anti-Ly6G antibody was injected *i.p.* every day from day 9 to 15 and then increased to 50µg from days 16-20 after vaccination. 50µg anti-rat kappa Ig (Invivogen) was injected *i.p.* every other day from day 9 to day 19 to combat host anti-antibody responses and extend neutrophil depletion. Peripheral blood and/or immune organs were collected to verify depletions.

Results:

Inflammatory monocyte depletion alters vaccine response

To further discern the impact of inflammatory monocytes on lymph node expansion and vaccine response, specific depleting reagents were next employed. MPS-vaccinated mice were treated with the CCR2-targeting MC-21 mAb²⁹⁻³¹ either early (days 1-5, LN expansion phase) or later (days 10-14, LN maintenance phase) after immunization (**Fig. 5e**). MC-21 mAb effectively depleted Ly6C^{hi} monocytes in the blood, LN, and scaffold during the treatment course, although numbers in the blood rebounded within days (**Supplementary Fig. 20a-c, Fig. 5f**). Early depletion of Ly6C^{hi} monocytes delayed the effector CD8⁺ T cell response to vaccination, which peaked later, after monocytes had been restored, relative to the MPS vaccine group (**Fig. 5g**). Furthermore, only the MPS-vaccinated group treated early

with MC-21 antibody had significantly elevated tetramer-specific CD8⁺ T cells by day 20, after the monocyte rebound, compared to the PBS controls (**Fig. 5h**). Administration of MC-21 mAb in the later phase of the LN response (days 10-14) had no discernible impact on the T cell response. These results further suggest a role of inflammatory monocytes in effector CD8⁺ T cell responses to MPS vaccination, potentially through direct antigen presentation or inflammatory stimulation.

LN expansion kinetics in the absence of inflammatory monocytes or other immune cell subsets were next assessed. MC-21 mAb and/or clodronate liposomes were used to deplete Ly6C^{hi} monocytes and macrophages, respectively (**Supplementary Fig. 21a**). Lymphocyte (anti-CD4, CD8, and B220) and neutrophil (anti-Ly6G) antibodies were also tested. No differences were observed in the magnitude or kinetics of LN expansion with depletion of any immune cell subset alone (**Supplementary Fig. 21b-g**). However, depleting both inflammatory monocytes and macrophages together restrained the maintenance of LN expansion (**Supplementary Fig. 21h**). Taken together, these data indicate a stimulatory and antigen-presenting role of inflammatory monocytes, and that these cells in association with macrophages may be required for sustained LN expansion.

Supplementary Figure 20. Inflammatory monocyte depletion. Mice were administered MPS vaccines (containing GM-CSF, CpG, OVA) or PBS. One group of MPS-vaccinated mice was treated with MC-21 CCR2-depleting mAb daily from days 1-5 (“MC-21 expansion”) and one group was treated daily from days 10-14 (“MC-21 maintenance”), according to the timeline in Fig. 5f. Peripheral blood was collected on days 6, 8, 14, and 20 for cellular analysis. (a) Representative flow cytometry plots depicting CD3⁺ B220⁻ cells in blood on day 14. At this time, the Ly6C^{hi} monocyte population was depleted in the MPS/MC-21 maintenance group (depicted). Numbers of Ly6C^{hi} monocytes in LNs (b) and MPS scaffolds (c) on day 5 of mice given MPS vaccines and treated with or without MC-21 daily from days 0-4. For b, statistical analysis was performed using analysis of variance (ANOVA) with Tukey’s post hoc test. For c, statistical analysis was performed using a Mann-Whitney test. For b-c, means depicted; error bars, s.d.; n = 5 biologically independent animals per group.

Figure 5. Prolonged LN expansion is associated with inflammatory monocyte engagement. (e-h) Mice were administered MPS vaccines (containing GM-CSF, CpG, OVA) or PBS. One group of MPS-vaccinated mice was treated with MC-21 CCR2-depleting mAb daily from days 1-5 (“MC-21 expansion”) and one group was treated daily from days 10-14 (“MC-21 maintenance”). Peripheral blood was collected on days 6, 8, 14, and 20 for cellular analysis and scaffold size was measured using calipers. $n = 5$ biologically independent animals per group. (e) Experimental timeline and conditions. (f) Inflammatory monocyte proportion in blood over time. Differences between groups are statistically significant at day 6 (MPS/MC21 expansion vs. MPS, $P = 0.005$; MPS/MC21 expansion vs. MPS/MC21 maintenance, $P = 0.03$). Significant differences between the MPS group and other groups are indicated on the figure. (g) Proportion of effector CD8⁺ T cells (CD44⁺ CD62L⁻) in blood over time. (h) Proportion OVA-tetramer⁺ of CD8⁺ T cells in peripheral blood 20 days after vaccination. For f-h, means depicted; error bars, s.d. For f-g, statistical analysis was performed using a Kruskal-Wallis test with Dunn’s post hoc test (day 6 timepoint) or analysis of variance (ANOVA) with Tukey’s post hoc test (days 8, 14, 20). Only differences between one group and all other groups are shown (* $P < 0.05$, ** $P < 0.01$, *** $P < 0.001$, **** $P < 0.001$). For h, statistical analysis was performed using analysis of variance (ANOVA) with Tukey’s post hoc test.

Supplementary Figure 21. Immune intervention alters maintenance of LN expansion after vaccination. (a) Timeline of treatments. Mice were treated with MPS vaccines (containing GM-CSF, CpG, and OVA) on day 0 and administered depleting reagents: MC-21 antibody (to deplete Ly6C^{hi} inflammatory monocytes), clodronate liposomes (to deplete macrophages/monocytes), anti-CD4/CD8/B220 (to deplete T and B lymphocytes), and anti-Ly6G/rat K (to deplete neutrophils). Lymph node expansion kinetics were tracked using HFUS imaging. LN expansion curves are shown for mice treated with (b) MPS vaccine alone, (c) MPS vaccine plus MC-21 (days 1-5), (d) MPS vaccine plus MC-21 (days 10-14), (e) MPS vaccine plus clodronate liposomes (days 9, 12, 15, 18), (f) MPS plus anti-CD4/CD8/B220 antibodies, (g) MPS plus anti-Ly6G and anti-rat kappa Ig, or (h) MPS vaccine plus MC-21 (days 0-4) and clodronate liposomes (days 0, 3, 5). Statistical significance is shown between LN volumes on day 11-12 and 23-25. Means depicted; error bars, s.d.

Discussion:

Strikingly, Ly6C^{hi} inflammatory monocytes rapidly increased in LNs following ‘strong’ vaccination and remained elevated throughout LN expansion. These monocytes upregulated antigen-presentation related genes in the MPS relative to the bolus vaccine group, supporting a role in antigen presentation²². Monocytes can also affect the quality of T cell response (e.g. effector phenotype) through cytokine secretion and T cell co-localization, and mobilize in response to type I adjuvants including CpG³². Notably, early depletion of inflammatory monocytes delayed effector CD8⁺ T cell responses and antigen specificity, supporting these results and suggesting a functional role of these cells in the vaccine response. Although depletion of monocytes alone did not affect the kinetics or magnitude of LN expansion, co-depletion with macrophages prevented long-term maintenance of LN enlargement, additionally implicating these antigen-presenting cell subsets in sustained vaccine responses.

5. The authors should consider introducing more different antigens and designing more reasonable tumor models to verify the universality of “jump-start” strategy. In the “Jump-start + bolus vax” strategy, the optimal timing of bolus vax administration should be demonstrated and discussed.

Previously, the bolus vaccine was administered 7 days following the MPS “jump-start” to match the peak of LN expansion. We have now performed an additional experiment to assess the optimal timing of bolus vaccine administration. Mice were injected with “jump-start” (MPS without antigen) followed by the bolus vaccine after either 11, 7, or 4 days and compared to mice receiving the bolus vaccine alone or naïve controls. Antigen-specific T cell response was measured in the blood 8 days after the bolus vaccine.

Our new results demonstrate that the bolus vaccine T cell response is enhanced by an MPS “jump-start” dosed 4, 7, or 11 days prior. Interestingly, different immunologic indicators of T cell antigen specificity and function varied based on this timing. Spacing the “jump-start” and bolus vaccine closer together (4 days apart) elicited the highest antigen-specific TNF α and IFN γ responses, along with the greatest OVA-tetramer binding. However, this timing also had the weakest Granzyme B response and highest PD-1 expression on CD8⁺ T cells. The opposite trends were noted with the longest spacing (11 days apart) and results were generally intermediate with the 7-day spacing initially chosen. Spacing the “jump-start” and bolus vaccine 7 days apart appeared to balance both trends, with strong antigen-specific, polyfunctional cytokine responses. These data are now included as a Supplementary figure, referenced in the text and discussion, and reproduced below:

Methods section:

When testing the timing of jump-start relative to bolus vaccination, the jump-start was administered 11, 7, or 4 days prior to bolus vaccination and blood was analyzed 8 days later.

Results section:

In these experiments, the jump-start was dosed 7 days prior to the bolus vaccine to match the peak of LN enlargement (**Supplementary Fig. 29a**). Spacing the jump-start closer to bolus vaccination (4 days) tended to increase antigen-specific cytokine expression (IFN γ and TNF α) and OVA-tetramer binding; however, increasing the dose separation (11 days) increased Granzyme B and reduced PD-1 expression, suggesting that the timing of jump-start and bolus vaccination can alter functional T cell outcomes, and the day 7 timepoint balances both sets of outcomes (**Supplementary Fig. 29b-h**). All additional experiments were conducted with a 7-day spacing.

Discussion section:

The timing of LN jump-start prior to bolus vaccination had implications on vaccine-responding T cell functionality, as spacing the jump-start and bolus vaccine closer together increased T cell expression of inflammatory cytokines IFN γ and TNF α , but at the cost of decreased Granzyme B and increased PD-1 expression.

Supplementary Figure 29. Determining optimal timing of "jump-start" strategy to improve vaccine response. Mice were injected with an MPS no-antigen "jump-start" (GM-CSF and CpG) on days -11, -7, or -4, and then treated with a bolus vaccine (OVA protein, CpG, GM-CSF) on day 0.

Control mice were treated with a bolus vaccine (no jump-start) on day 0 or left untreated (naïve). Draining inguinal lymph nodes were imaged on day 0, and mice were bled after 8 days for T cell analysis. (a) Day 0 lymph node volumes prior to bolus injection. Quantification of intracellular TNF α (b) and Granzyme B (c) in CD8⁺ T cells after ex vivo stimulation of PBMCs with OVA SIINFEKL peptide. (d) Representative flow cytometry plots depicting CD8⁺ T cell IFN γ production after restimulation with OVA SIINFEKL peptide (above) and OVA-tetramer binding (below). Quantification of (e) IFN γ production and (f) OVA-tetramer binding. (g) PD-1 expression on CD8⁺ T cells. (h) Pie charts depicting the percentage of CD8⁺ T cells expressing 0-3 of the cytokines IFN γ , TNF α , or Granzyme B in each group following OVA peptide restimulation. Tan represents T cells with no detectable cytokine expression, blue represents cells expressing a single cytokine, yellow represents two cytokines, and red represents cells expressing all three cytokines. For a, b, e, and f, statistical analysis was performed using analysis of variance (ANOVA) with Tukey's post hoc test. For c and g, statistical analysis was performed using a Kruskal-Wallis test with Dunn's post hoc test. For a and f-g, n = 5 biologically independent animals per group; for b-e and h, n = 4-5 biologically independent animals per group. For a-c and e-g, means depicted; error bars, s.d.

The intention behind the “jump-start” strategy was to investigate whether boosting LN expansion can improve subsequent immune responses. In these studies, we used the OVA antigen that was explored throughout our manuscript. We believe investigation of different antigens and tumor models is out of the scope of our studies, and have now adjusted our language and claims to match the data presented, with the caveat that OVA is a model antigen for purpose of our study. We have addressed this consideration in our Discussion section, included below.

Discussion: Studies conducted here made use of OVA, a commonly used model antigen, and exploration of more representative antigens (e.g. melanoma neoantigens, viral targets, peptides) can supplement this work.

6. The immunological data provided by the paper should also include the responses of the spleen, and the data of ELISPOT are suggested to fully support their conclusions.

To supplement our findings in LNs, we have now performed an additional experiment, investigating spleens 7, 14, and 20 days after vaccination with the full MPS vaccine, the MPS vaccine without antigen, a bolus vaccine, or PBS for cellular analysis. These new data showcase responses of several immune cell subsets (T cells, B cells, DCs, macrophages, and monocytes) with specific focus on inflammatory monocytes after MPS immunization.

We observed that while MPS vaccination did not alter total spleen cell counts, or largely impact numbers of lymphocytes, DCs, and macrophages, inflammatory monocyte frequency and number markedly increased in the spleen around day 20, relative to a bolus vaccine or naïve mice. Interestingly, although inflammatory monocytes in the LN upregulated MHCII, this effect did not occur in the spleen. We have included these new data in the form of Supplementary figures and additions to the Results and Discussion sections.

Results:

Unlike LNs, spleens did not demonstrate cellular expansion after vaccination (**Supplementary Fig. 18a**). Although total numbers of splenic immune cells including B cells and DCs were largely unaffected by vaccination, transient increases in T cells and macrophages were detected

(Supplementary Fig. 18b-e). Notably, significantly higher numbers and proportions of inflammatory monocytes were found in MPS-vaccinated mouse spleens compared to all other conditions on day 20 (Supplementary Fig. 18f,g). These cells also remained elevated in circulation at the latest timepoint (Supplementary Fig. 18h).

Numbers of monocyte-derived DCs (CD11c and MHCII-expressing Ly6C^{hi} monocytes) were also significantly increased in the MPS-vaccinated dLN at this time compared to PBS-treated mice, or any condition in the spleen (Supplementary Fig. 19a). In the spleen, MHCII expression on inflammatory monocytes was also unaltered with vaccination (Supplementary Fig. 19b). These results indicate that Ly6C^{hi} monocytes induced by MPS vaccination may engage in antigen presentation, specifically within the LN compartment.

Supplementary Fig. 18. MPS vaccination minimally impacts splenic immune cells, with the exception of inflammatory monocytes. Mice were treated with MPS or bolus vaccines (containing GM-CSF, CpG, OVA), MPS vaccine without antigen (GM-CSF and CpG only), or PBS, and LNs were collected on days 7, 14, and 20 for cellular analysis. n = 5 biologically independent animals per group per timepoint. (a) Total cell counts in the spleen. (b) Number of T cells (CD3⁺CD19⁻) in the spleen. Statistical analyses were performed using a one-way ANOVA with Tukey's post hoc test. Differences were significant (day 14 MPS vs. bolus p = 0.02, day 14 MPS vs. PBS p = 0.01). (c) Number of B cells (CD3⁺CD19⁺) in the spleen. (d) Numbers of dendritic cells (CD3⁺CD19⁻Ly6C⁺F4/80⁻CD11c⁺MHCII⁺) in the spleen. (e) Number of macrophages (CD3⁺CD19⁻Ly6C⁺CD11b⁺F4/80⁺) in the spleen. Statistical

analyses were performed using a one-way ANOVA with Tukey's post hoc test. Differences were significant (day 14 MPS vs. bolus $p = 0.005$, day 14 MPS vs. PBS $p = 0.01$). (f) Proportion of inflammatory monocytes (CD3⁺CD19⁺Ly6C^{hi}CD11b⁺) of total cells in the spleen. (g) Number of inflammatory monocytes in the spleen. (h) Proportion of inflammatory monocytes in the blood. For a-h, means depicted; error bars, s.d. Statistical analysis was performed using analysis of variance (ANOVA) with Tukey's post hoc test for normally distributed samples, and a Kruskal-Wallis test with Dunn's post hoc test otherwise; statistical significance is shown between the MPS dLN group and other groups (* $P < 0.05$, ** $P < 0.01$, *** $P < 0.001$, **** $P < 0.001$).

Supplementary Figure 19. Monocyte-derived DCs in LNs and spleens. Mice were treated with MPS or bolus vaccines (containing GM-CSF, CpG, OVA), MPS vaccine without antigen (GM-CSF and CpG only), or PBS, and LNs were collected on day 20 for cellular analysis. (a) Number of monocyte-derived DCs from Ly6^{hi} inflammatory monocytes (CD3⁺CD19⁺CD11b⁺Ly6C^{hi}CD11c⁺MHCII⁺ cells) in the LN and spleen at day 20. Statistical analysis was performed using a Kruskal-Wallis test with Dunn's post hoc test. (b) Proportion of inflammatory monocytes expressing MHCII in the spleen on day 20. Differences between groups are not significant. For a-b, means depicted; error bars, s.d.; $n = 5$ biologically independent animals per group.

In addition, we extracted spleens from mice 103 and 485 days after MPS vaccination to study long-term adaptive immune responses. At both timepoints, antigen-specific T cells remained detectable. The day 103 data was previously included in a Supplementary Figure, for which we have added additional text to highlight its importance demonstrating long-lived, vaccine-induced T cell responses in multiple lymphoid compartments. These data are included in new Supplementary Figures, with text added to the manuscript as included below.

Results:

Strikingly, in a tumor-free setting, MPS vaccination also enhanced long-term antibody production (day 90) and splenic CD8⁺ T cell (day 103) responses as compared to the bolus vaccine, and responses associated with earlier degree of LN expansion (**Supplementary Fig. 25a-f**). Sustained inflammatory cytokine expression in splenic CD8⁺ T cells suggested a long-lived adaptive immune response in multiple lymphoid organs.

Mice tracked out to 485 days after MPS vaccination did not display changes in weight, LN or spleen cell counts, or proportions of immune cell subsets in blood or secondary lymphoid organs, although elevated OVA-specific CD8⁺ T cells remained detectable in all immune compartments investigated (**Supplementary Fig. 26a-i**).

Supplementary Figure 26. Long-term response to MPS vaccination. Mice were injected on day 0 with PBS or an MPS vaccine consisting of GM-CSF, CpG, and OVA protein. 485 days later, mice were euthanized and blood, LNs, and spleens collected for analysis. (a) Mouse weight at 485 days. Total cell counts in draining LN (b) and spleen (c). (d) OVA-tetramer binding proportion of CD8⁺ T cells. Statistical analysis was performed using a two-tailed t test. Proportions of (e) T cells (CD19⁻CD3⁺), (f) B cells (CD19⁺CD3⁻), (g) DCs (CD19⁻CD3⁻Ly6C⁺F4/80⁻CD11c⁺MHCII⁺), (h) monocytes (CD19⁻CD3⁻CD11b⁺Ly6C⁺), and (i) macrophages (CD19⁻CD3⁻Ly6C⁺CD11b⁺F4/80⁺) in the blood, LNs and spleens. n = 4 (MPS) or 5 (PBS) biologically independent animals per group; means depicted; error bars, s.d.

Throughout the manuscript, we used intracellular cytokine staining to assess antigen specificity of T cells by flow cytometry. This technique correlates well with ELISPOT analysis [Villemonteix, J., *et al. Immunity, Inflammation and Disease* 2022, <https://doi.org/10.1002/iid3.617>; Maecker, H.T., *et al. BMC*

Immunology 2005, <https://doi.org/10.1186/1471-2172-6-17>]. As a secondary strategy to confirm antigen specific of collected T cells, tetramer staining was employed. CD8⁺ T cell tetramer binding and intracellular cytokine staining were consistent between assay groups, and we believe that a third method (e.g. ELISpot), although with the potential for additional nuance, is beyond the current scope of our work.

To address these topics and the new data included here, we have added additional text to our Discussion section:

Discussion: Notably, spleens did not expand to the degree of LNs, and broad immune cell populations in the spleen were largely unchanged after MPS vaccination. Furthermore, MHCII upregulation was only observed on inflammatory monocytes in the LN, suggesting that the MPS vaccine may exert specific influence on the local draining LN, as versus systemic compartments.

7. All the data about the correlation analysis in this manuscript, R² is generally low, such as Fig. 2j, Fig. 3b, Fig. 6d, 6e, 6f, etc. Is such a linear or nonlinear relationship still accurate? Please provide a more accurate analysis.

For guidance on statistical analysis, we have now consulted a biostatistician at the Harvard Catalyst center. In accordance with the reviewer's suggestion, our statistical consultant advised representing these data in an alternative manner, given variable R² values and relatively small number of data points, as changes were largely group-specific. We have re-plotted our data in light of this feedback and altered our language to more carefully reflect the experimental results. Now, rather than calling attention to the correlation, we present our data plain and without linear regression analysis. We believe that this representation showcases the relationships between LN expansion and vaccine outcomes in an unbiased manner. In addition, to further understand correlations among our data with higher power, we have analyzed larger datasets, combining multiple experiments to draw new conclusions. Notably, LN fold expansion by day 7 correlated strongly with the effector proportion of CD8⁺ T cells in the blood. These data are now included in Supplementary Figures, and our references in the text have been edited. All instances in which these changes are relevant are included below.

Results: Importantly, LN expansion associated positively with antibody titers, CD8⁺ T cell responses, and antitumor efficacy of cancer vaccine formulations (**Fig. 6d-f, Supplementary Fig. 24a-c**). The degree of LN expansion correlated strongly with effector CD8⁺ T cell proportions following vaccination across experiments (**Supplementary Fig. 24d**). Strikingly, in a tumor-free setting, MPS vaccination also enhanced long-term antibody production (day 90) and splenic CD8⁺ T cell (day 103) responses as compared to the bolus vaccine, and responses associated with earlier degree of LN expansion (**Supplementary Fig. 25a-f**).

Fig. 6 | Lymph node expansion correlates with adaptive immunity and therapeutic outcomes of vaccination. LN fold expansion 7 days after vaccination versus d, IFN γ ⁺ CD8⁺ T cells after SIINFEKL restimulation, e, anti-OVA IgG1 titers, and f, tumor area at the latest timepoint with all mice surviving (day 21).

Supplementary Figure 25. Long-lived vaccine responses and LN expansion. (b) IgG1 titer and (c) IgG2 titer versus the maximum LN volume in each mouse. (f) OVA-tetramer⁺ CD8⁺ T cell proportion versus maximum LN volume.

Supplementary Figure 24. The adaptive, antitumor vaccine response and LN expansion. LN expansion from Fig. 6b is plotted against vaccine response data from Fig. 6 and Supplementary Fig. 18. LN fold expansion 7 days after vaccination is plotted against (a) OVA-tetramer⁺ CD8⁺ T cells, (b) anti-OVA IgG2a titers, and (c) long-term survival. (d) LN fold expansion 7 days after vaccination is plotted against the proportion of effector (CD44⁺CD62L⁻) CD8⁺ T cells in the blood 8 days after vaccination, combined across multiple experiments. Linear regression was performed and results are statistically significant.

8. In clinical practice, the subunit vaccine is usually administered in two or more doses, while all data in this paper were obtained on the basis of a single dose. Do the changes in LN expansion after multiple doses show similar characteristics? Please provide relevant data.

We have now conducted a new experiment to investigate the response of LNs to a booster vaccine. Mice were administered the MPS vaccine on day 0, and their LNs were imaged throughout the course of expansion. By 42 days after vaccination, LNs had significantly contracted, and so the following day was selected for a booster dose adjacent to the same draining LN. Draining LNs retained a capacity for dramatic expansion following booster immunization. Interestingly, LN expansion after the boost was more rapid than after the prime vaccine, peaking within 4 days rather than 7-15 days. The strength of adaptive immune response after booster vaccination was also significantly stronger (increased T cell antigen specificity and antibody responses) than with the prime alone. We have now included these data in a new Supplementary Figure with the accompanying text:

Results:

A prime-boost vaccine format accelerates LN expansion and improves adaptive immune responses.

Finally, the impact of a booster vaccine format was assessed on LN expansion kinetics and adaptive immune responses. Following the MPS prime vaccine, draining LN volume increased over the following one to two weeks and declined by day 42 (**Supplementary Fig. 31a,b**). On day 43, a booster MPS vaccine was delivered, and this led to more immediate LN expansion, reaching peak volumes within four days, compared to day 7 with the initial vaccine. Seven days after the booster vaccine,

peripheral blood was collected and compared to mice that had received only prime vaccination at the same timepoint as the boost in the prime-boost group. No differences in the IFN γ ⁺ proportion of CD8⁺ T cells after OVA peptide restimulation were detected; however, the IFN γ ⁺ proportion of CD4⁺ T cells was significantly increased relative to both naïve control mice and mice that had received only prime vaccination (**Supplementary Fig. 31c,d**). The proportion of effector-phenotype (CD44⁺CD62L⁻) CD8⁺ T cells was elevated with the MPS prime, and further increased after the booster (**Supplementary Fig. 31e**). Both IgG1 and IgG2a titers against OVA were increased after the booster dose, compared to either the same mice on day 21 (pre-boost) or the prime-only mice at the same timepoint (**Supplementary Fig. 31f-g**). These results indicate that a booster vaccine may elicit more rapid LN expansion along with a stronger adaptive immune response.

Supplementary Fig. 31. Booster MPS vaccination expands lymph nodes and improves adaptive immune responses. (a) Timeline. Mice were immunized with MPS vaccines (GM-CSF, CpG, and OVA

protein). Draining LNs were longitudinally imaged using HFUS. Once LNs had contracted in size, a booster MPS vaccine was injected adjacent to the same dLN on day 43. On the same day, a new group of mice were injected with an MPS vaccine (“prime only”). After 7 and 21 days, blood was collected for T cell and antibody analysis, respectively. (b) Lymph node volume over time. The red arrow indicates the timing of booster vaccination. (c) Proportion of IFN γ -expressing CD8 $^+$ T cells after stimulation with OVA peptides. (c) Proportion of IFN γ -expressing CD4 $^+$ T cells after stimulation with OVA peptides. (e) Proportion of effector-type (CD44 $^+$ CD62L $^-$) CD8 $^+$ T cells. IgG1 (f) and IgG2a (g) titers against OVA.

Minor issues:

1. In the section “Vaccine formulation alters the magnitude and duration of lymph node expansion”, the data mentioned in the sentence “NdLNs did not change in volume with ether vaccine (Fig. 1c)”, should be Fig. 1d.

We have now corrected our text to reflect the appropriate figure (Fig. 1d).

2. Fig. 3c, 3e and Fig. 7b, 7c show almost the same data. Please check the full text for similar errors.

We have now removed the duplicate figure (Fig. 3e and former Supplementary Fig. 7b), which was included to show B cells alongside other immune cells in the supplementary figure. Now, B cells, T cells, and DCs are shown in the main figure while their subsets (conventional DCs, plasmacytoid DCs, CD4 $^+$ and CD8 $^+$ T cells) are included in the supplementary figure, now Supplementary Fig. 10. Fig. 3f (FDCs) was also shown in the Supplementary Figure and this second reference has similarly now been removed. These changes are now reflected in the figure legends and text.

3. Please adjust the alphabetical order in Fig.6.

We have now re-adjusted Fig. 6 so that the sub-figures are organized in alphabetical order. We have made this adjustment in additional figures (e.g. Supplementary Fig. 25) as well.

Reviewer #3 (Report for the authors (Required)):

This paper investigates the effect of the MPS vaccine adjuvant system, previously developed by the authors, on the expansion of the draining lymph node. While this study contains an

impressive body of work and reports interesting findings, it lacks proper control groups to support solid claims.

We thank the reviewer for their comments and suggestions, and appreciate the opportunity to improve our manuscript following their recommendations below.

More specific comments:

• The authors compare their own MPS formulation to a soluble mixture of GM-CSF/CpG without providing a clear rationale for this choice. Figure 6A suggests that the adjuvant effect of the soluble mixture of OVA+GM-CSF/CpG is very low, as it induces similar IgG2 levels as OVA+PBS. It would be more relevant to compare MPS to established depot systems such as alum and an o/w emulsion, and investigate whether the latter also induce long-term lymph node expansion.

The selection of the soluble GM-CSF and CpG mixture was selected to deliver the same active vaccine components as the MPS vaccine but in a rapid-release bolus format. We have now conducted additional experiments comparing the MPS vaccine to more commonly used formulations (Alum and emulsion-based vaccines) and an alternative controlled-release vaccine formulation (a polymer-based cryogel) to contextualize these results. These data are represented in new figures and text in the methods, results, and discussion sections.

Methods:

Alternative vaccine formulations. Alum and emulsion-based vaccines were formulated following manufacturer protocols. Briefly, alum-based vaccines were prepared by mixing 2% (20mg/mL) Alhydrogel adjuvant (Invivogen) in a 1:1 volume ratio with 200µg OVA protein antigen to reach a total volume of 100µL per vaccine dose. The MF59 oil-in-water emulsion vaccine was prepared by mixing AddaVax (Invivogen) in a 1:1 volume ratio with 200µg OVA protein antigen to reach a total volume of 100µL per vaccine dose. Both vaccines were injected s.c. through an insulin syringe.

Results:

While other published depot-based vaccine formulations including alum, MF59 emulsion, and cryogel-based scaffolds also induced LN expansion, expansion was notably lower than with the MPS vaccine (**Supplementary Fig. 3c**).

Supplementary Figure 3. Comparing lymph node expansion with MPS and alternative vaccine formulations. Mice were immunized with MPS vaccines (GM-CSF, CpG, and OVA protein). Additional mice were immunized with the same quantity of OVA antigen in alum, emulsion (MF59), or alginate cryogel-based vaccines (cryogel vaccines also contained GM-CSF and CpG in equivalent amounts to the MPS vaccine). Vaccine-draining LNs were longitudinally imaged using high frequency ultrasound. (c) Vaccine draining LN fold expansion comparing MPS, alum, MF59 emulsion, and cryogel formulations. Statistical analysis was performed using a one-way ANOVA with Dunnett's post hoc test (t= 7, 11, 21 days) or a Kruskal-Wallis test with Dunn's post hoc test (t= 4 days). P values for the comparison between MPS and other vaccine systems are displayed on the plot. n = 4 (full MPS) or 5 (other groups) biologically independent animals per group.

Discussion:

The MPS vaccine used here as a model of robust, persistent vaccination elicited a ~7-fold increase in dLN volume that was maintained for weeks. This is a striking outcome compared to reported vaccine formulations (~4-5-fold increase in mass or size, and contracting from day 7-14)^{2,29-34}. In a recent study, popliteal LNs expanded ~10-fold to vaccination with complete Freund's adjuvant, although this response was not explored beyond 14 days¹⁰. Although clinically successful vaccines such as those against SARS-CoV-2 have induced lymphadenopathy in a subset of patients, presenting potential discomfort, these responses largely resolved naturally, and could point to a productive adaptive immune response^{16,17,35}. In our experiments, the MPS vaccine promoted significantly greater LN expansion than commonly used vaccine delivery methods including Alum and MF59, an emulsion-based vaccine. The MPS vaccine also drove stronger LN enlargement than cryogel-based scaffold vaccines fabricated from alginate, an interesting result given the identical vaccine components in these two vaccines, suggesting that the selection of biomaterial itself can impact immune outcomes. This result is not unexpected given that silica can activate inflammasome activity, while alginate is widely used for its biocompatibility and lack of immunogenicity^{36,37}. The prolonged LN expansion observed here likely derives from persistence of the MPS vaccine and its sustained release of antigen and adjuvant¹⁸. Here, MPS vaccines without sustained antigen presentation, either antigen-free or with antigen rapidly released as a bolus, did not maintain LN expansion relative to when antigen was released in a sustained manner from MPS. Removing either CpG or GM-CSF from the vaccine also diminished LN expansion.

• The authors should also provide longitudinal data on antibody and T-cell responses against OVA or a more relevant antigen, as well as the extent of GC B cell formation in the dLN.

Previously, we have characterized antibody and germinal center B cell formation in response to the MPS vaccine [Dellacherie, M., *et al. Advanced Functional Materials* 2020, <https://doi.org/10.1002/adfm.202002448>]. The MPS vaccine induces robust and long-lived antibody responses against diverse antigens along with potent germinal center formation, identified through flow cytometry and immunohistochemistry. We also performed this analysis with another sustained-release vaccine system using poly(lactide-co-glycolide), finding similar results [Najibi, A., *et al. Advanced Functional Materials* 2022, <https://doi.org/10.1002/adfm.202110905>]. Because germinal center formation has been well characterized in these systems, we now highlight these findings and provide more thorough background on the MPS vaccine system to add context to our current results. These changes are included below.

Results:

Mesoporous silica (MPS) rod-based vaccines, previously found to elicit strong cellular and humoral responses against diverse antigen targets compared to a traditional bolus (liquid) vaccine, were explored^{18–21}. These high-aspect ratio, silica-based nanoparticles can adsorb vaccine antigens and adjuvants for sustained release, and form a three-dimensional scaffold promoting antigen-presenting cell recruitment in mouse models. MPS vaccines previously induced potent and long-lived germinal center responses dependent on sustained antigen release from the vaccine site^{22,23}. Here, MPS rods used in vaccine formulation had an average length of 85.9 μ m and released vaccine components CpG and GM-CSF in a sustained manner (**Supplementary Fig. 1a-e**).

To supplement this background in the context of our current work, we have now included new experimental results testing germinal center B cell numbers in the lymph nodes following vaccination. We found that the MPS vaccine group led to the highest percentage and number of germinal center (GL7⁺) B cells in LNs, compared to the bolus vaccine, the MPS vaccine without antigen, or PBS. These germinal center B cells expanded over time to the highest number by day 20 only in the MPS group. These results suggest that the MPS vaccine generates robust and persistent germinal center B cell responses, supporting prior results using this system. Furthermore, our single-cell RNA sequencing data also indicate formation of plasma cells in the MPS group, and we have extended our interpretation in the results and discussion sections.

Results:

Consistent with scRNAseq data and prior investigation on the MPS vaccine system, MPS immunization elicited robust and persistent germinal center B cell responses, also dependent on the presence of antigen in the vaccine (**Supplementary Fig. 17a-c**).

Supplementary Fig. 17. MPS vaccination induces potent germinal center B cell responses in the LN. Mice were treated with MPS or bolus vaccines (containing GM-CSF, CpG, OVA), MPS vaccine without antigen (GM-CSF, CpG only), or PBS, and LNs were collected on days 7, 14, and 20 for cellular analysis. n = 5 biologically independent animals per group per timepoint. (a) Representative flow cytometry plots depicting CD19⁺ B cells from LNs on day 20. GL7⁺ B cells (germinal center B

cells) are gated. (b) Proportion GL7^{hi} of B cells in LNs over time. (c) Number of GL7^{hi} B cells (i.e. germinal center B cells) in LNs over time. For b, statistical analyses were performed using analysis of variance (ANOVA) with Tukey's post hoc test (days 7 and 14) or Kruskal-Wallis test with Dunnett's post hoc test (day 20). For c, statistical analyses were performed using analysis of variance (ANOVA) with Tukey's post hoc test (day 14) or Kruskal-Wallis test with Dunnett's post hoc test (days 7 and 20). Means depicted; error bars, s.d. Only differences present between any group and the PBS control group are depicted (*P < 0.05, ** P < 0.01, ***P < 0.001, ****P < 0.001).

We also now highlight existing data and include new data demonstrating longitudinal and long-term T cell and humoral immune analysis that can supplement these findings. Serum antibody levels were assessed 39 and 90 days after vaccination, and were increased most strongly in the MPS group. We have also now tested antigen-specific T cell responses 20, 103, and 485 days after vaccination, in the latter case examining long-term responses of MPS-vaccinated mice across blood, LN, and spleens. At all timepoints, the MPS vaccine group had detectable OVA-specific CD8⁺ T cells. These data are incorporated in both Supplementary figures and main figures.

Results: Mice tracked out to 485 days after MPS vaccination did not display changes in weight, LN or spleen cell counts, or proportions of immune cell subsets in blood or secondary lymphoid organs, although elevated OVA-specific CD8⁺ T cells remained detectable in all immune compartments investigated (**Supplementary Fig. 26a-i**).

Supplementary Figure 26. Long-term response to MPS vaccination. Mice were injected on day 0 with PBS or an MPS vaccine consisting of GM-CSF, CpG, and OVA protein. 485 days later, mice were euthanized and blood, LNs, and spleens collected for analysis. (d) OVA-tetramer binding proportion of CD8⁺ T cells. n = 4 (MPS) or 5 (PBS) biologically independent animals per group; means depicted; error bars, s.d. Statistical analysis was performed using a two-tailed t test.

Results: Strikingly, in a tumor-free setting, MPS vaccination also enhanced long-term antibody production (day 90) and splenic CD8⁺ T cell (day 103) responses as compared to the bolus vaccine, and responses associated with earlier degree of LN expansion (**Supplementary Fig. 25a-f**).

Supplementary Figure 25. Long-lived vaccine responses and LN expansion. Mice were immunized with MPS or bolus vaccines delivering GM-CSF, CpG, and OVA protein (same setup as Fig. 1). 39 and 90 days after vaccination, blood was collected for OVA-specific serum antibody titer analysis using ELISA. 103 days after vaccination, mice were euthanized and spleens were collected for T cell analysis. (a) Serum IgG1 (left) and IgG2a (right) titers against OVA. $n = 7-8$ biologically independent animals per group; means depicted; error bars, s.d. Statistical analysis was performed using a Kruskal-Wallis test with Dunn's post hoc test. Proportion (left) and number (right) of OVA-tetramer⁺ CD8⁺ T cells (d) and IFN γ ⁺ CD8⁺ T cells (e) in spleens. For d-e, $n = 5-6$ biologically independent animals per group. Whiskers extend min-to-max with box displaying 25th to 75th percentiles. Statistical analysis was performed using analysis of variance (ANOVA) with Tukey's post hoc test (d, e right) or Kruskal-Wallis test with Dunn's post hoc test (e left).

Altogether, these new data and additional background have been incorporated into a new paragraph in our discussion section highlighting adaptive immune responses to vaccination, with specific focus on germinal center B cells, included below.

Discussion:

The MPS vaccine elicited strong and persistent humoral and cellular immune responses, consistent with prior findings^{18,22}. By day 20, the MPS formulation, in contrast to the bolus vaccine, led to mature Ig expression by LN plasma cells. In addition, GL7⁺ B cells were observed to expand in MPS-vaccinated mouse LNs, dependent on the presence of antigen. Our finding of heightened antibody titers for months after vaccination additionally supports these findings. Antigen-specific CD8⁺ T cells were also detectable 8, 20, 103, and up to 485 days after vaccination in blood, spleens, and LNs. Altogether, these results position the MPS vaccine as a potent inducer of both adaptive and humoral arms of immunity, also associated with strong LN expansion.

• From an immunological standpoint, I wonder whether the observed long-term response in the lymph nodes is actually desirable. Could prolonged innate activation in the dLN stimulate T-cell energy, and could excessive upregulation of immune checkpoints such as PD-L1 be a concern? This should be addressed by new experimental data.

To address this question, we have added new figures and text to our results and discussion sections with insights from existing and new experimental data.

First, long-term LN stimulation with the MPS vaccine did not lead to hallmarks of T cell dysfunction or anergy. Within 8 days of vaccination, we observed PD-1 upregulation in blood CD8⁺ T cells, indicative of immune activation. However, analysis of the spleen at a later timepoint (103 days after vaccination) indicated no difference in PD-1 expression on either CD8⁺ or CD4⁺ T cells. These T cells remained functional, capable of IFN γ production after *ex vivo* stimulation with target antigen. This result suggests that even after robust LN engagement and antigen-specific T cell response by the MPS vaccine, T cells can return to homeostasis, yet retain functional memory.

Supplementary Figure 22. Therapeutic study to assess correlations of LN expansion with vaccine efficacy. (g) PD-1 expression on CD8⁺ T cells. Statistical analysis was performed using analysis of variance (ANOVA) with Tukey's post hoc test. For c and e-g, means are depicted; error bars, s.d.

Results: PD-1 expression on splenic T cells was not different between the MPS vaccine group and PBS controls (Supplementary Fig. 25h-i).

Supplementary Figure 25. Long-lived vaccine responses and LN expansion. Mice were immunized with MPS or bolus vaccines delivering GM-CSF, CpG, and OVA protein (same setup as Fig. 1). 103 days after vaccination, mice were euthanized and spleens were collected for T cell analysis. Proportion (left) and number (right) of OVA-tetramer⁺ CD8⁺ T cells (d) and IFN γ ⁺ CD8⁺ T cells (e) in spleens. For d-e, n = 5-6 biologically independent animals per group. Whiskers extend min-to-max with box displaying 25th to 75th percentiles. Statistical analysis was performed using analysis of variance (ANOVA) with Tukey's post hoc test (d, e right) or Kruskal-Wallis test with Dunn's post hoc test (e left). (h) PD-1 expression on CD8⁺ T cells. (i) PD-1 expression on CD4⁺ T cells. Statistical analysis was performed using analysis of variance (ANOVA) with Tukey's post hoc test; no significant differences were found between groups. n = 5-6 biologically independent animals per group; means depicted; error bars, s.d.

Additional evidence supporting the idea that the MPS vaccine does not exhaust T cells comes from the MPS 'jump-start' experiments, in which mice were initially exposed to adjuvant without antigen (a situation potentially more likely for anergy). However, by 11 days after jump-start stimulation, PD-1 expression on CD8⁺ T cells was not statistically different from a bolus vaccine alone or naïve controls, again suggesting that PD-1 expression is short-lived.

Supplementary Figure 29. Determining optimal timing of “jump-start” strategy to improve vaccine response. Mice were injected with an MPS no-antigen “jump-start” (GM-CSF and CpG) on days -11, -7, or -4, and then treated with a bolus vaccine (OVA protein, CpG, GM-CSF) on day 0. Control mice were treated with a bolus vaccine (no jump-start) on day 0 or left untreated (naïve). Draining inguinal lymph nodes were imaged on day 0, and mice were bled after 8 days for T cell analysis. (g) PD-1 expression on CD8⁺ T cells. Statistical analysis was performed using a Kruskal-Wallis test with Dunn’s post hoc test. n = 5 biologically independent animals per group; means depicted; error bars, s.d.

We have also conducted new experiments testing MPS-vaccinated mice in the long-term (~16 months). At this time, we found no differences in T cell frequencies in the blood, lymph node, or spleen, although T cells capable of binding OVA tetramer remained detectable. These results further suggest that the T cell compartment is not adversely affected in the long-term after MPS vaccination.

Results: Mice tracked out to 485 days after MPS vaccination did not display changes in weight, LN or spleen cell counts, or proportions of immune cell subsets in blood or secondary lymphoid organs, although elevated OVA-specific CD8⁺ T cells remained detectable in all immune compartments investigated (**Supplementary Fig. 26a-i**).

Supplementary Figure 26. Long-term response to MPS vaccination. Mice were injected on day 0 with PBS or an MPS vaccine consisting of GM-CSF, CpG, and OVA protein. 485 days later, mice were euthanized and blood, LNs, and spleens collected for analysis. (d) OVA-tetramer binding proportion of

CD8⁺ T cells. Statistical analysis was performed using a two-tailed t test. (e) Proportion of T cells (CD19⁺CD3⁺) in the blood, LNs and spleens. n = 4 (MPS) or 5 (PBS) biologically independent animals per group; means depicted; error bars, s.d.

In addition, we have conducted new experiments to assess functional persistence of T cell immunity after MPS vaccination. Mice were vaccinated with MPS vaccines against OVA protein and then challenged 50 days later with B16-OVA melanoma tumor cells. The vaccine effectively prevented tumor outgrowth in all mice, while mice vaccinated with the MPS vaccine without antigen succumbed to tumor progression. These results further indicate that T cells elicited in the context of massive LN expansion after MPS vaccination retain memory and functional potential.

Results: MPS vaccine-generated T cells also retained functional, antigen-specific antitumor response when challenged 50 days after immunization (**Supplementary Fig. 27a-c**).

Supplementary Figure 27. MPS vaccine protects against tumor challenge. (a) Timeline. Mice were injected on day 0 with an MPS vaccine consisting of GM-CSF, CpG, and OVA protein, or a control vaccine without antigen (“MPS, no ag”). 50 days later, B16-OVA tumors were inoculated and their outgrowth was measured. n = 5 biologically independent animals per group. (b) Tumor growth curves. Statistical analysis was performed using a Mann-Whitney test. Means depicted; error bars, s.d. (c) Kaplan-Meier curves depicting survival of groups. Statistical analysis was performed using a log-rank (Mantel-Cox) test.

Furthermore, we have further referenced the scRNAseq dataset for indications of immune cell exhaustion 20 days after vaccination. No antigen-presenting cell subset demonstrated increases in *Cd274* (PD-L1) gene expression in the MPS group relative to naïve or bolus-treated mice. These results also suggest that robust MPS vaccine stimulation does not prohibitively exhaust the immune system.

Results: Despite robust LN expansion and immune engagement, *Cd274* (PD-L1) was not notably upregulated on myeloid cell subsets 20 days after immunization (**Supplementary Fig 14a,b**).

Supplementary Figure 14. scRNAseq gating analysis of Cd274 (PD-L1) expression. (a) Expression of Cd274 (PD-L1) among cells across conditions from Fig. 3h. (b) Expression level of Cd274 among myeloid cell subsets. n = 5 biologically independent animals per group.

Altogether, these results suggest that although the MPS vaccine robustly expands LNs and promotes a strong T cell response, no compensatory negative effects on T cell functionality appear to result. All of this new data has been incorporated into supplementary figures as reproduced above, and we have added text to our discussion section included below.

Discussion:

Although robust and maintained immune engagement could have the potential to induce T cell exhaustion or anergy, as previously suggested⁴¹, the data in this manuscript do not suggest this occurs with the MPS vaccine. Here, although PD-1 was upregulated on CD8⁺ T cells within 8 days after vaccination, these levels returned to baseline within weeks. In addition, 20 days after vaccination, no evidence of PD-L1 gene upregulation was detected on myeloid cells. Splenic T cells stimulated ex vivo 103 days after immunization remained functional, capable of antigen-specific cytokine secretion, and evidence of tetramer binding was found out to 485 days. The functional capacity of the immune system following vaccination was further supported by the ability of vaccine-induced T cells to restrain tumor challenge after 50 days. These results fall in line with efforts to extend vaccine delivery and LN targeting to improve adaptive immune responses⁴²⁻⁴⁴, and indicate that vaccine-induced T cells retain functional memory potential for months to years.

• From a translational standpoint, is there any risk of patient discomfort?

Extreme LN expansion could potentially come at the cost of patient discomfort. The vaccine response of LN expansion (i.e. lymphadenopathy) mirrors the LN swelling response to infection that can be detectable by touch. Certain patients receiving SARS-CoV-2 mRNA vaccines (estimated 3-16% [Kaya A., et al. *J. Nat. Med. Assoc.* 2023, <https://doi.org/10.1016/j.jnma.2023.01.003>]) report palpable, enlarged LNs that may feel sensitive or uncomfortable. However, these responses are typically not prolonged and resolve within days or weeks, or upon drug treatment otherwise (e.g. steroids) [Polack, F.P., et al. *NEJM* 2021 doi: 10.1056/NEJMoa2034577; Mehta, N. et al. *Clinical Imaging* 2021

<https://doi.org/10.1016/j.clinimag.2021.01.016>; Heaven, C.L. *et al.* *ANZ J Surg* 2022
<https://doi.org/10.1111/ans.17808>]. We expect that the MPS vaccine, being a strong vaccine, may lead to similar LN expansion that is detectable by humans, yet with a similar rate of resolution. Mice throughout our studies demonstrated both a dynamic expansion and later contraction of LN volume after immunization. We have now included additional text considering this point in the Discussion section of our manuscript.

Discussion: Although clinically successful vaccines such as those against SARS-CoV-2 have induced lymphadenopathy in a subset of patients, presenting potential discomfort, these responses largely resolved naturally, and could point to a productive adaptive immune response^{16,17,38}.

Reviewer #4 (Report for the authors (Required)):

Najibi et al. demonstrate that a formerly described mesoporous silica (MPS) rod-based vaccine induces a more potent and persistent LN expansion than standard immunization with the same antigen and adjuvants in liquid form, termed “bolus” vaccine. They also describe changes in the mechanics and cellularity of the LN that accompany the expansion. The authors then use the comparison between the two vaccine formulations to examine the impact of LN expansion on the quality and duration of the induced immune response. Data is presented that demonstrates that conventional dendritic cells and inflammatory monocytes were mainly found in long-term expanded LN. These factors correlated with an increased adaptive immune response and enhanced magnitude and efficacy of a therapeutic cancer vaccination.

The approach used by the authors allows to investigate how LN expansion is important for immune responses and efficacy of vaccines, as they compare different formulations of the same vaccine, with no change in antigen, adjuvant, dose or application route. The data presented is very extensive and convincing, with state-of-the-art techniques to characterize in depth the changes in LN morphology and function and the impact on immune cell phenotype and

function. Although LN expansion has been well characterized, including factors responsible for the expansion, the authors show novel findings with respect to how LN expansion may contribute to vaccine efficacy. This may have a therapeutic impact on vaccine development.

We thank the reviewer for their assessment and constructive comments, which we have addressed below.

Minor comments:

1. Page 2: Fig1c does not show ndLN, please correct in text.

We have now corrected our text to reflect the appropriate figure (Fig. 1d).

2. To compensate for differences in LN size between individual mice, the authors could consider expressing the LN size as a percentage of the non-draining lymph nodes.

We agree with the reviewer's suggestion that non-draining LNs could serve as an appropriate control, given that no changes in ndLN volume were observed following vaccination. For the longitudinal LN-tracking study reported in Figure 1, we now include a Supplementary figure for one cohort of mice whose dLNs and ndLNs were both measured across multiple timepoints. These data, and new changes to the text, are included below.

Results: NdLNs did not change in volume with either vaccine, and normalizing the dLN to ndLN volume within each mouse indicated a similar pattern of dynamic LN expansion and contraction (Fig. 1d, Supplementary Fig. 2e).

Supplementary Figure 2. MPS vaccination induces robust, prolonged LN expansion. Mice were immunized with MPS or bolus vaccines delivering GM-CSF, CpG, and OVA protein, and compared to PBS-injected controls. Vaccine-draining and non-draining LNs were longitudinally imaged using high frequency ultrasound. (e) Draining LN volume over time, normalized to the volume of the contralateral non-draining LN in the same mouse. n = 4 biologically independent animals per group.

For experiments in which ndLN volumes were not measured, we now express LN expansion data in the form of fold expansion relative to the day-0 timepoint, which can also serve as an internal control within each mouse. This has resulted in changes to Supplementary figures, reproduced below. New data added to the manuscript have also been normalized in this manner.

Supplementary Figure 22. Therapeutic study to assess correlations of LN expansion with vaccine efficacy. Mice were inoculated with B16-OVA tumors and three days later treated with MPS or bolus vaccines containing GM-CSF, CpG, and OVA protein, and compared to PBS-injected controls. A fourth group of tumor-free mice was treated with MPS vaccines (called “MPS, no tumor”). Inguinal dLNs were imaged using HFUS at multiple timepoints. n = 5-6 biologically independent animals per group. **(c) LN fold expansion over time, relative to the day 0 timepoint.**

Supplementary Figure 12. Comparing lymph node expansion with MPS and vaccines lacking sustained antigen release. Mice were immunized with MPS vaccines (GM-CSF, CpG, and OVA protein), MPS vaccines without antigen (GM-CSF and CpG alone), or MPS vaccines with bolus antigen (GM-CSF and CpG loaded into MPS, with OVA injected separately as a bolus). (a) LN fold expansion over time, normalized to volume at the day 0 timepoint. n = 5 biologically independent animals per group; means depicted; error bars, s.d. (b) Vaccine draining LN fold expansion over time, comparing the full MPS vaccine (antigen loaded in MPS material) to an MPS vaccine with a separate bolus injection of antigen. n = 4 (full MPS) or 5 (bolus antigen) biologically independent animals per group; means depicted; error bars, s.d.

Supplementary Figure 3. Comparing lymph node expansion with MPS and alternative vaccine formulations. Mice were immunized with MPS vaccines (GM-CSF, CpG, and OVA protein), MPS vaccines without CpG (GM-CSF and OVA), or MPS vaccines without GM-CSF (CpG and OVA). Additional mice were immunized with the same quantities of GM-CSF, CpG, and OVA in alum, emulsion (MF59), or alginate cryogel-based vaccines. A final group was dosed with an MPS vaccine containing a log-fold lower dose each of CpG and OVA. Vaccine-draining LNs were longitudinally imaged using high frequency ultrasound. (a) Quantification of vaccine draining LN fold expansion over time, comparing the full MPS vaccine with formulations lacking either CpG or GM-CSF. Statistical analysis was performed using a one-way ANOVA with Dunnett's post hoc test. $n = 4$ (full MPS) or 5 (other groups) biologically independent animals per group. (b) Vaccine draining LN fold expansion over time, comparing the full MPS vaccine with a low-dose MPS vaccine. $n = 4$ (full MPS) or 10 (low-dose MPS) biologically independent animals per group. The low-dose MPS group displays the combined results of two separate experiments. (c) Vaccine draining LN fold expansion comparing MPS, alum, MF59 emulsion, and cryogel formulations. Statistical analysis was performed using a one-way ANOVA with Dunnett's post hoc test ($t = 7, 11, 21$ days) or a Kruskal-Wallis test with Dunn's post hoc test ($t = 4$ days). P values for the comparison between MPS and other vaccine systems are displayed on the plot. $n = 4$ (full MPS) or 5 (other groups) biologically independent animals per group.

3. In fig 2 the authors show several parameters relating to the mechanics of the LN (actin, collagen and HA). However, the text does not mention what are the implied consequences of these changes, in particular with respect to immune cell function and migration. Please explain.

We have now added additional text in the both results and discussion sections to interpret the HA and F-actin signal in the context of our broader data, provided below.

Results section:

These changes suggest that LN expansion may be accompanied by changes in tissue mechanical properties, as both HA and F-actin are involved in cellular mechanotransduction and signaling pathways.

Discussion section:

HA provides both tissue scaffolding and cell signaling cues, and is regularly recycled through LNs. Its increased quantity in the LN periphery after vaccination may suggest altered recycling dynamics and could suggest a potential role in inflammatory signaling during the vaccine response. F-actin plays a key role in cytoskeletal maintenance and cell motility, and the observed pattern of F-actin localization indicates that the tissue-level mechanical changes in LNs may translate down to individual cell behavior, such as immune cell migration in the expanded node.

4. In fig 3H/I/J the authors show that they can distinguish 9 cellular populations using single cell RNAseq. However, the more important conclusion of their RNAseq follows in SI 9c, where the authors show that after MPS vaccination cell populations change in comparison to a bolus vaccination. I suggest to include SI 9c in the main figure.

We now include the former Supplementary Fig. 9c in the main figure as Fig. 3k, to more directly highlight the changes in cell populations between MPS and bolus vaccination.

Fig. 3 | MPS vaccination alters cellularity of lymph nodes. Mice were injected with MPS or bolus vaccines (GM-CSF, CpG, OVA) and dLNs were collected on at a late timepoint (day 20-21). Naïve mice were included as controls. n = 5 biologically independent animals per group, barcoded and pooled for sequencing. **k**, Frequency of individual cell clusters within each sample. Statistical analysis was performed using analysis of variance (ANOVA) with Tukey's post hoc test. n = 5 biologically independent animals per group.

5. In figure 4 the authors discuss the change in DC subpopulations but then quickly move to monocytes. The change between cell types is unexpected and lacks explanation in the text.

We have now modified our phrasing in the results section to more clearly transition between our investigation of DCs and monocytes, and to motivate our selection of these cell populations. Both DCs and monocytes demonstrated significant transcriptional changes between MPS and bolus vaccine groups, and both cell types can present antigen to LN T cells. In addition, monocytes experienced the

most dramatic cellular expansion (by proportion and number) in LNs, leading us to pay further attention to this cell type. Our changes and rationale are included in the Results section, recapitulated below:

Results:

Inflammatory monocytes also experienced significant transcriptional alterations between the MPS and bolus vaccine groups, notable given the dramatic increase in cell number in this group (**Fig. 4c, Supplementary Fig. 10n**).

Because monocytes demonstrated both transcriptional changes and the most striking expansion by both number and proportion following MPS vaccination (**Fig. 4f, Supplementary Fig. 10n**), their gene expression was next assessed in greater depth. Monocytes, like DCs, can present antigen to T cells in LNs²², and particular attention was paid to potential T cell interactions.

6. In figure 6D/E/F the authors show a correlation between LN expansion and different immunological effects. As they are comparing very different subgroups, this method of correlation can suggest a false relationship. (see for example: <https://www.ncbi.nlm.nih.gov/pmc/articles/PMC5079093/>)

We agree that these correlations were largely driven by different subgroups. For guidance on statistical analysis, we have now consulted a biostatistician at the Harvard Catalyst center. In accordance with the reviewer's suggestion, our statistical consultant advised representing these data in an alternative manner, given variable R² values and relatively small number of data points, as changes were largely group-specific. We have re-plotted our data in light of this feedback and altered our language to reflect the experimental results. Now, rather than calling attention to the correlation, we present our data plain and without linear regression analysis. We believe that this representation showcases the relationships between LN expansion and vaccine outcomes in an unbiased manner. In addition, to further understand correlations among our data with higher power, we have analyzed larger datasets, combining multiple experiments to draw new conclusions. Notably, LN fold expansion by day 7 correlated strongly with the effector proportion of CD8⁺ T cells in the blood. These data are now included in Supplementary Figures, and our references in the text have been edited. All instances in which these changes are relevant are included below.

Results: Importantly, LN expansion associated positively with antibody titers, CD8⁺ T cell responses, and antitumor efficacy of cancer vaccine formulations (**Fig. 6d-f, Supplementary Fig. 24a-c**). The degree of LN expansion correlated strongly with effector CD8⁺ T cell proportions following vaccination across experiments (**Supplementary Fig. 24d**). Strikingly, in a tumor-free setting, MPS vaccination also enhanced long-term antibody production (day 90) and splenic CD8⁺ T cell (day 103) responses as compared to the bolus vaccine, and responses associated with earlier degree of LN expansion (**Supplementary Fig. 25a-f**).

Fig. 6 | Lymph node expansion correlates with adaptive immunity and therapeutic outcomes of vaccination. LN fold expansion 7 days after vaccination versus d, IFN γ ⁺ CD8⁺ T cells after SIINFEKL restimulation, e, anti-OVA IgG1 titers, and f, tumor area at the latest timepoint with all mice surviving (day 21).

Supplementary Figure 25. Long-lived vaccine responses and LN expansion. (b) IgG1 titer and (c) IgG2 titer versus the maximum LN volume in each mouse. (f) OVA-tetramer⁺ CD8⁺ T cell proportion versus maximum LN volume.

Supplementary Figure 24. The adaptive, antitumor vaccine response and LN expansion. LN expansion from Fig. 6b is plotted against vaccine response data from Fig. 6 and Supplementary Fig. 18. LN fold expansion 7 days after vaccination is plotted against (a) OVA-tetramer⁺ CD8⁺ T cells, (b) anti-OVA IgG2a titers, and (c) long-term survival. (d) LN fold expansion 7 days after vaccination is plotted against the proportion of effector (CD44⁺CD62L⁻) CD8⁺ T cells in the blood 8 days after vaccination, combined across multiple experiments. Linear regression was performed and results are statistically significant.